# Mechanical underwater adhesive devices for soft substrates

Ziliang Kang[1,2,3], Johanna A. Gomez[1,9], Alisa MeiShan Ross[2,9], Ameya R. Kirtane[2,3,4], Ming Zhao[1,2], Yubin Cai[5], Fu Xing Chen[3], Corona L. Chen[3], Isaac Diaz Becdach[3,6], Rajib Dey[1,2], Andrei Russel Ismael[2], Injoo Moon[3], Yiyuan Yang[1], Benjamin N. Muller[1,2,6], Mehmet Girayhan Say[1], Andrew Pettinari[1,3], Jason Kobrin[1], Joshua Morimoto[3], Ted Smierciak[3], Aaron Lopes[2,3], Ayten Ebru Erdogan[2], Matt Murphy[1], Niora Fabian[1,3,7], Ashley Guevara[3], Benedict Laidlaw[1,3], Kailyn Schmidt[1], Alison M. Hayward[1,2,3,7], Alexandra H. Techet[1], Christopher P. Kenaley[8] & Giovanni Traverso[1,2,3,6 ✉]

Achieving long-term underwater adhesion to dynamic, regenerating soft substrates that undergo extreme fluctuations in pH and moisture remains a major unresolved challenge, with far-reaching implications for healthcare, manufacturing, robotics and marine applications[1–16]. Here, inspired by remoras—fish equipped with specialized adhesive discs—we developed the Mechanical Underwater Soft Adhesion System (MUSAS). Through detailed anatomical, behavioural, physical and biomimetic investigations of remora adhesion on soft substrates, we uncovered the key physical principles and evolutionary adaptations underlying their robust attachment. These insights guided the design of MUSAS, which shows extraordinary versatility, adhering securely to a wide range of soft substrates with varying roughness, stiffness and structural integrity. MUSAS achieves an adhesion-force-to-weight ratio of up to 1,391-fold and maintains performance under extreme pH and moisture conditions. We demonstrate its utility across highly translational models, including in vitro, ex vivo and in vivo settings, enabling applications such as ultraminiaturized aquatic kinetic temperature sensors, non-invasive gastroesophageal reflux monitoring, long-acting antiretroviral drug delivery and messenger RNA administration via the gastrointestinal tract.

Underwater adhesion presents a significant challenge for high-tech industries including healthcare, manufacturing, robotics and marine sectors, having a vital role in production line, exploration, sensing and clinical procedures. Despite its importance, most commercial adhesives function on dry solids, driving extensive research into underwater adhesion[1–16]. Encouraging progress has been reported in underwater polymeric adhesives[3,11–13] and in wet adhesion on tissue surfaces leading to wound repair solutions[14–16]. Mechanical adhesion platforms have also demonstrated underwater adhesion to compliant substrates but are limited to stiff or smooth surfaces and often require external actuation[5,9]. However, many soft substrates of interest, including vertebrate non-bony tissue, exhibit heterogeneous surface anatomy and extreme softness (11 Pa–1 MPa)[10]. Indeed, fundamental challenges remain in achieving long-term adhesion on dynamic, morphable soft substrates that undergo rapid regeneration and experience extreme pH and moisture conditions, for example, the gastrointestinal (GI) tract. For instance, gastric epithelial cells renew every 3 days[17], posing a significant barrier to long-term residence. Additional factors such as substrate roughness, stiffness, intactness and dynamic morphing

further complicate adhesion. Despite decades of research, most current solutions fail in vivo unless external stimuli (for example, electric fields) significantly alter the surface anatomy[18]. Therefore, adhesion strategies capable of interfacing with diverse soft substrates under extreme environmental conditions could unlock transformative applications, such as robotic-assisted wet manufacturing, aquatic exploration, drug delivery and continuous health monitoring.

Remoras (family Echeneidae) are ray-finned fish capable of adhering to diverse marine hosts, including sharks, cetaceans, turtles, boats and even divers (Extended Data Fig. 1a). Their unique adhesion ability arises from an evolutionary modification of the first dorsal fin into a sucker-like adhesive disc, comprising intercalary backbones that support plate-like lamellae lined with minute pectinate spinules, all encapsulated within soft tissue compartments (Fig. 1e and Extended Data Fig. 1a). Although previous studies have explored remora disc mechanics[19,20] and adhesion behaviours on rigid surfaces[5–8], underwater adhesion to dynamic, morphable soft substrates and interspecies disc anatomy adaptations remain underexplored. Here we comprehensively investigated remora adhesion on dynamic soft substrates through anatomical, behavioural,

[1]Department of Mechanical Engineering, Massachusetts Institute of Technology, Cambridge, MA, USA. [2]Division of Gastroenterology, Hepatology and Endoscopy, Brigham and Women's Hospital, Harvard Medical School, Boston, MA, USA. [3]David H. Koch Institute for Integrative Cancer Research, Massachusetts Institute of Technology, Cambridge, MA, USA. [4]Department of Pharmaceutics, University of Minnesota, Minneapolis, MN, USA. [5]Program in Media Arts and Sciences, Massachusetts Institute of Technology, Cambridge, MA, USA. [6]Broad Institute of MIT and Harvard, Cambridge, MA, USA. [7]Division of Comparative Medicine, Massachusetts Institute of Technology, Cambridge, MA, USA. [8]Department of Biology, Boston College, Chestnut Hill, MA, USA. [9]These authors contributed equally: Johanna A. Gomez, Alisa MeiShan Ross. ✉e-mail: cgt20@mit.edu

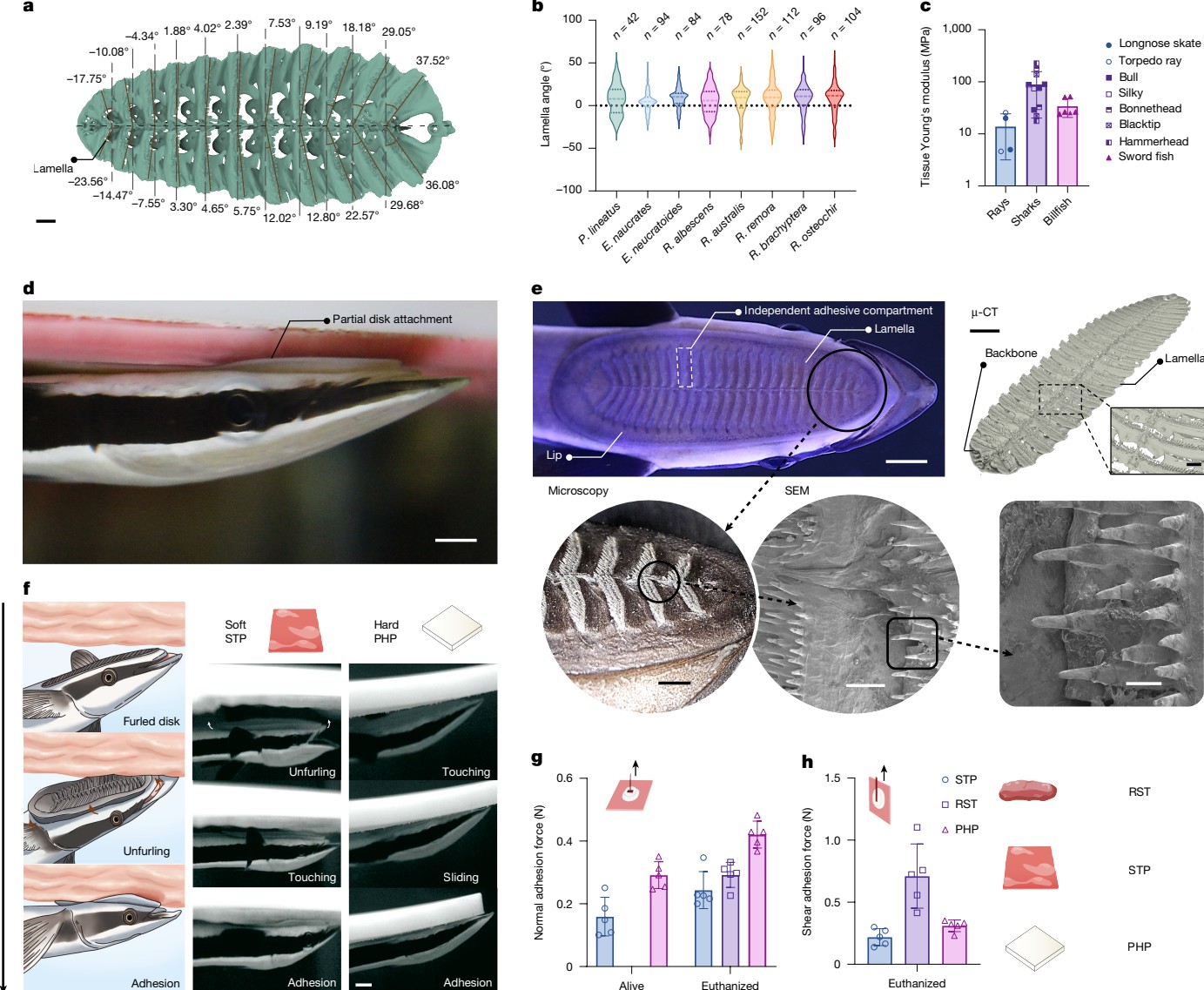

**Fig. 1 | Remoras adhere to various soft substrates. a**, Representative measurement of lamella orientation of *P. lineatus*. Scale bar, 1 mm. **b**, Remora species exhibit distinct lamella orientation distributions, mainly unimodal (for example, *R. australis*, *R. brachyptera* and *R. osteochir*), bimodal (for example, *E. neucratoides* and *P. lineatus*) and non-dominant (*R. albescens*). The dashed and dotted lines represent the median and the Q1 and Q3 quartiles, respectively (see Extended Data Fig. 1 for measurement). **c**, Host tissue stiffness of representative remora species: rays (*R. albescens* hosts, *n* = 4 statistics)[44,45], sharks (*E. neucratoides* hosts, *n* = 11 statistics)[46,47] and billfish (*R. brachyptera* hosts, *n* = 6 samples; see Supplementary Fig. 1 for details). The error bars represent mean ± s.d. **d**, Firm adhesion achieved as a live remora (*E. naucrates*) established only partial contact of its adhesive disc to an STP). Scale bar, 5 mm. **e**, Anatomy of a studied remora (*E. naucrates*) adhesive disc via high-resolution camera (scale bar, 5 mm), micro-computed tomography (μ-CT; scale bars, 5 mm (left), 500 μm (right)), microscopy (scale bar, 1 mm; *n* = 3 *E. naucrates*),

and scanning electron microscopy (SEM; 300 μm (left), 100 μm (right); *n* = 3 *E. naucrates*). **f**, Distinct remora adhesion behaviours on hard and soft substrates. Left: illustration of remora's adhesion to soft substrates via disc unfurling and mechanical interlocking for multicompartmental adhesion. Middle: high-speed image of remora's adhesion to STP by unfurling its adhesive disc. Right: remora's adhesion to polypropylene hard plate (PHP) via sliding and friction. Arrows (left and middle) indicate unfurling (also see Supplementary Video 1). Scale bar, 5 mm. **g**,**h**, Remora's adhesion performance to soft and hard substrates under normal (**g**; *n* = 5 independent experiments) and shear (**h**; *n* = 5 independent experiments) directions. The error bars represent mean ± s.d. (top left insets in **g**,**h** illustrate the normal and shear adhesion test setups, with arrows indicating the direction of the force applied to the adhesion disks (white); see Supplementary Fig. 2 for details). RST, real pig stomach tissue; STP, stomach tissue phantom; PHP, polypropylene hard plate.

physical and biomimetic analyses across remora species. Leveraging insights from these studies, we developed the Mechanical Underwater Soft Adhesion System (MUSAS)—a miniaturized, passive platform that autonomously adheres to dynamic soft substrates in harsh conditions by utilizing endogenous biomechanical forces such as GI contractions. MUSAS was rigorously validated through independent in vitro, ex vivo and in vivo testing with established animal models, facilitating applications including ultraminiaturized kinetic temperature sensing for

aquatic research, non-invasive gastroesophageal reflux disease (GERD) monitoring, sustained HIV/AIDS pre-exposure prophylaxis (PrEP) drug release and efficient messenger RNA delivery in the GI tract.

## Remoras adhere to varied soft substrates

Understanding host-specific adaptations in remora adhesive disc anatomy and adhesion strategies to soft substrates can inspire underwater

adhesion mechanisms. Remoras include nine distinct species with varying host preferences (Extended Data Fig. 1a). A reef-associated clade—*Phtheirichthys lineatus*, *Echeneis neucratoides* and *Echeneis naucrates*—are generalists, adhering to a wide range of hosts including sharks, actinopterygian fish and turtles[20,21]. In contrast, a pelagic clade—*Remora australis*, *Remora brachyptera*, *Remora osteochir* and *Remora albescens*—typically adheres to fewer host species, except *R. remora* (Extended Data Fig. 1a). For example, *R. australis* exclusively adheres to cetaceans, whereas *R. brachyptera* and *R. osteochir* primarily attach to billfish such as spearfish, marlins and swordfish[20–22]. Although most remoras attach externally to the host's body surface, *R. albescens* uniquely exhibits mucoadhesion within the oral cavity and gill chambers of manta rays (*Mobula* spp.)[23–25]. Our comparative measurements across all species revealed that disc anatomical adaptations are largely defined by lamellar orientation (Fig. 1a,b). Species associated with fast-swimming hosts (for example, billfish and cetaceans[26–29]), including *R. australis*, *R. brachyptera* and *R. osteochir*, exhibit unimodal lamellar orientation distributions, indicating mostly parallel lamellae. In contrast, generalists such as *E. neucratoides* and *P. lineatus* display bimodal distributions, with a mix of parallel and angled lamellae (Fig. 1b). The exception *R. albescens*, which adheres to ray mucosa, shows no dominant orientation pattern but spans a broad range of lamella angles. We propose that these lamella orientation diversities represent phenotypic adaptations to host swimming speed and tissue stiffness. Notably, *R. albescens*' broad lamella orientation range may best suit adhesion to soft substrates. Mechanical testing confirmed that host tissue stiffness varies significantly among representative species—*E. neucratoides*, *R. albescens* and *R. brachyptera*—with ray tissue being much softer than that of sharks and billfish (Fig. 1c).

To understand remora adhesion on soft substrates, we conducted behavioural and mechanical studies using live *E. naucrates*, owing to limited specimen availability. Adhesion tests were performed on a stomach tissue phantom (STP; LifeLike Biotissue), a realistic simulator that also addresses animal welfare concerns. Remarkably, its adhesive disc contacts the phantom only partially yet achieves firm adhesion (Fig. 1d). Our anatomical studies revealed that the disc comprises multiple independently functioning compartments, each with pectinate spinules at the lamellar tips that enhance friction and enable multicompartmental adhesion and interlocking (Fig. 1e). We also observed a distinct behaviour: on soft substrates, the remora fully unfurls its disc and actively erects its lamellae to maximize multicompartmental adhesion and interlocking—contrasting with the sliding-based adhesion used on hard surfaces (Fig. 1f and Supplementary Video 1). These findings reveal two key adhesion strategies underlying remora adhesion to soft substrates. First, robust attachment through coordinated multicompartmental adhesion and mechanical interlocking, compensating for individual compartment failure owing to dynamic morphing of substrates (Fig. 1d,e). Second, unfurling the adhesive disc to optimize hydrodynamic differentiation and multicompartmental sealing on morphable soft surfaces (Fig. 1f). We subsequently quantified adhesion forces of live remoras when feasible, supplemented with tests on euthanized specimens on various soft and hard substrates, including an STP, real pig stomach tissue (RST) and a 3D-printed polypropylene hard plate (PHP). The results confirmed that the soft-substrate adhesion strategies are effective with comparable adhesion performance on both soft and hard substrates (Fig. 1g,h). It is noted that adhesion forces in live specimens were slightly lower owing to measurement sampling rate differences and water-induced fluctuations (see Supplementary Text 6 for extended discussion).

## Understanding remora's soft adhesion

We conducted a series of physics-based simulations and biomimetic experiments to elucidate remora adhesion strategies and anatomical adaptations on soft substrates. Using finite element analysis, we modelled the hydrodynamics of multicompartmental and unfurling adhesion strategies under water. Three disc designs were compared: a furled multicompartmental disc, an unfurled multicompartmental disc and a benchmark one-piece disc (Fig. 2a). A coupled Eulerian–Lagrangian approach captured complex fluid–solid interactions among water, stomach tissue and the disc, as well as solid–solid contact (Methods and Supplementary Text 1). Notably, the unfurled multicompartmental design expelled the most water, achieving the strongest vacuum-based adhesion (Fig. 2b). This informed the design of MUSAS, a capsule-sized platform mimicking the unfurled multicompartmental remora disc. Optimized for soft tissue adhesion in kinetic biosensing and GI-targeted delivery (Fig. 2c), MUSAS integrates tilt-angled shape memory alloy (SMA) lamellae inspired by *R. albescens*, supported by a stainless-steel backbone and soft elastomeric lips forming discrete compartments (Fig. 2d). MUSAS compacts into a size 000 capsule (26 mm length × 9.5 mm diameter), the largest Food and Drug Administration (FDA)-approved ingestible form, with elastomeric fabrication options (Fig. 2e). Upon exposure to body temperature, SMA lamellae actively deform to achieve remora-like interlocking (Fig. 2f).

We further optimized MUSAS by dissecting the physico-mechanical roles of its components (Fig. 2g). The benchmark used eight rows of parallel-angled lamellae (III in Fig. 2g). We identified an optimal 2.0× soft lip thickness ratio that maximized adhesion relative to device weight (Fig. 2h). Four biomimicry experiments tested: (1) whether lamella-only or compartment-only designs sufficed; (2) the function of the backbone plate; (3) the role of lip curvature; and (4) elastomer compatibility (Fig. 2i). The results showed that the backbone and smoothed lip edge significantly enhanced adhesion. Both multicompartment suction and lamella interlocking were essential. Notably, MUSAS maintained robust adhesion across elastomer types, with no statistically significant differences.

To probe lamella design, we studied how orientation and row number affect performance. We analysed three representative remora species adapted to different hosts (Fig. 1c): *R. brachyptera* and *R. albescens* (host-specific to billfish and rays, respectively), and *E. neucratoides* (generalist), and defined three lamella configurations: unimodal parallel-angled (para), tilt-dominant with variable angles (tilt) and bimodal (parallel + tilt). The tilt-angled, eight-row design from *R. albescens* achieved the highest adhesion on soft substrates under both normal and shear frictional loads (Fig. 2j). Interestingly, parallel-angled discs resisted shear drag and retained adhesion while sliding post-plateau—a phenomenon we termed shear sliding (Fig. 2k and Supplementary Video 4). These discs slid more than twice their lamella length while maintaining shear frictional adhesion (Extended Data Fig. 2).

To further determine whether a particular angle of lamella orientation contributes to adhesion, we compared the tilt-dominant design, with five lamella configurations featuring specific angle domination and varied row numbers, on soft substrates of different stiffness and roughness. Mechanical performance, micro-computed-tomography imaging and hydrodynamic analysis confirmed the superiority of tilt-dominant designs. Importantly, enhanced adhesion was due to the diversity of lamella orientations, not any specific angle (Extended Data Figs. 3 and 4, and Supplementary Fig. 7). Adhesion decreased on rough versus smooth surfaces, and soft versus stiff substrates (Supplementary Figs. 7 and 8). More compartments improved adhesion robustness, and shear sliding re-emerged in unimodal designs dominated by specific angles (Supplementary Figs. 7 and 8).

These findings explained host-specific lamella adaptations across remora species. *R. albescens* probably evolved variational, tilt-dominant lamella orientation to improve adhesion to the soft mucosa of rays' oral cavities and gill chambers, substantiating the optimal design of MUSAS. Furthermore, remoras such as *R. australis*, *R. brachyptera* and *R. osteochir* show unimodal lamella orientation, possibly enhancing robustness for hitchhiking on high-speed swimmers.

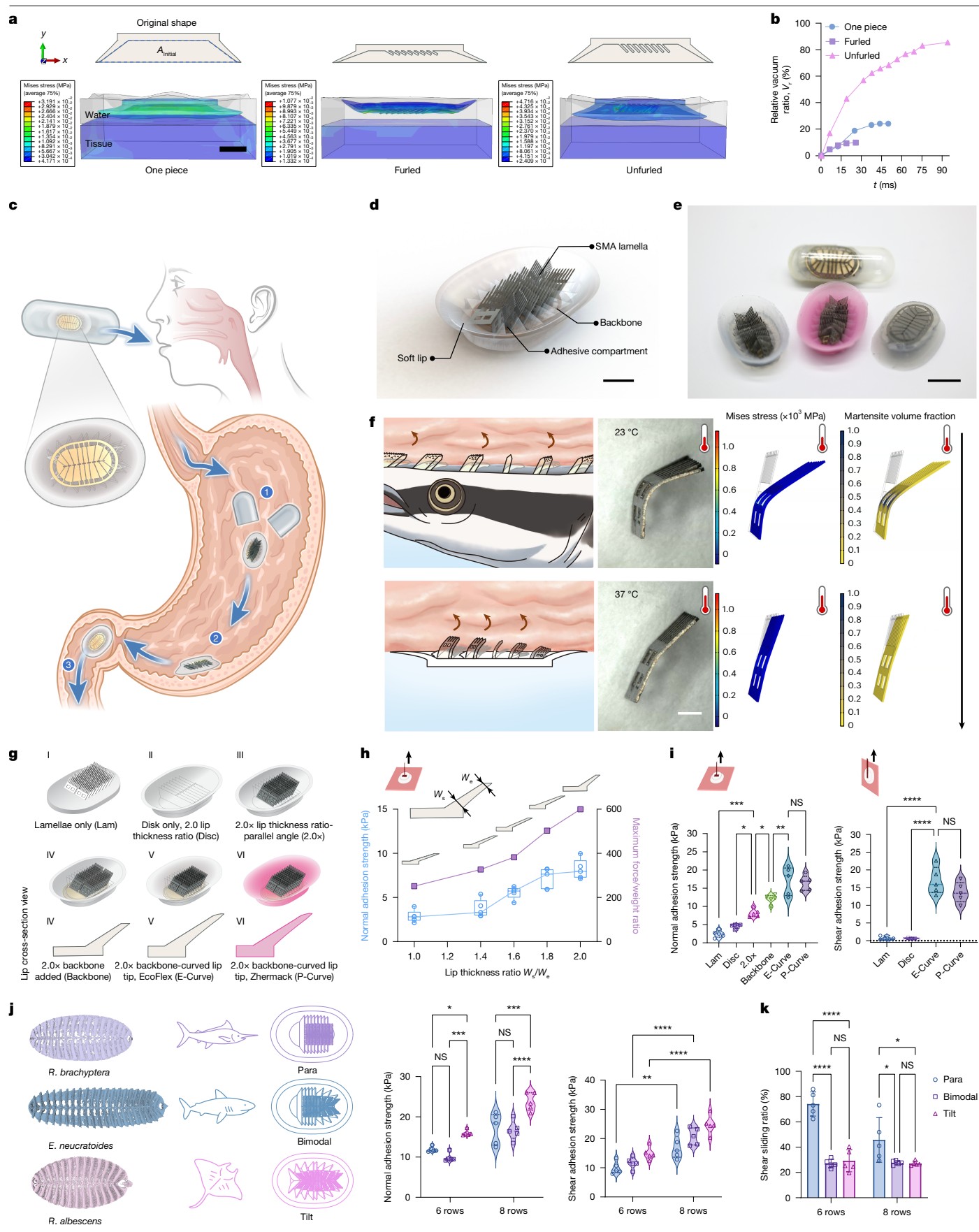

**Fig. 2** | See next page for caption.

**Fig. 2 | Understanding remora's adhesion to soft substrates through biomimicry. a**, Modelling remora adhesive disc mimicries contacting stomach tissue: a benchmark one-piece disc (left), a furled multicompartmental disc (middle) and an unfurled multicompartmental disc (right). Scale bar, 5 mm. **b**, Relative vacuum $V_r$ (expelled water volume relative to theoretical maximum volume $A_{Initial}$) achieved by disc mimicries post-tissue contact. **c**, Illustration of MUSAS delivery: (1) release, (2) adhesion and (3) safe passage. **d**, Schematic of the optimal MUSAS, inspired by *R. albescens*. Scale bar, 5 mm. **e**, MUSAS compacts into a size 000 capsule, with elastomeric fabrication flexibility (transparent, Ecoflex 0030; pink, Zhermack Elite Double 8). Scale bar, 1 cm. **f**, Active mechanical interlocking of MUSAS via SMA lamellae. Left: active lamella deformation (bottom) mimicking remora's interlocking (top). Middle: temperature-induced SMA lamella erection. Right: modelling stress and martensite volume fraction during lamella erection. Scale bar, 1 mm. **g**, Designs (I–VI) tested to dissect the physico-mechanical roles of disc components, including the lamellae, multicompartment adhesive disk, backbone, disk lip curvature, and disk

elastomer compatibility. **h**, Adhesion performance across different lip thicknesses, where $W_s$ and $W_e$ denote the lip thickness at the start and end of the disk lip, respectively. **i**, Contribution of disc components to adhesion. **j**,**k**, Adhesion performance (**j**) and shear sliding measuring retained adhesion post-plateau (**k**; also see Extended Data Fig. 2) of designs with various lamella orientations and row numbers, mimicking three representative remora species. It is noted that the parallel-eight-rows design (**j**) is the benchmark E-Curve tested in **i**. $n = 5$ devices per design were tested in **h**–**k**. For **h**, boxes represent the median and the Q1 and Q3 quartiles, and whiskers represent the minimum and maximum. For **i** and **j**, the dashed and dotted lines represent the median and the Q1 and Q3 quartiles, respectively. For **k**, the error bars represent mean ± s.d. One-way (**i**) and two-way (**j**,**k**) analysis of variance with Šídák multiple comparison test were used to compare different designs. Statistical significance is indicated as follows: NS, non-significant, $*P \leq 0.05$, $**P \leq 0.01$, $***P \leq 0.001$, $****P \leq 0.0001$.

Their ability to endure shear drag by sliding across the host's surface without fully losing adhesion may help them relocate to optimal adhesion zones under sudden swimming speed boosts or volatile movements of the hosts. This echoes the skimming and sliding behaviour observed in *R. australis* when attached to cetaceans during suboptimal hydrodynamic conditions[30]. An extended discussion is available in Supplementary Text 3.

## Characterization of MUSAS

We conducted comprehensive in vitro and ex vivo characterization of MUSAS to assess its adhesion performance for translational applications (unless otherwise noted, MUSAS evaluated throughout the rest of the study refers to the optimal design; Fig. 2d). Confocal microscopy revealed water redistribution during the multicompartmental adhesion process. Before adhesion, water accumulated around the edges owing to surface tension (Fig. 3a). Upon adhesion, water redistributed towards the lamellae and soft junctions between the lamellae and lip, creating negatively pressured hollow spaces at the centre of each compartment, enabling independent suction (Fig. 3b).

Subsequently, we evaluated the underwater adhesion of MUSAS on dynamic, morphable soft substrates varying in surface stiffness, roughness and intactness (Fig. 3c,d). Test specimens included Bis-Tris polyacrylamide gel (Thermo Fisher Scientific), porous double-network tough hydrogel, pig stomach tissue, STP, nitrile gloves (MedPride) and styrene–ethylene–butylene–styrene (SEBS), a common soft electronics substrate (Fig. 3e–g and Supplementary Fig. 9). Consistent with previous findings, adhesion was generally stronger on smooth, stiff surfaces (Fig. 3c,d). Nevertheless, MUSAS maintained robust adhesion across all substrates owing to multicompartmental adhesion balancing and lamellae-enabled mechanical interlocking. Notably, MUSAS achieved an adhesion-force-to-weight ratio of up to 1,391 (Fig. 3d) and adhered effectively to soft, non-intact surfaces, including those with holes, such as the porous tough hydrogel (Fig. 3e–g).

We further tested MUSAS ex vivo on swine stomach tissue under wet and underwater conditions, considering environmental pH, previous research and commercial bioadhesives[3,14], including *N*-hydroxysuccinimide (NHS)–1-ethyl-3-(3-dimethylaminopropyl) carbodiimide (EDC)-based covalent adhesives, Carbopol-based hydrogen-bonding adhesives, and the biomedical skin adhesive Silbione (Elkem, https://www.factor2.com/v/vspfiles/sds%20tds%20 2023/tds%202023/RTV-4717%20ELKEM.pdf). MUSAS outperformed all comparators by a factor of 2 to 300 (Fig. 3h). It is noted that unlike polymeric adhesives that require prolonged, pressurized pre-adhesion and neutral to alkaline pH for strength bonding[14], MUSAS achieved immediate mechanical adhesion. For fair comparison, pre-adhesion times for NHS–EDC and Carbopol hydrogels were limited to 3 min, with all tests conducted on freshly collected, untrimmed tissue (Supplementary

Text 2). Importantly, MUSAS showed strong insensitivity to pH and moisture owing to its intrinsic mechanical adhesion mechanism. The ex vivo performance of MUSAS across various swine organs is shown in Fig. 3i. Notably, MUSAS achieved a shear adhesion force of up to 3.5 N on gastric mucosa (Supplementary Fig. 11)—nearly 4 times the 0.9-N threshold needed to retain devices on GI mucosa[31].

## MUSAS resisting dynamic interference

We validated the adhesion performance and biostability of MUSAS in vivo in dynamic, unstructured and often unpredictable environments using the swine model, selected for its resemblance to adult human anatomy for translational relevance, as well as the fish model[32]. MUSAS showed consistent performance across 58 pigs at various GI locations (buccal cavity, oesophagus, stomach and small intestine) and 8 fish, with GI retention evaluated in over 9 pigs and body surface retention in over 6 fish under survival conditions—defined as uninterrupted normal feeding, resting and behaviour (Fig. 4a–c and Extended Data Fig. 5). Notably, optimal MUSAS showed robust resistance to dynamic interference (Extended Data Fig. 5a–c and Supplementary Video 6) and remained in place for an average of over 9 days, lasting up to 3.5 weeks in the swine stomach before passing safely through the GI tract (Fig. 4b,c). We also demonstrated additional merits of MUSAS including programmable retention by tuning lamella biodegradability (Fig. 4b,c and Extended Data Fig. 5d,e), and targeted delivery (Extended Data Fig. 6). Importantly, self-adhesion during oral delivery was facilitated by GI contraction and peristalsis, enabling motor-free, ingestible delivery driven solely by endogenous forces, with preserved active pharmaceutical ingredient loading for drug release (Fig. 4d,e and Supplementary Video 7). Furthermore, histological analysis revealed lamella spinule penetration depths of 300–800 μm and hole diameters of about 100 μm (Fig. 4f,g). These findings establish MUSAS as a non-invasive microneedle platform capable of breaching mucosal barriers—overcoming a key challenge in delivering large-molecule drugs to the GI tract[33].

## MUSAS for biosensing and drug delivery

The exceptional adaptability of the MUSAS to versatile soft substrates under extreme environmental conditions enables broad application possibilities. Here we present four distinct application scenarios in vivo that address major challenges in biological exploration, bioelectronics and biomedical treatment. Independent in vitro and ex vivo characterizations of the applications and extended discussions are provided in Supplementary Information.

The first application scenario addresses the challenge of monitoring environmental conditions to understand ecological needs and social behaviours of aquatic species[34]. Here we demonstrated applying

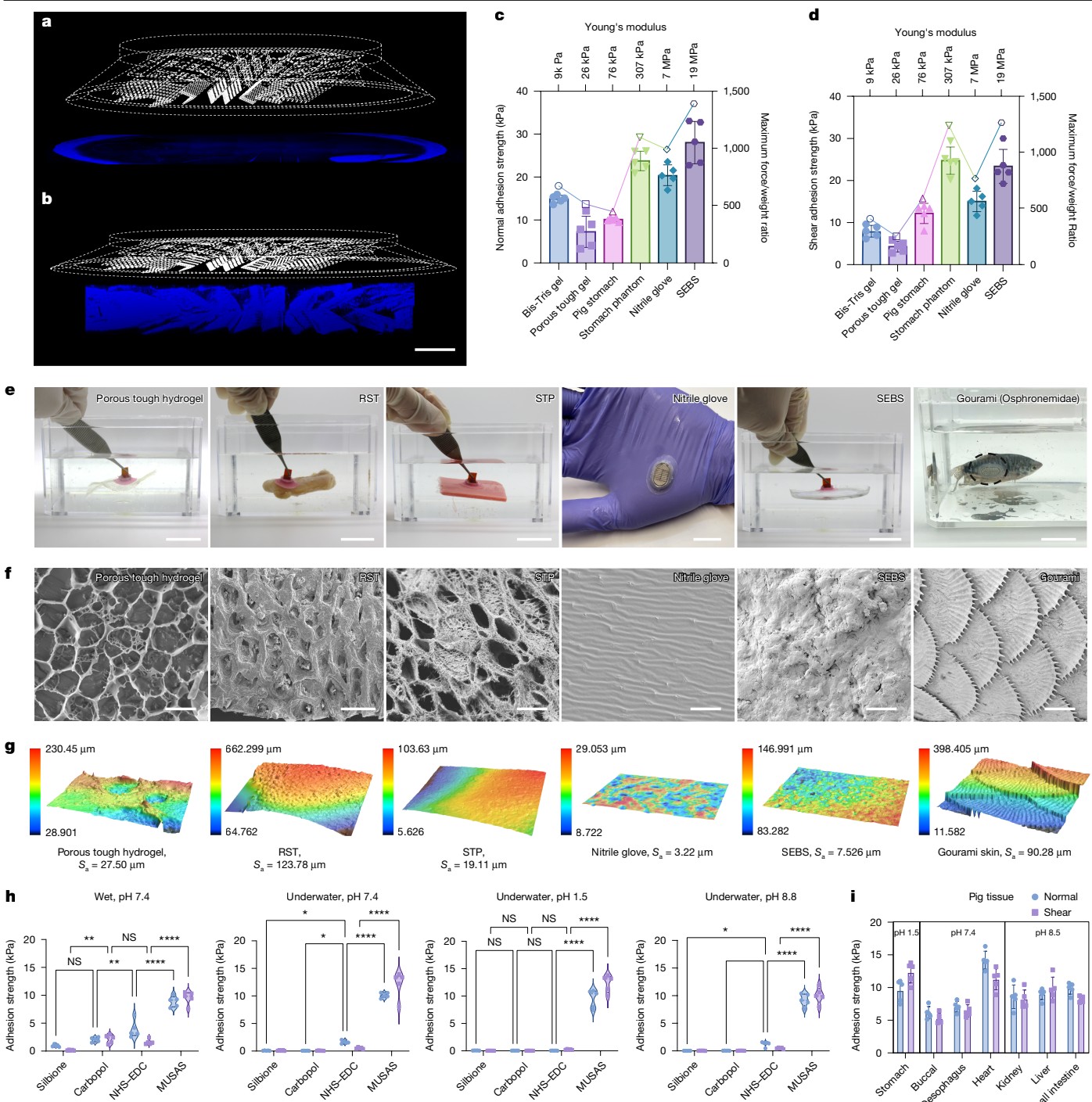

**Fig. 3 | In vitro and ex vivo characterization of MUSAS. a,b,** Confocal microscopy images showing the blue-dyed (cyanine) fluorescent water distribution of MUSAS before (**a**) and after (**b**) adhesion, within the upper limits of planar and *Z*-depth detection of the confocal microscopy. Scale bar, 2.5 mm; *n* = 2 independent measurements. **c,d,** Adhesion performance of MUSAS on various soft substrates with different surface stiffness, roughness and intactness in normal (**c**) and shear (**d**) directions, where bars represent adhesion strength (*n* = 5 devices), and lines indicate the maximum adhesion force-to-weight ratio. The error bars represent mean ± s.d. **e–g,** Underwater adhesion of MUSAS to various soft substrates (**e**; scale bars, nitrile glove 1 cm, all other substrates 2 cm; also see Supplementary Video 5), with associated scanning electron microscopy images of the surface intactness (**f**; scale bars,

hydrogel and STP 1 μm, RST 100 μm, nitrile glove 3 μm, SEBS 10 μm, gourami 200 μm; *n* = 3 independent samples) and surface roughness (**g**; *n* = 2 samples; colour bars represent surface height) of tested substrates. **h,** Ex vivo characterization of wet and underwater adhesion of MUSAS on swine stomach tissue under varying pH, compared with prevalent solutions. *n* = 5 per adhesives. The dashed and dotted lines represent the median and the Q1 and Q3 quartiles, respectively. **i,** Ex vivo adhesion performance of MUSAS on various swine organs. *n* = 5 devices. The error bars represent mean ± s.d. Two-way analysis of variance with Tukey multiple comparison test was used to compare different adhesives (**h**). Statistical significance is indicated as follows: *$P \le 0.05$, **$P \le 0.01$, ***$P \le 0.001$, ****$P \le 0.0001$. Details of the mechanical test can be found in Methods and Supplementary Information.

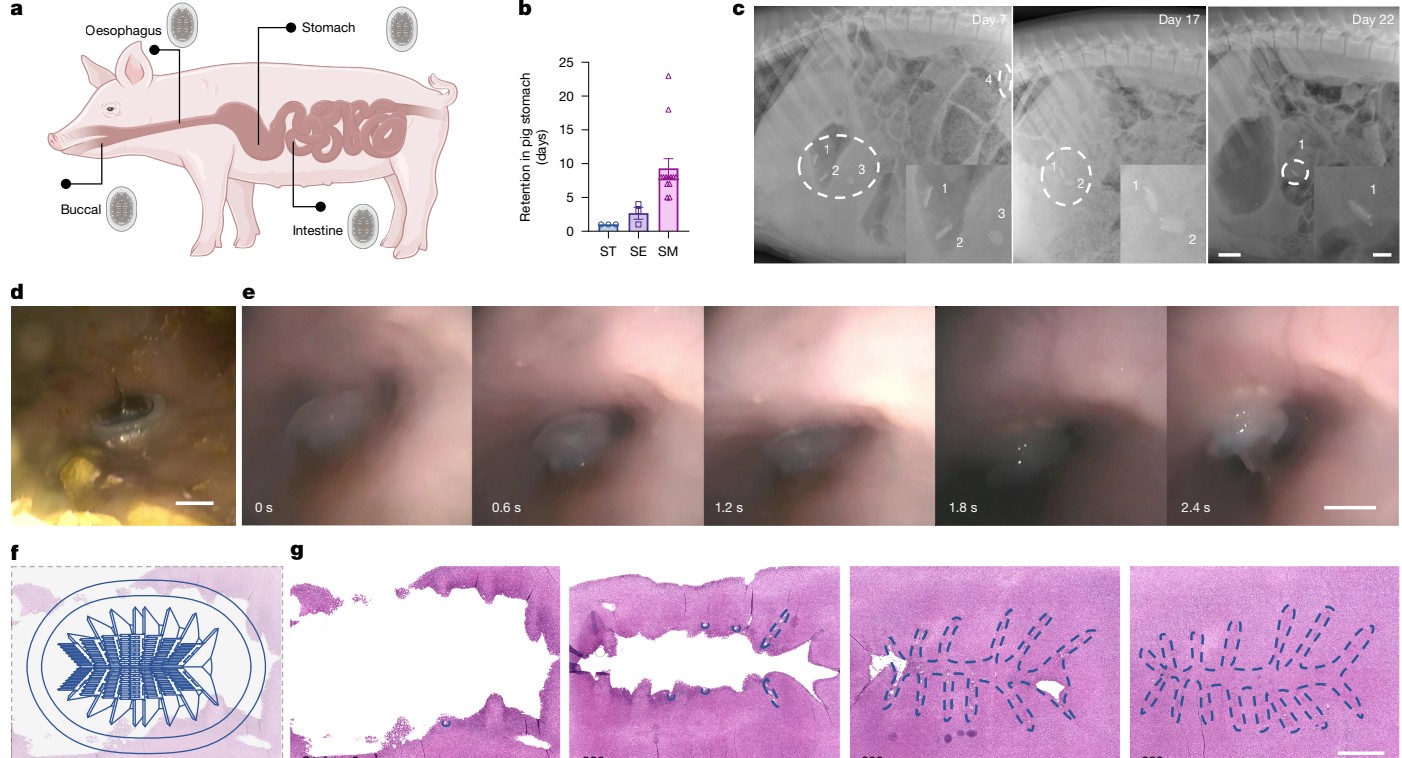

**Fig. 4 | In vivo characterization and resistance to dynamic interference of MUSAS. a**, In vivo adhesion and retention achieved by MUSAS in different parts of the GI tract in a swine model. **b**, Controlled retention and detachment of optimal MUSAS in the swine stomach through programmed mechanical interlocking with various lamella materials. $n = 3$ devices per stainless steel (ST) and superelastic nitinol (SE) lamellae-based MUSAS; $n = 13$ devices for shape-memory nitinol (SM) lamellae-based MUSAS. The error bars represent mean ± s.d. See Extended Data Fig. 5d,e for further information. **c**, X-rays of long-term retention of four MUSAS delivered in the stomach before safe passage in the GI tract. Scale bars, 5 cm (left), 1 cm (right). Numbers 1–4 denote four individual devices. **d**, Endoscopic picture of MUSAS residing in the stomach. Scale bar, 1 cm. **e**, Oral-delivered MUSAS leveraging oesophagus contraction to achieve self-adhesion. Scale bar, 1 cm. Also see Supplementary Video 7. **f,g**, Histology of lamella spinule penetration depth of MUSAS on swine stomach tissue, confirming MUSAS as a non-invasive microneedle platform capable of breaching mucosal barriers. **f**, The interaction area between MUSAS and the stomach tissue. **g**, The penetration sites from top to bottom, with dashed lines encircling the penetration holes of lamellae interaction. Scale bar, 5 mm. $n = 2$ Yorkshire pigs evaluated.

MUSAS for kinetic temperature sensing on a tilapia (*Oreochromis niloticus*) model. In vivo results showed that MUSAS can adhere to various tilapia body locations, including the operculum, head and body (Fig. 5a(ii)), maintaining attachment for up to 110 h without affecting swimming or feeding behaviour (Fig. 5a(iii)). We further developed a battery-free, ultraminiaturized wireless radiofrequency identification temperature sensor (1.3 × 6 × 12 mm) with a 1-m working distance (Fig. 5a(i)). In vivo testing confirmed its ability to measure real-time temperature on a swimming tilapia simulating a natural environment (Fig. 5a(iv) and Supplementary Video 9). These results showed robust adhesion of MUSAS and kinetic biosensing capabilities in unpredictable, unstructured environments (see Extended Data Fig. 7 and Supplementary Text 8 for extended discussion).

The second application scenario focuses on digital health monitoring, specifically for kinetic biosensing of GERD. GERD affects 10–20% of adults in Western countries and nearly 5% in Asia, contributing to over 5.6 million annual clinical visits[35]. The Bravo capsule, the most widely adopted sensor for detecting reflux, uses a pin-like needle to invasively clip onto the mucosa, which can be uncomfortable for patients[36]. Leveraging MUSAS' self-adhesion (Fig. 4e), we developed a flexible impedance sensor that integrates with MUSAS to monitor pH variations in the oesophagus (Fig. 5b). We validated the sensor in vivo using a gastric fluid reflux swine model by periodically spraying fluid into the oesophagus via endoscopy. The results showed that the MUSAS-based impedance sensor precisely detected gastric reflux, offering a non-invasive solution for GERD monitoring (Fig. 5b(ii); see Extended Data Fig. 8 and Supplementary Text 9 for extended discussion).

The third application scenario explores sustained oral delivery of HIV/AIDS PrEP, enabled by the reliable retention performance of MUSAS (Fig. 4c). As of 2022, the World Health Organization reported over 40.4 million people living with HIV worldwide, underscoring the urgent need for effective prevention strategies (https://www.who.int/news-room/fact-sheets/detail/hiv-aids). Cabotegravir (CAB), an FDA-approved antiviral for PrEP, is limited in its long-acting injectable form by burdensome initial tolerance establishment and frequent clinic visits, which can reduce patient compliance[37]. Importantly, we wanted to understand whether long-term delivery of CAB was feasible through oral application. We introduced a MUSAS-enabled, slow-release ingestible formulation that leverages polycaprolactone and Ecoflex elastomer as sustained-release matrices for prolonged retention of CAB in the GI tract (Fig. 5c). In vitro release kinetics studies showed that both polycaprolactone and Ecoflex 0030 are well suited as sustained-release matrices (Fig. 5c(ii)). In vivo studies in a swine model further confirmed that MUSAS-enabled long-acting CAB pills provided extended drug release over a 7-day study window (Fig. 5c(iii); see Supplementary Text 10 for extended discussion).

The fourth application scenario features the long-standing challenge of delivering mRNA therapeutics to the GI tract. GI delivery of mRNA therapeutics could enable non-invasive vaccination at mucosal surfaces, inducing localized immune responses distinct from systemic

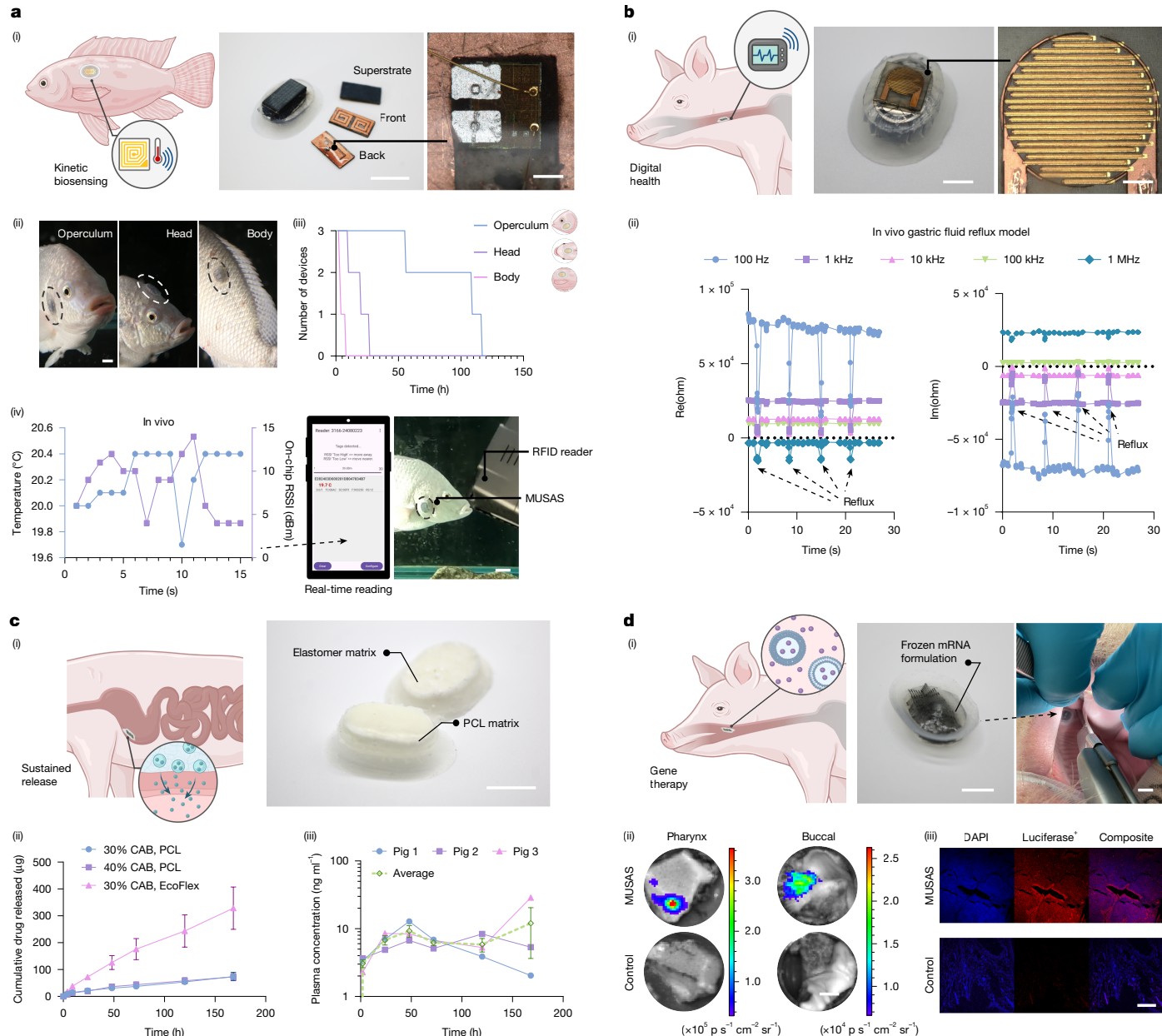

**Fig. 5 | MUSAS for kinetic biosensing and drug delivery. a**, (i) Kinetic biosensing via ultraminiaturized, battery-free, wireless MUSAS radiofrequency identification (RFID) temperature sensor on a tilapia model. Scale bars, 1.5 cm (left), 200 μm (right). (ii) Demonstration of MUSAS adhesion to various location of tilapias. Scale bar, 1 cm. Also see Supplementary Video 8. (iii) Retention time of MUSAS on swimming tilapias. *n* = 3 tilapias per body location. (iv) In vivo kinetic temperature sensing of MUSAS on a swimming tilapia. Scale bar, 2 cm. Also see Supplementary Video 9. RSSI, received signal strength indicator. **b**, (i) Digital health monitoring of GERD with MUSAS-based impedance sensor. Scale bars, 5 mm (left), 1 mm (right). (ii) MUSAS detecting gastric reflux in a swine gastric fluid reflux model. **c**, (i) MUSAS-enabled sustained drug delivery of HIV PrEP CAB. Scale bar, 1 cm. (ii) In vitro release kinetics of CAB via different sustained-release matrices. *n* = 3 samples. The error bars represent mean ± s.d. (iii) Plasma pharmacokinetics of sustained CAB release through a 7-day study on a swine model. *n* = 3 Yorkshire pigs per treatment. The error bars represent mean ± s.e.m. **d**, (i) Luciferase mRNA delivered via MUSAS. Scale bar, 5 mm (left), 1 cm (right). (ii) In vivo imaging system detecting luminescent signal of luciferin conjugating proteins expressed through functionally transfected luciferase mRNA. Colour bars denote radiance measurements. Scale bar, 1 cm. (iii) Fluorescent immunohistochemistry to evaluate transfection of luciferase mRNA in the buccal and pharyngeal regions of a swine model. Scale bar, 400 μm. *n* = 3 Yorkshire pigs per treatment. Also see Extended Data Fig. 9.

vaccination[38]. It further allows direct access to GI tissues for gene therapies in treating GI disorders such as Crohn's disease, which systemic delivery cannot effectively treat[39]. Recent studies show mechanical GI delivery systems enhance the bioavailability of large molecules and nanoparticles by physically penetrating the mucosal barrier while reducing systemic immunogenicity[40,41]. Here we utilized MUSAS as a microneedle adhesive platform to deliver frozen lipid nanoparticle mRNA formulations in a swine model (Fig. 5d). Its unique adhesion

and microneedle features enabled mucosal penetration and effective delivery of firefly-luciferase-mRNA lipid nanoparticles to buccal and pharyngeal regions, achieving a substantial drug-loading capacity (215 μl) and efficient tissue absorption (Fig. 5d). Robust protein expression was double-confirmed in three pigs through bioluminescence imaging (Fig. 5d(ii)) and immunofluorescence histology (Fig. 5d(iii)), demonstrating significant advancement in overcoming biological barriers for localized GI mRNA delivery

(see Extended Data Fig. 9 and Supplementary Text 10 for extended discussion).

## Discussion

We explored previously unexamined adhesion mechanisms of remoras on soft substrates, as well as the anatomical evolution of their adhesive disc across all species. These biological insights directly informed the design and development of MUSAS, a biomimetic adhesion system that captures the key functional principles of remora attachment. Building on these biological insights, we rigorously validated MUSAS through comprehensive in vitro, ex vivo and in vivo studies, pushing the boundaries of technical feasibility and expanding research capabilities in this field. Our approach is strengthened by multimodal cross-validation across diverse application scenarios. Furthermore, we developed a range of MUSAS-based applications, each addressing independent, long-standing challenges and opening frontiers in bioadhesion, biomedical treatments and biological exploration. Extensive testing in over 58 pigs and 8 fish revealed consistent trends in adhesion performance, application validity and safety, highlighting the robustness and potential generalizability of the platform to human use (see Extended Data Fig. 10 and Supplementary Text 11 for extended discussion).

In contrast to the unique advantages of polymeric adhesives in applications such as hemostasis and sealing tissue defects, MUSAS remains limited in these scenarios. Instead, it is mostly suitable for in situ adhesion and long-term retention on dynamic, morphable soft substrates that undergo rapid turnover and regeneration in extreme pH and moisture environments. Future improvements and studies of MUSAS-enabled technologies could be pursued along several directions. First, advancing computational theories in multicompartmental adhesion could help model and guide the design of mechanical adhesives. Second, although the microscale tissue penetration of MUSAS limits its applicability in leakage-sensitive organs such as the vasculature and lungs, this same feature, combined with retention, makes MUSAS ideal for breaching mucosal barriers. This opens avenues for sustained delivery of small-molecule drugs, submucosal delivery of biologics for mRNA-based mucosal vaccines, and gene/protein replacement therapies in the GI tract.

In addition, the silicone rubber family that constitutes MUSAS provides ideal substrates for soft electronics[42], enabling future integration with components such as acoustic, temperature and pressure sensors to expand its operational range and functionality. Furthermore, MUSAS' adhesion and retention performance can be enhanced. For example, mucus secreted by remoras and their hosts may increase fluid viscosity at the adhesion interface, improving the watertight seal and facilitating repositioning during hitchhiking[43]. Incorporating a mucus-like polymeric coating could replicate this effect. Future efforts may also explore enhanced antifouling, waterproof silicone rubbers and finer setal structures in the 10–1,000-nm range[4,13] to improve the durability and performance of MUSAS (see Supplementary Text 12 for extended discussion). In conclusion, we envision MUSAS catalysing further exploration into mechanical approaches for developing cost-effective, manufacturing-friendly solutions for underwater adhesion on soft substrates across diverse industries.

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

## Methods

### Micro-computed tomography

We used a SkyScan 1173 micro-computed tomography unit (Bruker) with SkyScan 1173 application software to scan individual heads of various remora species. The scanning parameters included a voltage range of 51–130 kV, an amperage range of 40–136 µA, an exposure time range of 337–730 ms and an image rotation range of 0.06–0.07°. Slice reconstruction of the osteological structure of the remora suction disc was performed using NRecon (Micro Photonics) and rendered in Mimics 15.0 (Materialise). The lamella angle of the remoras, viewed dorsally, was characterized using Fusion 360 (Autodesk).

### Fabrication of MUSAS

MUSAS are flexible to be fabricated using a variety of silicone rubbers and biodegradable lamella materials. A detailed fabrication procedure is provided in Supplementary Fig. 3. Unless otherwise specified, all testing of MUSAS was conducted using MUSAS fabricated with Ecoflex 0030 elastomer and shape memory nitinol lamellae.

### Universal mechanical testing

Universal tensile testing was conducted to measure the stiffness and adhesion performance of the specimens and materials of interest, including tissue and material samples, euthanized remora, devices and adhesives. A 5944 universal testing system (Instron) with Bluehill V3.11 was used for the tensile test. For both stiffness and adhesion study, the tensile extension rate was set to 30.0 mm min$^{-1}$, with a measurement interval of 20–100 ms for time, $1 \times 10^{-5}$ N measurement accuracy for load, and $1 \times 10^{-5}$ mm measurement accuracy for displacement. All test specimens, including devices and adhesive materials used in the adhesion test, were bonded to the top of a double-stacked 0.5-mm-thick polyimide Kapton strip (McMaster-Carr) using ultraviolet-cured 5055 silicone adhesive (Loctite). The Kapton strips were secured with an Instron tensile grip (Supplementary Figs. 2 and 4). A preload of 0.5 N was applied to all devices and adhesive materials before test. To ensure a fair comparison, the pre-adhesion pressurization for other adhesive materials was for 3 min, whereas for MUSAS there was no prolonged pressurization. Soft substrates used in the mechanical adhesion testing were freshly prepared inert materials or collected tissue (<1 h post-euthanasia) without surface washout, tissue trimming or liquid removal. Soft substrates were only partially secured, with the four corners of the tissue squares glued to the holder to allow for natural sliding and dynamic morphing. All devices and adhesive materials used in the adhesion test had the same adhesion surface area of roughly 250 mm$^2$, which equals the adhesion surface area of MUSAS. The adhesion pressure was calculated by dividing the force by the adhesion surface area. Extended discussion of the experiment set-up and measurement details can be found in Supplementary Text 2, Extended Data Fig. 2, and Supplementary Figs. 2 and 4–11.

### Numerical simulation

**Solid–fluid interactions between water, tissue and MUSAS.** Finite element analysis was conducted to characterize the hydrostatic differentiation of the adhesive disc of remoras. Commercial software Abaqus 2021 (SIMULIA) was used for the study. The remora disc phantom devices were assumed to be composed of Ecoflex 0030, with a density of 1.07 g cm$^{-3}$, a Young's modulus of 125 kPa and a Poisson's ratio of 0.49 (ref. 48). The physical parameters of stomach tissue include a density of 1.088 g cm$^{-3}$, a Young's modulus of 700 kPa and a Poisson's ratio of 0.49 (ref. 10). We used coupled Eulerian–Lagrangian techniques to model the solid–fluid interactions between tissue, device and water. Water was treated as a Newtonian laminar flow, with a density of 0.997 g cm$^{-3}$ and a dynamic viscosity of $8.90 \times 10^{-4}$ Pa s. The simulation was configured so that the mimicry remora suction cups descended at a constant speed of 0.3 mm s$^{-1}$ until they touched the stomach tissue submerged

in water. Solid–solid interactions were modelled as hard contact for normal behaviour, with tangential behaviour modelled using a penalty method with a friction coefficient of 0.02. Details of calculation of relative vacuum ratio $V_r$ are specified in Supplementary Information.

**Self-actuation of nitinol lamella of MUSAS.** The self-actuation of nitinol lamellae in MUSAS in response to temperature stimuli was simulated using the commercial software COMSOL Multiphysics 6.2 (COMSOL). The Lagoudas phenomenological inelastic constitutive model for SMAs (nitinol), including the relevant material properties, was implemented to simulate the phase transformation of the SMAs[49].

### Synthesis of materials for in vitro adhesion characterization

**Polyacrylamide–alginate tough hydrogel.** The tough hydrogel was composed of alginate and polyacrylamide (pAAm) double networks (pAAm–alginate) crosslinked by numerous dimethacrylate monomers. The synthesis was based on a previously reported protocol[14]. Hydrogel fabrication was achieved via one-step aqueous free-radical polymerization. In brief, in a 50-ml tube, 30 ml phosphate buffer (100 mM, pH 7), 3.6 g acrylamide, 600 mg sodium alginate (medium viscosity), 1.3 mg *N*,*N*′-methylenebisacrylamide and 10 mg ammonium persulfate were added and vortexed to form solution A (pAAm–alginate).

Calcium sulfate was added to deionized water and stirred to form a homogeneous suspension. Then, 5 ml pre-gel solution A was loaded into a 5-ml syringe (diameter 12 mm), and 120 mg calcium sulfate and 29.4 mg *N*,*N*,*N*′,*N*′-tetramethylethylenediamine were loaded into another 5-ml syringe. The two syringes were connected with a syringe connector and mixed over ten times. Afterwards, the gel solution was poured into a glass mould covered with a 3-mm-thick glass plate. After 12 h, the hydrogel was ready to be removed from the mould. In particular, for tests leveraging tough hydrogel as soft substrates to evaluate adhesion performance of MUSAS (Fig. 3c–g), air bubbles were manually introduced during the preparation procedure to create a porous and rough substrate surface.

**NHS–EDC bridging polymers for tough hydrogel.** The bridging polymers, which include chitosan, polyallylamine, gelatin and polyethyleneimine, were prepared following a previously reported protocol[14]. Sulfated NHS and EDC were used as coupling reagents. Right before the adhesion of the tough hydrogel to tissue surfaces, the bridging polymers and coupling reagents were mixed to achieve a concentration of 12 mg ml$^{-1}$ of both NHS and EDC in the bridging polymer solutions. A 250 µl mixed solution was then smeared onto the surface of the tough hydrogel, followed by immediate compression of the tough hydrogel to the tissue surfaces for 3 min before the adhesion test.

**Carbopol tough hydrogel.** To prepare Carbopol tough hydrogel, 100 mg Carbopol 971P was dissolved in the pAAm–alginate solution described in the pAAm–alginate tough hydrogel protocol. The rest of the preparation procedures were identical to those used for the pAAm–alginate tough hydrogel. A 3-min compression of the Carbopol tough hydrogel to the tissue surfaces was applied before the adhesion test.

**SEBS thermoplastic elastomer.** The SEBS substrate was prepared by mixing 10 ml toluene with 4 g SEBS (Kraton G1645). After dissolution, the ink was homogenized using a speedmixer (FlackTek 330) at 2,000 rpm for 5 min. The ink was then drop-cast onto stainless steel Petri dishes to achieve a film thickness of 3 mm. The substrate was dried in a fume hood for 2 h and cured at 60 °C for 1 h. After curing, the stretchable SEBS substrate was peeled off from the petri dish.

### In vivo testing

All swine studies were approved by and performed in accordance with the Committee on Animal Care at the Massachusetts Institute of Technology. All fish studies were approved by and performed in

accordance with the Institutional Animal Care and Use Committee of Boston College. Additional details and extended discussion can be found in Supplementary Text 6.

**Fabrication of MUSAS fish tag with a temperature sensor.** A Proto-Laser R4 laser cutter (LPKF) was used to pattern the top and bottom 35-μm-thick copper claddings of an RT/duroid 6010.2LM laminate (Rogers), which features a 0.635-mm-thick ceramic–PTFE composite dielectric core. The laser was then employed to ablate a via hole through the substrate, establishing an electrical connection to the top copper layer using a soldered 32 AWG feedthrough wire. The antenna geometry, measuring 12 mm × 6 mm, was laser-cut from the patterned RT/duroid laminate. A Magnus S3 tag chip (Axzon) was mounted onto the ground plane with a non-conductive ultraviolet-cured epoxy adhesive. Chip-to-antenna interconnections were made via thermosonic gold ball bonding, reinforced with silver conductive paste and cured at 65 °C for 40 min using a C174740 Mech-El MEI Marpet 1204B wire bonder. A superstrate with hatched top and bottom copper claddings on an RT/duroid 6010.2LM core was then fabricated and affixed to the top surface of the antenna with an adhesive layer. Finally, the assembled antenna was packaged onto the remora device by underfilling and encapsulating the bottom and sides with 5055 UV Curing Silicone Adhesive (Henkel Loctite) epoxy resin. See Supplementary Information for details of in vitro and in vivo tests on a tilapia model.

**Fabrication and evaluation of MUSAS impedance biosensor for detecting gastroesophageal reflux.** The impedance sensor was laser-cut on single-sided flexible copper laminate (Pulsar Professional fx FR4 5 mil ½-oz copper). Each of the 13 traces were 75 μm wide and 75 μm apart from each other. Alignment of the cut was performed with camera on a R4 laser (LPKF). Parylene coating was then performed on a PDS 2010 Labcoater (Special Coating System), to prevent unnecessary shortcut of the impedance sensor. A Kapton tape mask (0.03-mm thickness, McMaster-Carr) was cut with the R4 laser to cover the impedance sensor traces during the parylene-coating procedure. After completion of the parylene coating, the Kapton tape mask was peeled off. To improve the electric conductivity and sensitivity of the traces of the impedance sensor, electron beam evaporation was performed to deposit gold on the copper traces. The deposition procedure included coating a 10-nm adhesion layer of titanium and then 200-nm gold and was performed on EE-4 E-beam evaporator (Denton).

For impedance measurement of tissue, we used a ISX-3 Impedance Analyzer (Sciospec) equipped with Sciospec software v2.0.8 for four-point impedance measurements. The measurements spanned frequencies between 100 Hz and 1 MHz. For the in vivo study, MUSAS impedance biosensors were placed via an over tube into the oesophagus of anaesthetized female Yorkshire pigs (70–95 kg) and self-adhered through the contraction of the oesophagus. A gastroesophageal reflux model was created by using an endoscope to periodically spray gastric fluid, obtained from the pig itself, into the oesophagus.

**Fabrication of gastric resident dosage forms for sustained release of CAB.** The polycaprolactone (PCL) matrices containing CAB were prepared with melt mixing. The drug stability of the matrices was confirmed in previous research[50]. Specifically, PCL and CAB were weighed in a 10-ml glass vial. The vial and a negative mould for the PCL patch were then heated on a heat plate (Thermo Scientific) to 75 °C. The matrices were melted and well mixed before transferring to the mould. After cooling to room temperature, the PCL–CAB patches were adhered onto fabricated MUSAS with ultraviolet-cured epoxy (Henkel Loctite).

The Ecoflex matrices containing CAB were directly prepared by mixing CAB with Ecoflex 0030 for moulding the suction cup of MUSAS.

**In vitro and in vivo evaluation of pharmacokinetics of CAB.** In vitro evaluation of the drug release of CAB was conducted in a release medium of simulated gastric fluid containing 5% w/v Tween 20 surfactant (Thermo Scientific). The drug-loaded MUSAS were placed in 10 ml of the release medium in a 37 °C incubator (New Brunswick Innova 44/44R) shaking at 250 rpm. At 2 h, 6 h, 1 day, 2 days, 3 days, 5 days and 7 days, 1 ml of the medium was sampled and stored at −20 °C until high-performance liquid chromatography (HPLC) analysis, as described later. During each sampling, the remaining release medium was replaced with fresh medium.

In vivo pharmacokinetics were performed in female Yorkshire pigs (55–95 kg) in an unblinded fashion. MUSAS loaded with 40% CAB in the PCL matrices were either dropped through an over tube or endoscopically placed in the stomach of anaesthetized Yorkshire pigs, which were fitted with ear catheters. The pigs were fed and monitored daily in the morning and afternoon with a laboratory mini-pig grower diet, 5081, along with midday snacks of fruits and vegetables. At 2 h, 6 h, 1 day, 2 days, 3 days, 5 days and 7 days, 5 ml of blood was sampled via the ear catheter. The blood samples were centrifuged for serum separation at 4,000 rpm (Eppendorf 5810r) for 10 min and then stored at −80 °C until bioanalysis, as described later.

**Synthesis and in vitro characterization of mRNA nanoparticles.** Messenger-RNA-loaded lipid nanoparticles (LNPs) were made using a typical four-component lipid mixture. Specifically, ethanol-based solutions of SM102, 1,2-distearoyl-*sn*-glycero-3-phosphocholine, cholesterol and 1,2-dimyristoyl-*sn*-glycero-3-phosphoethanolamine-*N*-[(methoxy(polyethylene glycol)−2000) (ammonium salt) were mixed to achieve a molar ratio of 50:10:38.5:1.5. Firefly luciferase mRNA (Trilink) was dissolved in 10 mM citrate buffer, pH 3. The lipid solution was mixed with mRNA solution at a volume ratio 1:3 to achieve a SM102/mRNA weight ratio of 12.86. The LNPs were placed on ice for 10 min to complete the complexation. For measuring the activity of fresh LNPs, the LNPs were diluted in appropriate media and added to the cells. For measuring the activity of freeze-thawed LNPs, the LNPs were diluted with 200 mg ml$^{-1}$ sucrose in 10 mM citrate buffer at a volume ratio of 1:1 and incubated at 4 C for 1 h. The LNPs were then frozen at −20 °C for 1 h. The LNPs were then thawed, diluted with complete media and added to the cells.

In vitro studies were conducted with primary human oral epithelial cells (Celprogen). Cells were seeded in a 96-well plate overnight. On the next day, LNPs (fresh, freeze-thawed with sucrose solution and freeze-thawed without sucrose solution) were added to the cells to achieve an mRNA concentration of 1 μg ml$^{-1}$. The cells were incubated with the LNPs overnight. Transfection efficiency was measured using the SteadyGlo assay using the manufacturer's recommendations.

**Administration of mRNA nanoparticles to swine.** The mRNA LNPs were prepared as described for the in vitro studies. The firefly-luciferase-mRNA-loaded LNPs diluted in sucrose solution were transferred into the MUSAS and frozen at −20 °C for 1 h. Each MUSAS was loaded with 12.5 μg of mRNA. The MUSAS was applied to the pig buccal and pharynx manually with surgical forceps 8 h to 24 h before euthanasia. Immediately after the pig was euthanized, the site of administration and control were collected and placed in cold DMEM media (Thermo Fisher Scientific) containing 10% fetal bovine serum. Within 30 min of pig euthanasia, the tissue was immersed in 0.3 mg ml$^{-1}$ potassium luciferin solution (Gold Biotechnology) in PBS without calcium and magnesium. For increased diffusion of the substrate, luciferin solution was injected into the swine oesophageal tissue after submersion and before imaging. Bioluminescence was captured over a 30-min span with an in vivo imaging system (PerkinElmer) to capture bioluminescence, via LivingImage software 4.8.2 (Perkin-Elmer). The luminescent images were taken using Field of View D, automatic exposure time, medium binning, F/Stop = 1, and taking images every minute.

## Imaging

**Profilometry.** Sample surface roughness and three-dimensional reconstructions were quantitatively determined by an optical VK-X3000 profilometer (Keyence) using the ×5 and ×10 lens. The ring and lens lighting were used together and set to maximum intensity. To compare the smoothness across the specimens, the surface roughness parameter $S_a$ (areal average roughness) was evaluated across representative fields of view using the included VK viewer software v2.2.0.135 (Keyence).

**Confocal microscopy. Confocal microscopy to measure hydrostatic differentiation of MUSAS' underwater adhesion.** A near-infrared fluorescent dye solution, Sulfo-Cyanine5.5 (Cy5.5) (Thermo Fisher Scientific), was prepared at a concentration of 0.01 mg ml$^{-1}$ (40 ml) to stain 1 ml of water spread onto a microscopic glass slide for confocal microscopy. Confocal microscope imaging was performed on MUSAS before and after adhesion to the water-rich glass slide, using a 635-nm laser line with a measuring depth of 400 µm, with a FV1200 Laser Scanning Confocal Microscope (Olympus).

**Firefly-luciferase-mRNA transfection visualization in pharyngeal tissue using MUSAS via immunofluorescence confocal microscopy.** Fixed oesophageal pig tissues transfected for firefly luciferase expression with MUSAS along with untransfected controls were stained using DAPI (Thermo Fisher Scientific), as well as firefly luciferase polyclonal antibody primary antibody (Thermo Fisher Scientific) with a 1:2,000 dilution ratio, conjugated with goat anti-rabbit IgG (H+L) cross-adsorbed, Alexa Fluor 647 secondary antibody (Thermo Fisher Scientific) with a 1:500 dilution ratio. Five immunohistochemistry (IHC) slides per block of pig oesophageal tissue were prepared for imaging. Fluorescence images were taken with a FV1200 Laser Scanning Confocal Microscope (Olympus) with two channels: DAPI (405-nm laser line) and AlexaFluor647 (635-nm laser line). Images were taken using a ×10 objective. All images were processed using Fiji (ImageJ1.54) software.

**Scanning electron microscopy.** Before scanning electron microscopy, all samples were mounted inside a copper vise and then subjected to vacuum and liquid nitrogen inside the electron microscope vacuum cryo manipulation tool (Leica). Using the portable temperature-controlled vacuum arm, each cryogenic sample was transferred to the ACE 600 high-vacuum sputter coater (Leica), which cryo-fractured a fresh cross-section and sputter-coated approximately 10 nm of platinum. This conductive coating prevented excessive surface charging artefacts during scanning electron microscopy imaging. Using the same portable vacuum arm, each cryogenic sample was transferred to the Gemini 360 SEC scanning electron microscope (Zeiss). While under high vacuum, using the secondary electron detector, low-voltage imaging (2–3 kV) was used to prevent damage from electron bombardment while providing high surface detail. Typical imaging conditions would also entail a probe current near 2 nA and a working distance between 6 mm and 8 mm.

**Fluorescence microscopy.** An ex vivo study was performed to evaluate the bioavailability of a nanoparticle formulation delivered through MUSAS; 215 µl of fluorescent polystyrene nanoparticles were prepared with FluoSpheres Polystyrene Microspheres (Thermo Fisher Scientific) and stored at −80 °C. The fluorescent nanoparticles were then subcutaneously injected, smeared with a pipette or delivered via MUSAS to freshly collected porcine oesophagus tissue. The oesophagus tissues, including a negative control, were resected, rapidly frozen in OCT gel (Agar Scientific) using liquid nitrogen and sectioned using a cryostat microtome. Subsequently, the sectioned tissues were imaged using an EVOS fluorescence microscope (Life Technologies) with excitation and emission wavelengths of 580 nm and 605 nm, respectively.

## Bioanalytics

**In vitro HPLC.** Dissolution samples in simulated gastric fluid were directly analysed by HPLC and ultraviolet detection on a 1260 Infinity system (Agilent Technologies). Samples were injected at a volume of 2 µl onto an Agilent EC-C18 Poroshell column (3.0 × 50 mm, 2.7 µm particle size $d_p$) held at 25 °C. The mobile phase consisted of 0.1% formic acid in water (v + v, A) and acetonitrile (B), pumped at 800 µl min$^{-1}$ with a gradient programme of: 0 min, 5% B; 8 min, 60% B; 8.1 min, 95% B, over 10 min and with an equilibration time of 2 min. Eluite was quantified with a diode array detector at CAB's local absorption maximum at 258 nm in the ultraviolet region at 5 Hz.

**In vivo liquid chromatography–mass spectrometry.** Porcine serum samples were prepared via protein precipitation at a 1:3 volume ratio of serum to acetonitrile with bictegravir or verapamil as an internal standard. The liquid chromatography–mass spectrometry method used was validated according to FDA recommendations, on high-performance liquid chromatograph's coupled to Agilent 6495 triple quadrupole mass spectrometers in positive mode. Concentrations were calculated based on the linear regression of CAB response relative to internal standard response.

Specifically, samples were injected at a volume of 10 µl onto the same column using the same mobile phase above, but without temperature control. The gradient programme used a 400 µl min$^{-1}$ flow rate with 0 min, 5% B; 0.5 min, 5% B; 4 min, 95% B, with a run time of 5 min and equilibration time of 2 min. The Agilent Jet Stream source used the following parameters: gas temperature, 200 °C; gas flow, 14 l min$^{-1}$; nebulizer pressure, 20 psi; sheath gas temperature, 250 °C; sheath gas flow, 11 l min$^{-1}$; capillary voltage, 3,000 V; nozzle voltage, 1,500 V; high radiofrequency, 150 V; low radiofrequency, 60 V. The same transitions for CAB above were used, except both at 38-V collision energy. Verapamil was used as an internal standard, quantified with the transition from 455.1 $m/z$ to 303.1 $m/z$ at 35 V, and qualified with the transition from 455.1 $m/z$ to 303.1 $m/z$ at 40 V.

## Histology fixation

Unless specified, histology samples were fixed with 10% formalin for 24 h and then stored in 70% ethanol before embedded in paraffin for histopathological analysis.

## Statistical quantification and analysis

Statistical quantification and analysis were performed via Prism 9.3 (GraphPad). All error bars represent standard deviation (s.d.), except in pharmacokinetics studies conducted on independent Yorkshire pigs, where standard error of the mean (s.e.m.) was used to measure the population variability. For box plots, the box represents the median and the Q1 and Q3 quartiles (25% and 75%), and whiskers extend to the maximum and minimum values. For violin plots, the density distribution is calculated using the often-used Gaussian kernel density estimator, with dashed and dotted lines representing the median and the Q1 and Q3 quartiles (25% and 75%), respectively. The theoretically optimal Gaussian kernel density estimator, which minimizes the mean integrated squared error, is calculated with a bandwidth given by $1.06\hat{\sigma}n^{-1/5}$, where $\hat{\sigma}$ is the standard deviation and $n$ is the sample size. Student $t$-test and analysis of variance ($F$-test) were performed to compare differences between two groups, and among three or more groups, with details defined in the legends of relevant figures. We chose several $P$ values to systematically evaluate the statistical significance, including $P \le$ 0.05 (*) as the entry level of significance, $P \le 0.01$ (**) for highly significant, $P \le 0.001$ (***) and $P \le 0.0001$ (****) for extremely significant. The number of independent experiments of replicates and definition of significance level are further elaborated in each figure, figure caption and relevant methods section where statistical quantification and analysis was performed.

## Reporting summary

Further information on research design is available in the Nature Portfolio Reporting Summary linked to this article.

## Data availability

All data supporting the findings of this study are available within this paper, the Extended Data figures and Supplementary Information. Source data underlying the graphical representations used in the figures are available via GitHub at https://github.com/TroyKang/MUSAS.

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

**Acknowledgements** We are grateful for discussions with M. Boucher for histopathology, G. Liu and V. Feig for material synthesis, J. Li for drug delivery studies, A. R. Gomez, K. Nan, B. Ying and A. Nygren for early exploration of device fabrication, and D. Freitas and J. Jenkins for animal work. This work was supported in part by the following grants: Gates Foundation (INV-002177), Karl van Tassel (1925) Career Development Professorship, the Department of Mechanical Engineering, Massachusetts Institute of Technology (MIT), the Division of Gastroenterology, Brigham and Women's Hospital, the Advanced Research Projects Agency for Health (ARPA-H) under Award Number D24AC00040-00. The content is solely the responsibility of the authors and does not necessarily represent the official views of the Advanced Research Projects Agency for Health. C.P.K. was supported by the Morrissey College of Arts and Sciences, Boston College. M.G.S. acknowledges support from the Knut and Alice Wallenberg Foundation (KAW 2021.0317) for postdoctoral research at MIT. Y.C. was supported by the K. Lisa Yang Brain-Body Center Fellowship. We thank the Koch Institute's Robert A. Swanson (1969) Biotechnology Center at Massachusetts Institute of Technology for technical support, specifically the Hope Babette Tang (1983) Histology Facility, the Microscopy Core Facility, and the PMIT: Preclinical Imaging and Testing Facility. We thank Harvard Center for Nanoscale Systems for the assistance in cyro-scanning electron microscopy. We thank E. Kalodner-Martin for manuscript proofreading. We acknowledge V. E. Fulford, Alar Illustration (Figs. 2, 4 and 5) and Z. Zhou (Figs. 1 and 2, and Extended Data Fig. 1) for their illustration work.

**Author contributions** Z.K. and G.T. conceived of, designed and interpreted the research. Z.K. designed the device. Z.K., J.A.G. and A.M.R. fabricated, tested and validated the device. Z.K., C.P.K., F.X.C., C.L.C., J.A.G. and A.M.R. performed the remora studies. C.P.K. and Z.K. performed the micro-computed tomography. Z.K., Y.C. and A.H.T. developed and performed particle image velocimetry. Z.K., A.R.K., M.Z., F.X.C., C.L.C., I.D.B., R.D. and J.K. performed the material synthesis, drug delivery and organ-targeting studies. Z.K. and A.R.I. performed finite element analysis. Z.K., Y.C., I.M., Y.Y. and M.G.S. developed, tested and validated the electronics. B.N.M. and Z.K. performed the scanning electron microscopy and profilometry. J.M., T.S., A.L. and A.E.E. performed the bioanalytics. Z.K., I.M., Y.Y. and M.M. designed and validated the device fabrication process. A.P., Z.K., N.F., A.G., B.L., K.S. and A.M.H. performed the in vivo swine studies. Z.K., C.P.K. and Y.C. performed the in vivo fish studies. A.M.R. (Figs. 1 and 2 and Supplementary Information), Z.K. (Figs. 1 and 2, and Extended Data Figs. 1 and 3–5), J.A.G. (Supplementary Information) contributed to portions of the illustration work listed. Z.K., G.T., C.P.K. and A.R.K. wrote the paper. All authors edited and approved the paper.

**Competing interests** Z.K., G.T., A.R.K., M.Z. and Y.C. are co-inventors on a patent application describing the system reported: US Provisional Application No. 63/702,561. Complete details of all relationships for profit and not for profit for G.T. can found at https://www.dropbox.com/sh/szi7vnr4a2ajb56/AABs5N5iOq9AfT1IqIJAE-T5a?dl=0. The other authors declare no competing interests.

**Additional information**
**Correspondence and requests for materials** should be addressed to Giovanni Traverso.

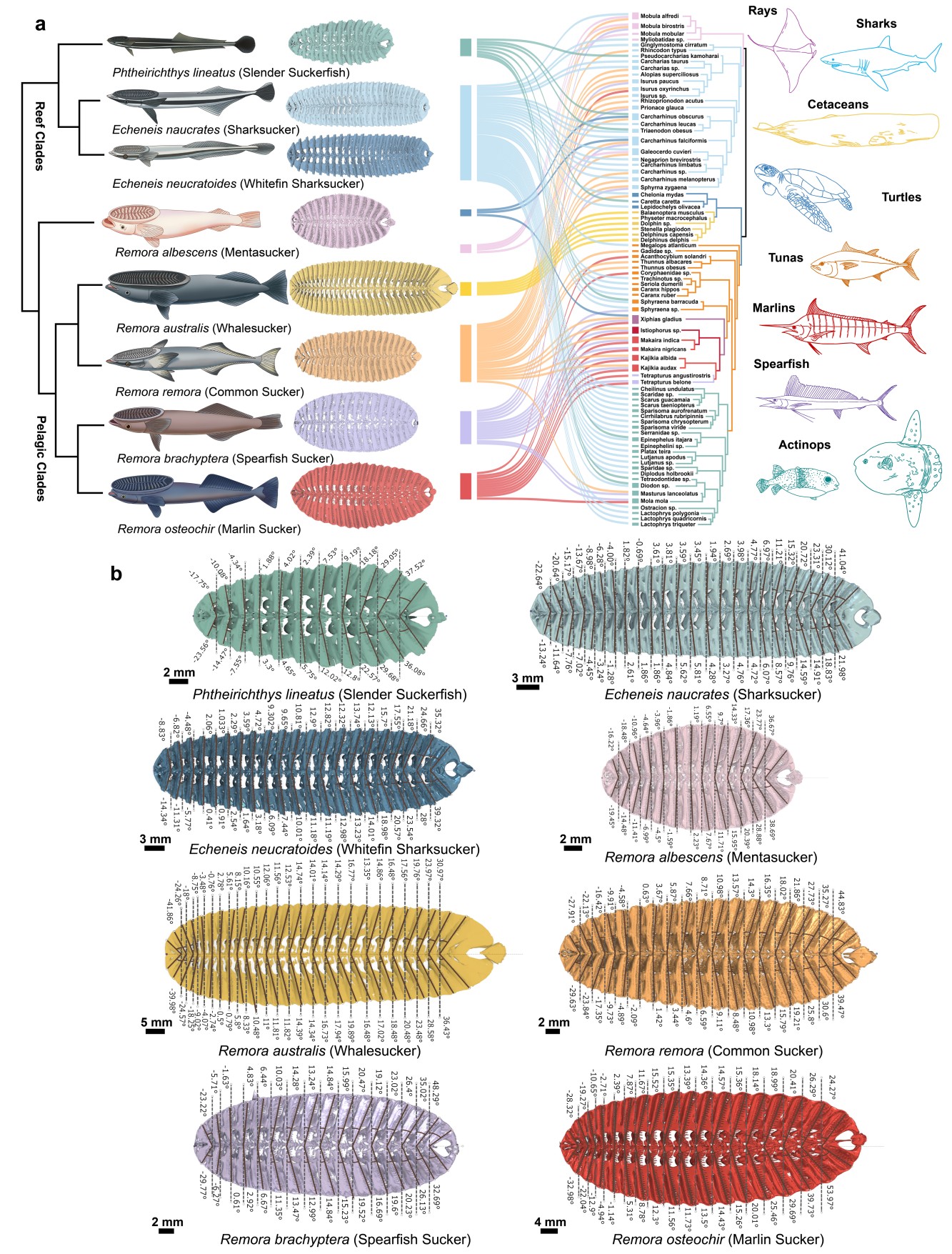

**Extended Data Fig. 1 | Overview of host-specific adaptations in remora adhesive disc anatomy. a**, Cophylogenetic tanglegram of remora-host association[20–25], and dorsal view μ-CT scan of anatomy of remora adhesive disk across species. **b**, Representative measurement of lamella orientation of remora species from a dorsal view.

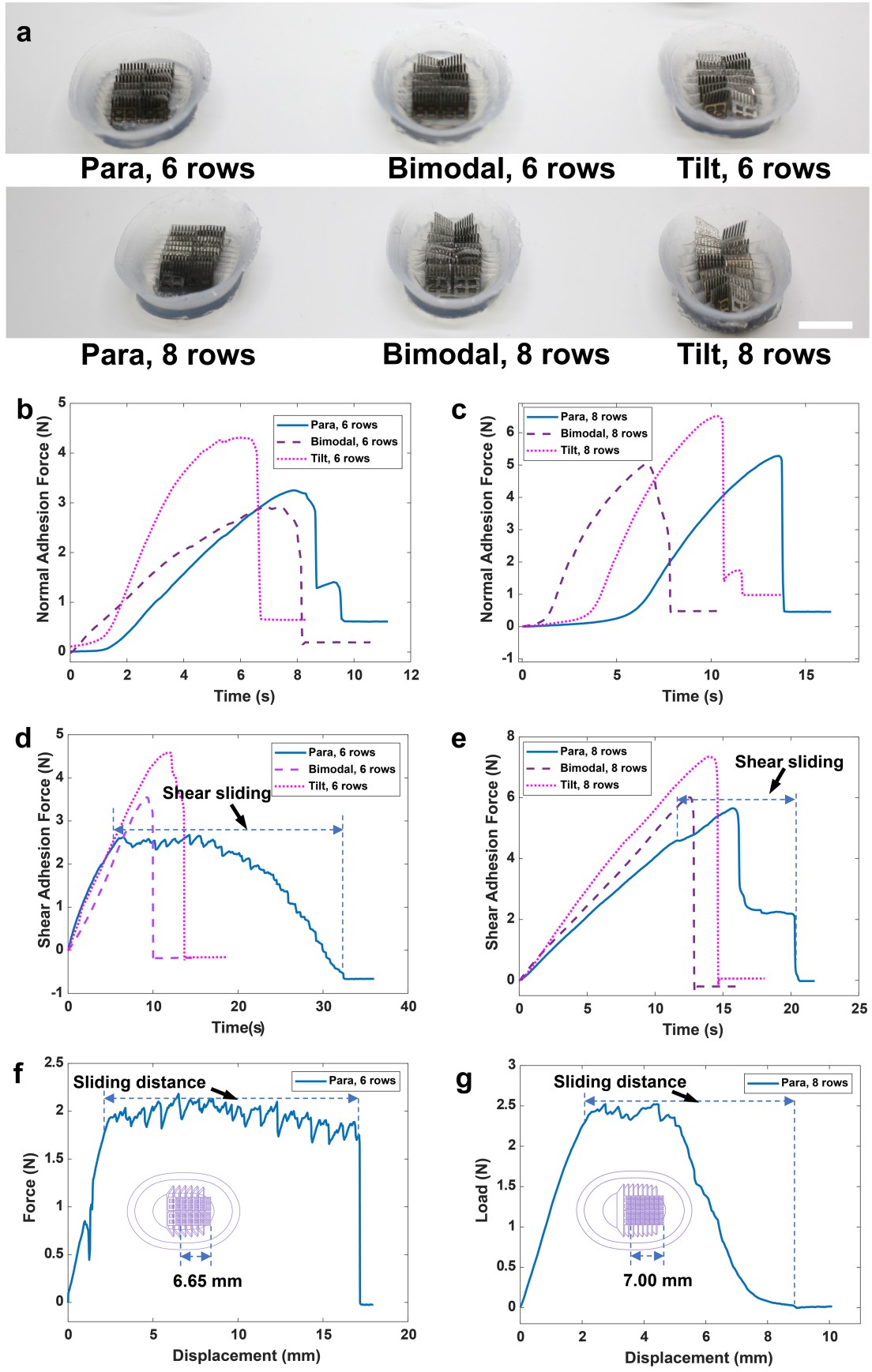

**Extended Data Fig. 2** | See next page for caption.

**Extended Data Fig. 2 | Adhesion performance of MUSAS with various lamella orientations and rows. a**, Photograph of unimodal parallel-angled (para), a bimodal mixture of parallel and tilted-angled (bimodal), and tilted-angled (tilt) MUSAS with 6 and 8 rows of lamella (scale bar: 5 mm). **b,c**, Representative measurements of normal adhesion forces of MUSAS with different lamella orientations and number of rows (n = 5 devices per design). **d,e**, Representative measurements of shear adhesion forces of MUSAS with various lamella orientations and number of rows; the shear sliding ratio is calculated as the time MUSAS maintains shear adhesion after reaching the maximum or plateau, relative to the total time of the shear drag test before losing adhesion (n = 5 devices per design). **f,g**, Representative measurements of shear sliding distance (n = 5 devices per design).

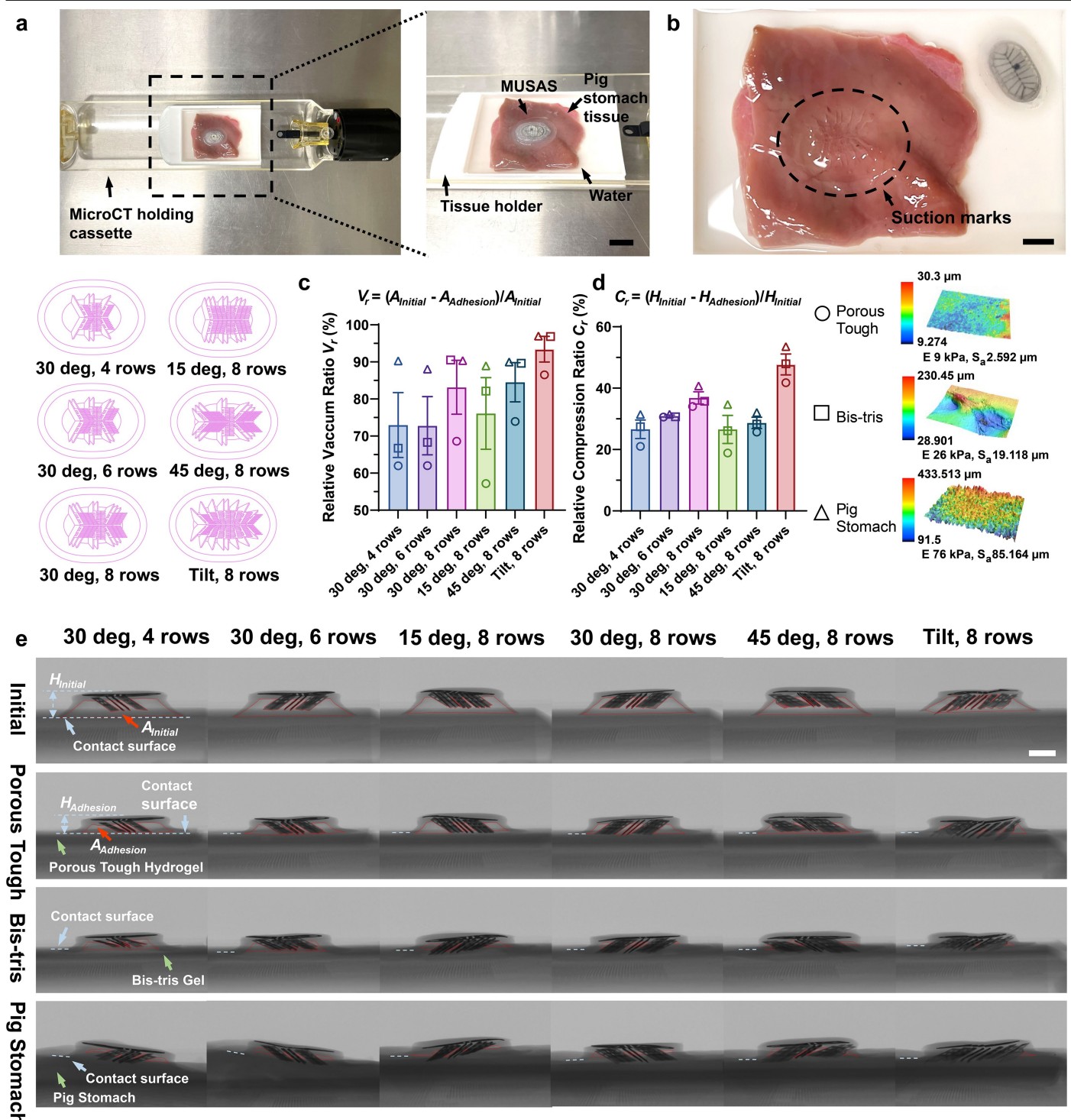

**Extended Data Fig. 3 | μ-CT imaging of internal volume changes in MUSAS with varying angles of lamella orientation and row numbers, adhering to soft substrates with distinct stiffness and roughness. a**, Setup of μ-CT imaging (scale bar: left, 1 cm; right, 5 mm). **b**, Representative image of suction marks left by MUSAS after adhesion to pig stomach tissue, demonstrating non-homogeneous substrate morphing. **c**,**d**, Relative vacuum ratio (**c**) and relative compression ratio (**d**) of different MUSAS designs adhering to representative soft substrates with distinct stiffness and roughness (n = 3 substrates, error bars represent mean ± s.d.; colour bars represent surface height). **e**, Representative μ-CT imaging of MUSAS with varying lamella orientation angles and row numbers, adhering to soft substrates with distinct stiffness and roughness. The red box represents the non-vacuum area $A_{Initial}$ or $A_{Adhesion}$ (scale bar: 5 mm).

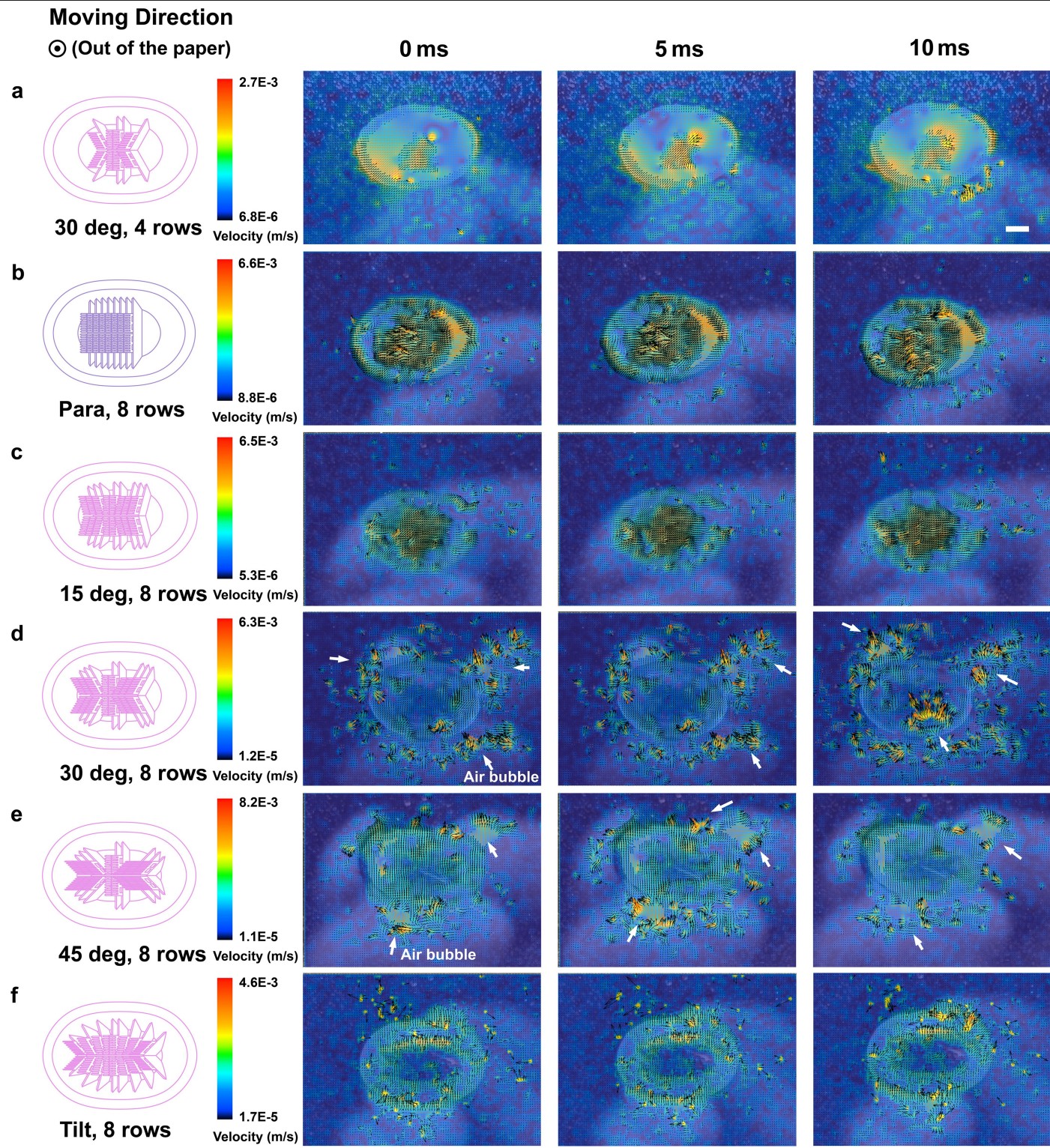

**Extended Data Fig. 4 | Particle imaging velocimetry (PIV) assessment of MUSAS hydrodynamics with varying angle of lamella orientations and row numbers, demonstrated in instantaneous velocity fields. a**, Bottom-view fluid velocity maps reveal that fewer lamella rows result in minimal water expulsion, as seen in the 4 rows design. **b,c**, In terms of 8 rows design, unsubstantial angle variations incline to unidirectional water expulsion, making the performance of the 15 degree dominated design (**c**) nearly identical to that of the parallel-angled configuration (**b**). **d,e**, While lamella orientations dominated by 30 degree (**d**) and 45 degree (**e**) induce multidirectional water expulsion, the flow is often uneven and accompanied by large air bubble formation in specific regions. **f**, Notably, the tilt-dominant design with the substantial lamella angle variation exhibits superior efficiency in achieving homogeneous, multidirectional water expulsion (scale bar: 5 mm).

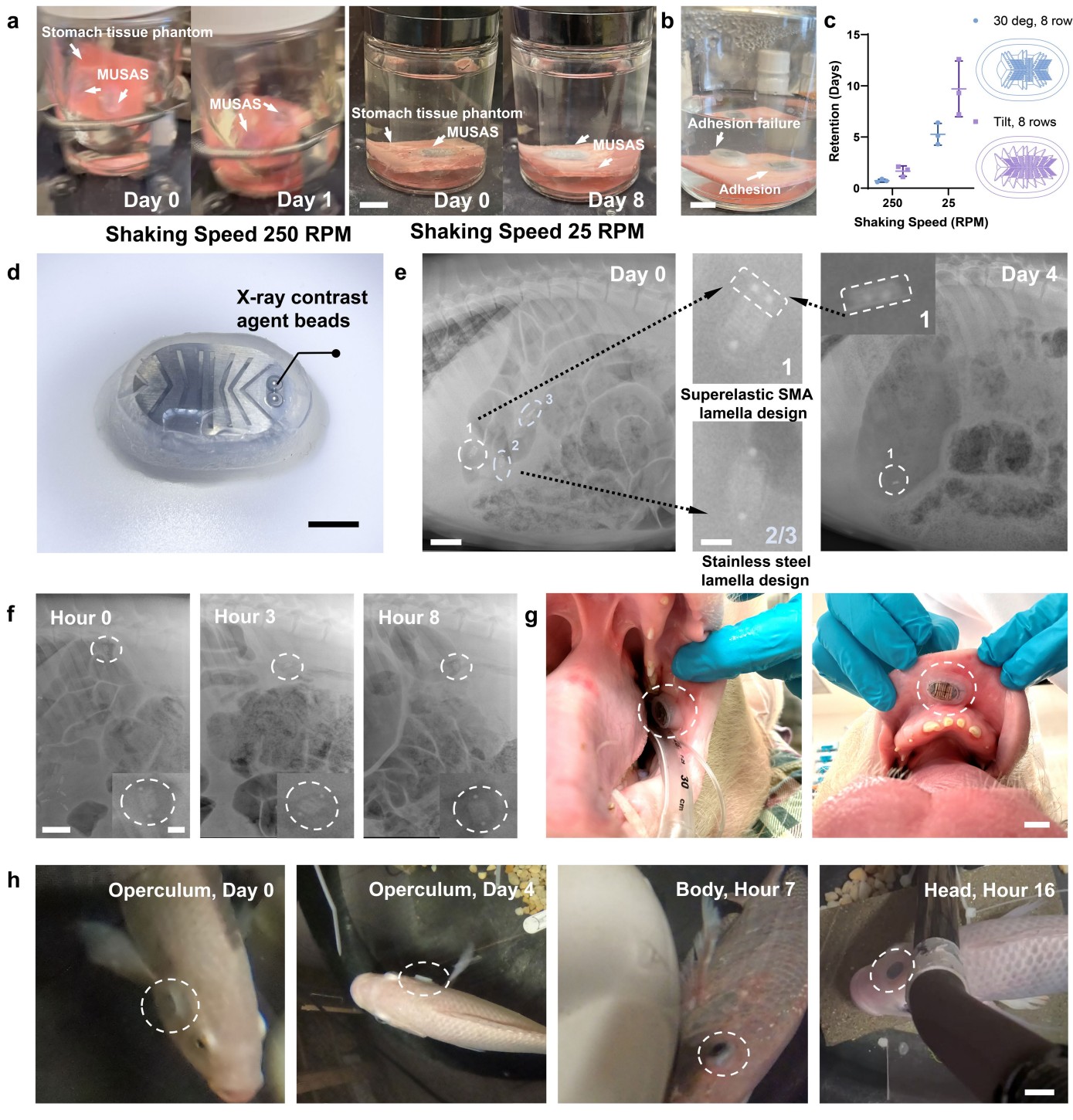

**Extended Data Fig. 5 | Representative in vitro and in vivo evaluation of adhesion and retention performance of MUSAS. a**, Representative retention performance of optimal tilt-dominant MUSAS under dynamic shaking interference in a 37 °C incubator (New Brunswick Innova 40/40 R, see Supp. Video S6 for further demonstration, scale bar: 1 cm). **b**, Illustration of the retention failure mode (scale bar: 1 cm). **c**, Dynamic interference evaluation of MUSAS retention performance with different angles of lamella orientation (n = 3 devices per design, error bars represent mean ± s.d.). **d**, MUSAS with radioactive imaging agents for routine X-rays monitoring (scale bar: 5 mm).

**e**, Representative X-rays depicting retention of MUSAS with superelastic nitinol lamella and stainless steel lamella delivered in the stomach before safe passage in the GI tract (scale bars: left 5 cm, right 1 cm). **f**, Representative retention performance of MUSAS in the duodenum of the small intestine (SI) in a swine model, evaluated in a terminal study on the day of euthanasia (scale bars: left 5 cm, right 1 cm). **g**, Adhesion of MUSAS in various buccal regions for mRNA delivery (scale bar: 1 cm). **h**, Representative retention performance of MUSAS on different body surfaces of a tilapia model (scale bar: 2 cm).

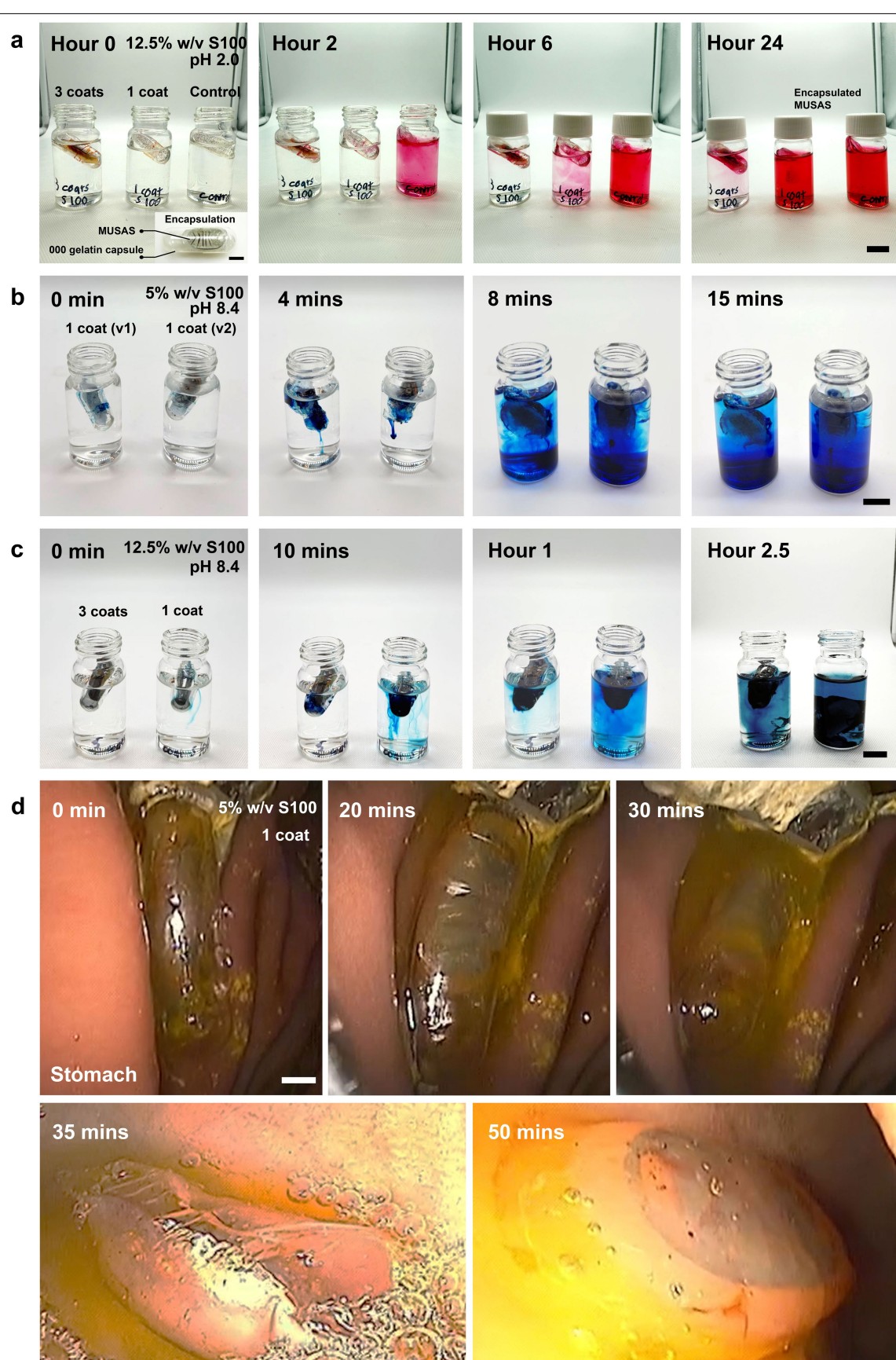

**Extended Data Fig. 6** | See next page for caption.

**Extended Data Fig. 6 | pH-responsive coating enabled programmable targeted delivery of MUSAS to the small intestine. a**, In vitro demonstration of the prolonged stability of MUSAS encapsulation in simulated gastric fluid, dip-coated with the pH-dependent copolymer Eudragit S100 (scale bars: left, 5 mm; right, 1 cm). **b,c**, In vitro demonstration of the programmable controlled release of MUSAS in a simulated intestinal environment, dip-coated with different concentrations of the copolymer Eudragit S100 (scale bars: 1 cm). **d**, In vivo demonstration of programmable targeted delivery of MUSAS to the small intestine in a swine model, including safe gastric emptying in the stomach from 0 to 30 min and timely deployment in the small intestine from 35 mins to 50 mins (scale bars: 5 mm).

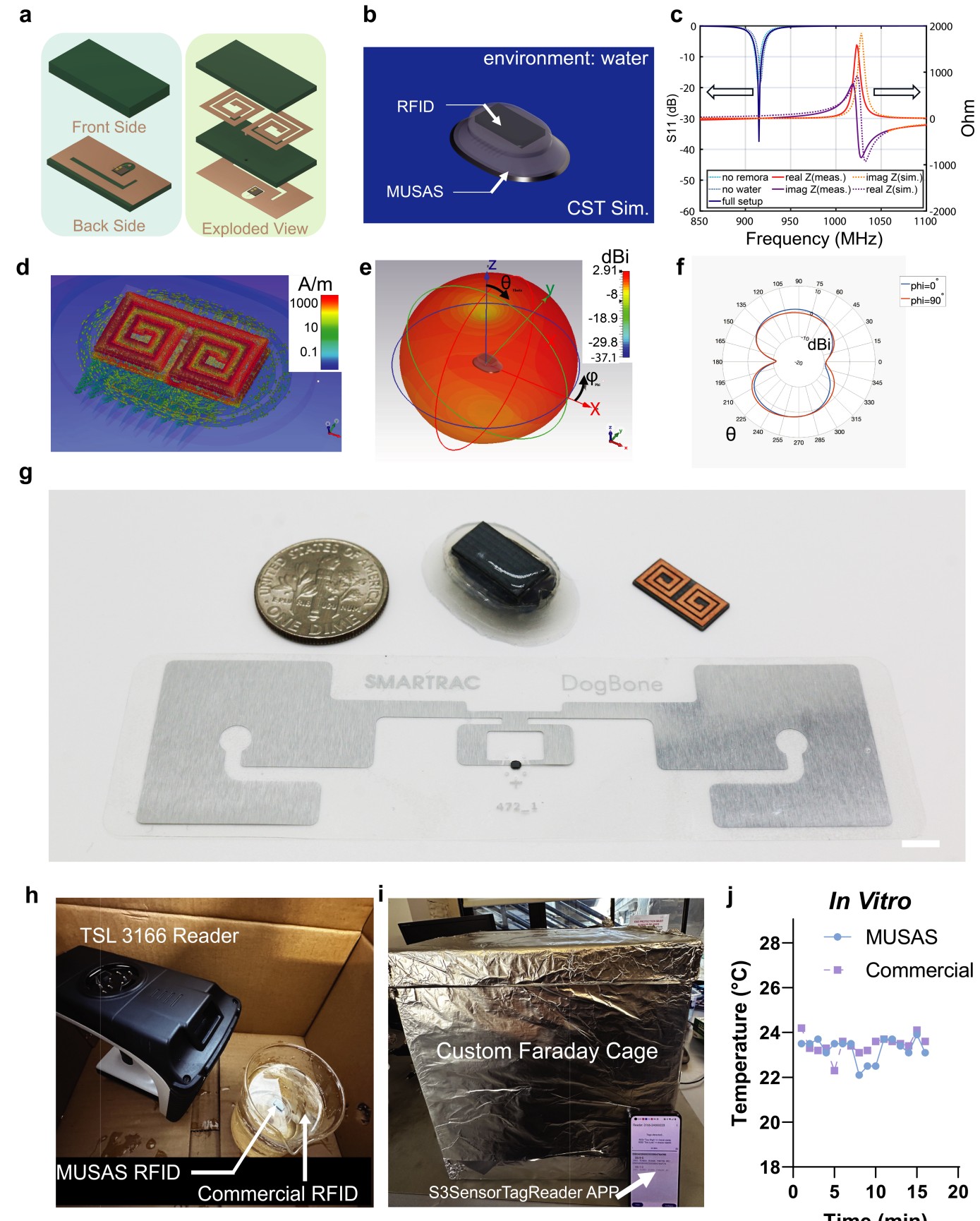

**Extended Data Fig. 7** | See next page for caption.

**Extended Data Fig. 7 | Design, simulation and in vitro characterization of RFID temperature sensor. a**, Schematic of the proposed RFID temperature sensor. The RFID sensor comprises a ground plane connected to a Magnus-S RFID temperature sensor chip (Axzon), a substrate, an antenna layer, and a superstrate. **b**, Full simulation setup in CST electromagnetic software (Simulia), featuring the RFID antenna encased in Ecoflex polymer and attached to a MUSAS, immersed in an underwater environment. **c**, Simulated S11 results for three scenarios: "no MUSAS" (RFID sensor in water without MUSAS), "no water" (RFID sensor attached to MUSAS in air), and "full setup" (complete assembly as shown in **b**). Measured (meas.) and simulated (sim.) impedance values are compared, showing significant overlap in both real and imaginary components, thereby validating the simulation accuracy. **d**, Simulated surface current distribution at the resonant frequency with a zero-degree phase. **e**, 3D far-field radiation pattern. f. 2D far-field radiation pattern. MUSAS, while not electrically connected to the RFID antenna, influences current distribution and causes distortion in the antenna's backside radiation pattern. **g**, Comparison between the commercial Smartrac RFID temperature tag (Avery Dennison) and the MUSAS RFID temperature sensor (scale bar, 5 mm). **h**, Close-up view showing the positioning of the TSL 3166 reader (Technology Solution UK Ltd) and the RFID tags under in vitro test. **i**, Custom Faraday cage setup for the in vitro test, including the S3SensorTagReader app installed on an Android smartphone for collecting temperature and received signal strength indicator (RSSI) data. **j**, In vitro underwater temperature reading of MUSAS RFID tag, compared to a commercial sensor (**g**, Avery Dennison).

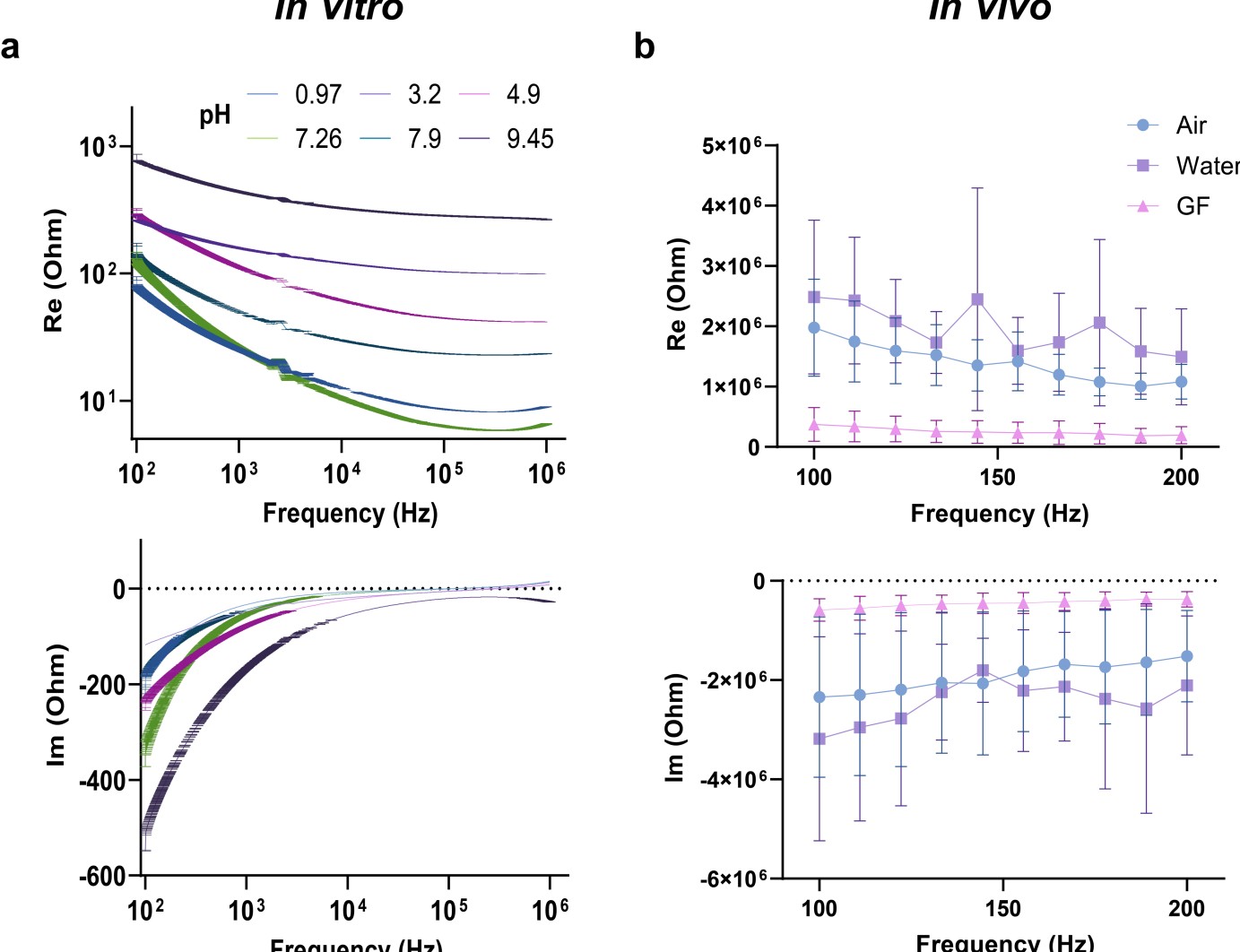

**Extended Data Fig. 8 | In vitro and in vivo validation of MUSAS-based impedance sensor for GERD monitoring. a**, In vitro resistive (real, Re) part and reactive (imaginary, Im) impedance for fluid pH differentiation (n = 5 independent measurements, error bars represent mean ± s.d.). **b**, In vivo differentiation of air inhalation, water, and gastric fluid (GF) consumption via MUSAS-based impedance sensor in the swine oesophagus (n = 5 independent measurements, error bars represent mean ± s.d.).

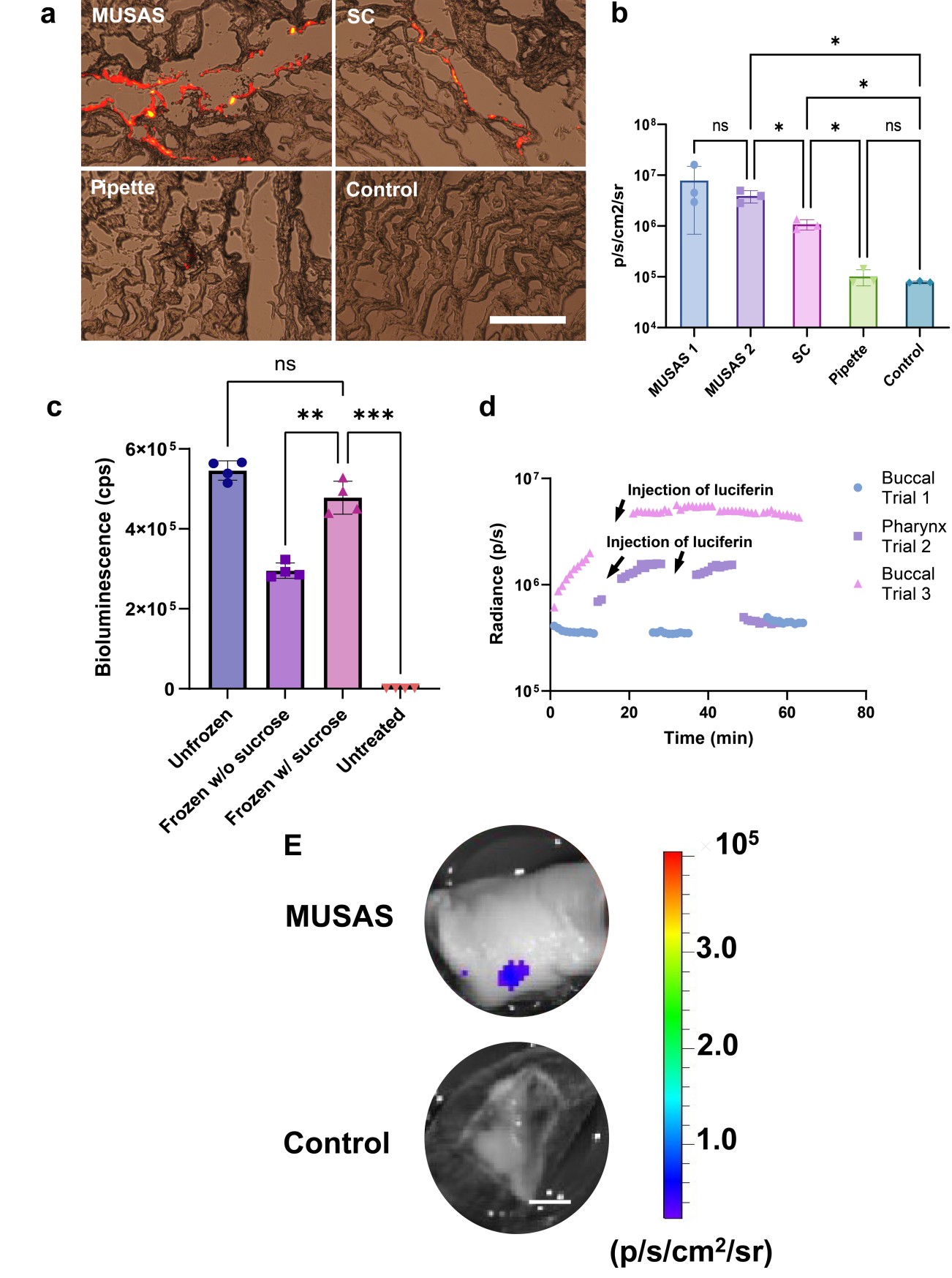

**Extended Data Fig. 9 | Extended in vitro and in vivo characterization of MUSAS-enabled mRNA delivery. a**, Ex vivo delivery of fluorescent nanoparticles via MUSAS, subcutaneous injection (SC), and pipette smearing, compared to negative control on oesophagus tissue (scale bar: 200 μm). **b**, Bioavailability of fluorescent nanoparticles (n = 3 samples per treatments, error bars represent mean ± s.d.). **c**, In vitro studies of mRNA transfection efficacy with different preparation methods of LNPs (unfrozen, frozen and thawed without sucrose, frozen and thawed with sucrose) for delivery to human oral epithelial cells (n = 4 samples per formulation, error bars represent mean ± s.d.). **d**, Time course of the IVIS radiance signal to evaluate transfection of luciferase in the buccal and pharyngeal regions of a swine model. **e**, In vivo trial 3 of delivering luciferase mRNA to the buccal region (scale bar: 1 cm; colour bars denote radiance measurements; see Fig. 5 for trial 1 and trial 2). Observing substantial variance differences between groups, Brown-Forsythe and Welch ANOVA with Dunnett's T3 multiple comparison test (**b**) to compare different treatments and unpaired *t*-test with Welch's correction (**c**) to compare different formulations were used, statistical significance was indicated as follows: non-significant (ns), $p \le 0.05$ (*), $p \le 0.01$ (**), $p \le 0.001$ (***).

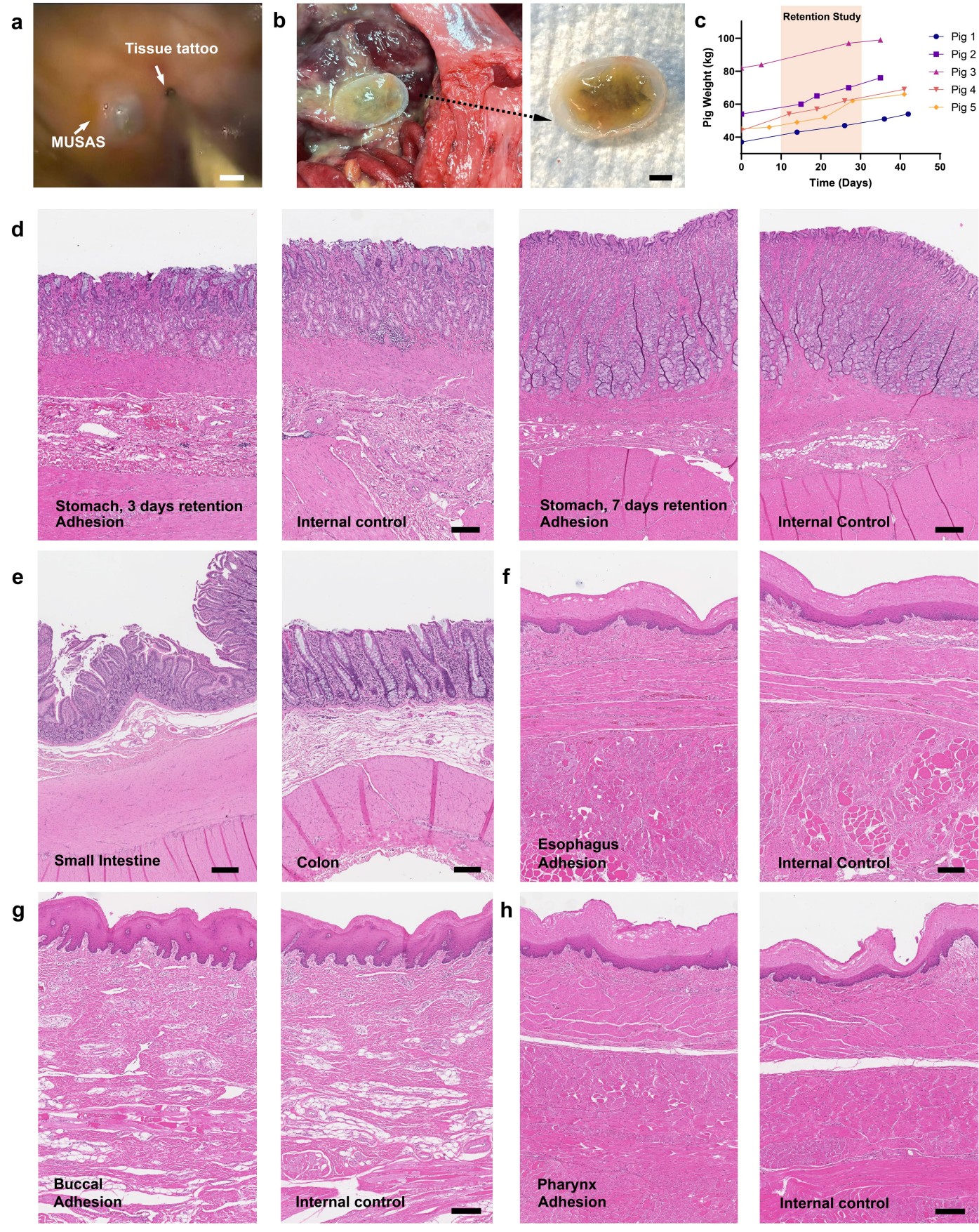

**Extended Data Fig. 10** | See next page for caption.

**Extended Data Fig. 10 | In vivo biocompatibility evaluation of MUSAS in a swine model. a**, Illustration of tattooing adhesion sites in a swine model (scale bar: 5 mm). **b**, Detached MUSAS retrieved from a pig stomach after euthanasia, demonstrating thorough lubrication and accumulation of stomach contents and mucoid materials for safe passage (scale bar: 5 mm). **c**, Representative pig weight measurements show normal weight gain during MUSAS residence in the stomach (n = 4 devices delivered per retention study). **d**–**h**, Histopathology of H&E stained the gastrointestinal tract showing no damage from MUSAS adhesion and passage that causes haemorrhage, inflammation or indication of tissue repair (fibrosis), including **d**, stomach. MUSAS adhesion and adjacent internal control sites, collected after 3 and 7 days of MUSAS residence (scale bar: 100 μm, n = 9 Yorkshire pigs evaluated). **e**, Small intestine and colon, collected after MUSAS residence in the stomach and safe passage through the GI tract (scale bars: small intestine, 200 μm; colon, 100 μm, n = 4 Yorkshire pigs evaluated). **f**, Oesophagus, adhesion and adjacent internal control sites, collected 1 day after adhesion (scale bar: 200 μm, n = 3 Yorkshire pigs evaluated). **g**,**h**, Buccal tissue and pharynx, adhesion and adjacent internal control sites, collected 1 day after mRNA delivery (scale bar: 100 μm, n = 3 Yorkshire pigs evaluated).

# Reporting Summary

## Statistics

For all statistical analyses, confirm that the following items are present in the figure legend, table legend, main text, or Methods section.

| n/a | Confirmed | |
|---|---|---|
| ☐ | ☒ | The exact sample size (*n*) for each experimental group/condition, given as a discrete number and unit of measurement |
| ☐ | ☒ | A statement on whether measurements were taken from distinct samples or whether the same sample was measured repeatedly |
| ☐ | ☒ | The statistical test(s) used AND whether they are one- or two-sided *Only common tests should be described solely by name; describe more complex techniques in the Methods section.* |
| ☐ | ☒ | A description of all covariates tested |
| ☐ | ☒ | A description of any assumptions or corrections, such as tests of normality and adjustment for multiple comparisons |
| ☐ | ☒ | A full description of the statistical parameters including central tendency (e.g. means) or other basic estimates (e.g. regression coefficient) AND variation (e.g. standard deviation) or associated estimates of uncertainty (e.g. confidence intervals) |
| ☐ | ☒ | For null hypothesis testing, the test statistic (e.g. *F*, *t*, *r*) with confidence intervals, effect sizes, degrees of freedom and *P* value noted *Give P values as exact values whenever suitable.* |
| ☒ | ☐ | For Bayesian analysis, information on the choice of priors and Markov chain Monte Carlo settings |
| ☒ | ☐ | For hierarchical and complex designs, identification of the appropriate level for tests and full reporting of outcomes |
| ☒ | ☐ | Estimates of effect sizes (e.g. Cohen's *d*, Pearson's *r*), indicating how they were calculated |

*Our web collection on statistics for biologists contains articles on many of the points above.*

## Software and code

Policy information about availability of computer code

| Data collection | Bluehill V3.11 (Instron); MESUR Lite 2.0.0 (Mark-10); TSL ASCII Software Development Kit – for Android v2.8.0 (Technology Solutions UK LTD); Sciospec software 2.0.8 (Sciospec); SkyScan 1173 and SkyScan 1276 application software (Bruker); VK Viwer 2.2.0.135 (Keyence); LivingImage 4.8.2 (PerkinElmer); Abaqus 2021 (SIMULIA); COMSOL multiphysics 6.2 (COMSOL); CST studio suite 2022 (SIMULIA) |
|---|---|
| Data analysis | MATLAB R2022a (Mathworks), Prism 9.3 (Graphpad), ImageJ 1.54 |

For manuscripts utilizing custom algorithms or software that are central to the research but not yet described in published literature, software must be made available to editors and reviewers. We strongly encourage code deposition in a community repository (e.g. GitHub). See the Nature Portfolio guidelines for submitting code & software for further information.

## Data

Policy information about availability of data

All manuscripts must include a data availability statement. This statement should provide the following information, where applicable:

- Accession codes, unique identifiers, or web links for publicly available datasets
- A description of any restrictions on data availability
- For clinical datasets or third party data, please ensure that the statement adheres to our policy

All data supporting the findings of this study are available within this paper, the Extended Data Figures and Supplementary Information. Source data underlying the graphical representations used in the figures are available in https://github.com/TroyKang/MUSAS.

## Research involving human participants, their data, or biological material

Policy information about studies with human participants or human data. See also policy information about sex, gender (identity/presentation), and sexual orientation and race, ethnicity and racism.

| | |
|---|---|
| Reporting on sex and gender | N/A |
| Reporting on race, ethnicity, or other socially relevant groupings | N/A |
| Population characteristics | N/A |
| Recruitment | N/A |
| Ethics oversight | N/A |

Note that full information on the approval of the study protocol must also be provided in the manuscript.

# Field-specific reporting

Please select the one below that is the best fit for your research. If you are not sure, read the appropriate sections before making your selection.

☒ Life sciences  ☐ Behavioural & social sciences  ☐ Ecological, evolutionary & environmental sciences

For a reference copy of the document with all sections, see nature.com/documents/nr-reporting-summary-flat.pdf

# Life sciences study design

All studies must disclose on these points even when the disclosure is negative.

| | |
|---|---|
| Sample size | Sample sizes were not determined due to the proof-of-concept nature of this study. Here, we are conducting small-scale experiments and exploratory investigations to assess the feasibility and viability of the device concept. The primary objective is to gather initial evidence or insights, rather than to determine a sample size based on statistical considerations. |
| Data exclusions | No data were excluded. |
| Replication | Ex vivo studies for mechanical characterization were performed on freshly harvested tissue (< 1-hour post-euthanasia) without surface washout, tissue trimming, or liquid removal. n = 2-4 tissue samples were used per MUSAS design or adhesives of interest. In vivo studies of MUSAS were performed in over 58 pigs at various GI locations (buccal cavity, esophagus, stomach, and small intestine) and 8 fish, with GI retention evaluated in over 9 pigs and body surface retention in over 6 fish in total under survival conditions—defined as uninterrupted normal feeding, resting, and behaviors. In addition, treatment studies of MUSAS were validated in n = 3 pigs. Specific n values and appropriate statistics (mean, median, quartiles, whiskers, error bars) with detail methods are described in the figures legends. All attempts at replication were successful and all experimental data were reproducible between independent experiments. |
| Randomization | Randomization of devices and animals was conducted to minimize physiological bias and enhance the validity of experimental results. Additionally, controlled experimental conditions (e.g., animal pre-treatment) and counterbalancing (weight, tissue harvesting order) were introduced where deemed meaningful. (See Methods section) |
| Blinding | No blinding was conducted as the studies focused on objective measurements and observations where inanimate objects were studied. |

# Reporting for specific materials, systems and methods

We require information from authors about some types of materials, experimental systems and methods used in many studies. Here, indicate whether each material, system or method listed is relevant to your study. If you are not sure if a list item applies to your research, read the appropriate section before selecting a response.

## Materials & experimental systems

| n/a | Involved in the study |
|-----|----------------------|
| ☐ | ☒ Antibodies |
| ☐ | ☒ Eukaryotic cell lines |
| ☒ | ☐ Palaeontology and archaeology |
| ☐ | ☒ Animals and other organisms |
| ☒ | ☐ Clinical data |
| ☒ | ☐ Dual use research of concern |
| ☒ | ☐ Plants |

## Methods

| n/a | Involved in the study |
|-----|----------------------|
| ☒ | ☐ ChIP-seq |
| ☒ | ☐ Flow cytometry |
| ☒ | ☐ MRI-based neuroimaging |

## Antibodies

| | |
|---|---|
| Antibodies used | Firefly luciferase polyclonal primary antibody (Thermal Fisher Scientific) with 1:2000 dilution ratio, conjugated with goat anti-rabbit IgG (H+L) cross-absorbed, Alexa Fluor™ 647 secondary antibody (Thermo Fisher Scientific) with 1:500 dilution were used for validating luciferase transfection in pig tissue |
| Validation | This Antibody is commercially available and was verified by Cell treatment to ensure that the antibody binds to the antigen stated by the manufacture. Validation statements and relevant citations published on the manufacturer's website are listed as follows. Firefly luciferase polyclonal primary antibody: https://www.thermofisher.com/antibody/product/Firefly-luciferase-Antibody-Polyclonal/PA5-32209 Alexa Fluor™ 647 secondary antibody: https://www.thermofisher.com/antibody/product/Goat-anti-Rabbit-IgG-H-L-Cross-Adsorbed-Secondary-Antibody-Polyclonal/A-21244 |

## Eukaryotic cell lines

Policy information about cell lines and Sex and Gender in Research

| | |
|---|---|
| Cell line source(s) | Human oral epithelial primary cell culture (Celprogen) was used for in vitro evaluation of transfection efficacy of frozen LNP formulation. |
| Authentication | Primary cell cultures were verified by manufacturer, and published papers. In addition, primary cell cultures were frequently checked by their morphological features. Authentication statements and relevant citations published on the manufacturer's website are listed as follows. Human oral epithelial primary cell culture: https://celprogen.com/human-oral-epithelial-primary-cell-culture-frozen-vial/ |
| Mycoplasma contamination | All cell lines were regularly tested to be mycoplasma-negative by commercial test kit (Lonza). |
| Commonly misidentified lines (See ICLAC register) | No commonly misidentified cell lines were used in this research. |

## Animals and other research organisms

Policy information about studies involving animals; ARRIVE guidelines recommended for reporting animal research, and Sex and Gender in Research

| | |
|---|---|
| Laboratory animals | Yorkshire pigs (Tufts), female, 55-95 kg, 3-5 months old; gouramis (Osphronemidae) (Petsmart), female, 5 cm body length (BL), 5-7 month old; tilapias (Oreochromis) (procured from a local supermarket), 6-9 month old, sex undetermined, 20 cm BL; remoras (Echeneis naucrates) (procured from a local fish store), age and sex undetermined, 10 cm BL |
| Wild animals | The study did not involved wild animals |
| Reporting on sex | The primary objective of the studies conducted was to investigate the mechanism of mechanical adhesion, with a specific emphasis on validating MUSAS as a versatile platform for biomedical sensing and drug delivery. Furthermore, the study design aimed to minimize potential confounding factors and sources of variation that could influence the outcomes. A standardized experimental setup was selected to enhance the ability to detect and evaluate the effects of the specific adhesive parameters or intervention under investigation. While recognizing that sex can play a significant role in certain areas of research, the specific study did not involve aspects where sex-specific findings were expected or directly relevant to the research objectives. |
| Field-collected samples | This research did not include field-collected samples |
| Ethics oversight | Committee on Animal Care at MIT and the Institutional Animal Care and Use Committee of Boston College |

Note that full information on the approval of the study protocol must also be provided in the manuscript.

## Plants

Seed stocks

N/A

Novel plant genotypes

N/A

Authentication

N/A

