## [Peer Review File · Nature]

Mechanical Underwater Adhesive Devices for Soft Substrates

Corresponding Author: Professor Giovanni Traverso

Version 0:

Reviewer comments:

Referee #1

(Remarks to the Author)

The manuscript (2024-09-19511) by Traverso's group on the mechanism and mimetics of remora attachment to different marine hosts is impressively exemplary in several respects: 1) the diversity of anatomical designs, 2) the anatomical details of lamellar organization, 3) the identification of essential features for adhesion, 4) the mechanical testing of natural and mimetic discs to specified surfaces, and 5) the exploration of specific need-driven applications. The work is clearly articulated and presented. Results are appropriately tested with replicates/controls.

To this reviewer's eye, however, there may be two less exemplary areas that pertain to the validity of approach: A) the authors seem to be persuaded that the biochemistry and natural supramolecular organization are irrelevant to better understanding and mimicking/translating this system. This resembles the drift of earlier efforts on gecko adhesion with respect to setal structure in the 10 -1000 nm length scale range. This blind-eye to biochemistry has impeded an important reckoning of how to keep setae from fouling with ageing/use among other issues. This could easily be an issue here as well. B) The second area has to do with lamellar architecture in which aligned bayonet-like spinules appear to be embedded in softer lip-like flaps. There is no mention (or did I miss it?) of the stiffness mismatch between the spinules and the flaps. If spinules resemble the modulus of martensite, how do lamellae circumvent extensive contact damage at the spinule/tissue interface?

Referee #2

(Remarks to the Author)

In this study, the authors developed a Mechanical Underwater Soft Adhesion System (MUSAS) inspired by remoras, which utilize sucker-like adhesive disks to adhere to various substrates. The authors analyzed and summarized in detail the angles and distributions of lamellar structures in various remora species for adhesion processes and contact targets. Based on this analysis, the MUSAS system addresses the challenges of underwater adhesion to soft substrates by mimicking the anatomical and behavioral mechanisms of remoras. The authors demonstrated that MUSAS exhibits versatility, capable of adhering to soft substrates with varying surface textures and environmental conditions, achieving a high adhesion-force-to-weight ratio. The platform is insensitive to environmental pH and humidity, demonstrating potential applications in a variety of settings, including noninvasive gastrointestinal drug delivery and environmental biosensing.

However, this paper requires rigorous evaluation of novelty, validation of claims, and areas of potential improvement compared to previous studies. Although the authors have developed a material optimized for soft surfaces and proposed applications through studies on the angles of the remora of the suction cup structure, the underwater adhesion concept and some application technologies of the lamellar-based sucker-mimicking adhesive material have been reported in previous studies (doi/10.1126/scirobotics.abm6695; doi/full/10.1126/scirobotics.aan8072;doi.org/10.1088/1748-3190/ab9418; doi.org/10.1016/j.matt.2020.01.018), so it is difficult to judge its novelty as the best. In particular, adhesion to soft surfaces has already been demonstrated depending on the number of lamellar structures of the artificial sucker fish (doi/10.1126/scirobotics.abm6695). Although the angle of the proposed lamella structure of the MUSAS is described as being deeply related to the adhesive interaction with soft or hard tissues, an in-depth methodological analysis of this relationship has not been presented. Additionally, an insightful analysis of the activation process of the suction-based adhesive interaction and the resulting mechanism of adhesion change is lacking. The correlation between the change in the internal volume of the MUSAS structure and the lamellar angle, the number of lamellar structures, and the smoothness of the

target surface should be elucidated.

The authors seem to have only summarized the concept (without sufficient verification data) of various series of application technologies using their adhesive material. Rather, it seems more powerful to thoroughly analyze and verify one application. From an application perspective, an in-depth elucidation of specific mechanisms, as well as thorough verification of biocompatibility and stability with controllable adhesion capability, are essential to emphasize its distinctive advantages over other bioadhesive studies in their applications into ingestible devices.

Overall, considering the quality standards of Nature, this manuscript would be more suitable for other specialized journals. For additional questions, please see below.

To further strengthen the soft tissue interaction aspect claimed by the authors, a comprehensive analysis of the adhesive interaction with soft or hard tissues according to the angle of the proposed lamellar structure of MUSAS should be incorporated. For instance, it is pertinent to examine how the adhesive interaction with various modulus samples varies according to the angle of the lamellar structure. Additionally, the effect of this angle on adhesive retention should be investigated. Such an in-depth analysis would further substantiate the argument presented in the article.

The correlation between the change in the internal volume of the MUSAS structure and the angle of the lamellar, the number of lamellar structures, and the smoothness of the target surface warrants clarification. The authors currently only demonstrate the volume change according to the presence or absence of lamellar and the furled shape, as well as the water redistribution and adhesion data through confocal images. These data do not elucidate how the angle and number of lamellar, which are fundamental to the MUSAS structure, influence the adhesive interaction with soft tissues. Therefore, it is imperative to analyze the effect of lamellar (e.g., water drainage or capture) depending on the angle and number, and to present a specific mechanism for the resulting volume change.

A more explicit comparison of the state-of-the-art in bio-inspired underwater adhesives is needed. ([doi/10.1126/scirobotics.abm6695](https://doi.org/10.1126/scirobotics.abm6695); [doi/full/10.1126/scirobotics.aan8072](https://doi.org/10.1126/scirobotics.aan8072); doi.org/10.1088/1748-3190/ab9418; doi.org/10.1016/j.matt.2020.01.018). While this paper refers to existing technologies, a more comprehensive comparative analysis would strengthen the claim of superiority. Furthermore, a detailed discussion of the limitations of MUSAS compared to existing solutions should be included. For example, are there specific types of soft tissue or environmental conditions where MUSAS performs poorly? Addressing these limitations would provide a more balanced perspective.

Recently, the specification of preload in adhesion has become a very important issue. Simulation and confocal images suggest that preload is required for adhesion, and the authors should specify whether preload is present and its exact value when measuring adhesion.

It would be important to confirm whether the biocompatibility of the materials used in MUSAS has been thoroughly tested, particularly in the context of in vivo applications. The manuscript would benefit from a discussion of the measures taken to ensure the long-term stability and safety of the material within biological environments.

The mechanical stability of MUSAS during prolonged residence in the body seems crucial, especially in terms of potential damage or degradation. A description of the tests conducted to verify the system's resistance to such issues over extended periods would enhance the manuscript.

In drug delivery applications, ensuring the precise localization of MUSAS at target sites within the body would be essential. It would be helpful to describe any specific strategies or mechanisms employed to control the positioning of MUSAS for more effective delivery. At present, it is unclear whether controlled adhesion at precise locations is possible.

Is controllable adhesion of the device possible after a desired period of time? The process by which MUSAS detaches and eventually is expelled after a long-term residence in the GI tract seems to require further explanation. A detailed analysis of how MUSAS safely detaches from soft tissue surfaces before passing through the body would enhance the discussion. Given the mechanical properties and size of the device, is there a possibility that the device could cause damage to the small and large intestine during its expelling process after detachment from the organ?

Referee #3

(Remarks to the Author)

Inspired by the remarkable underwater adhesion of the Remora fish, Kang et al. developed the Mechanical Underwater Soft Adhesion System (MUSAS) to overcome current limitations of adhesives within these environments. Through a detailed evaluation of the physical process by which remoras adhere to soft and stiff substrates and characterization of the adhesive units of a range of remora species, the authors uncovered biological mechanisms of adhesion and then developed a bioinspired device that is able to adhere to a range of substrates underwater through a simple compressive force. Overall, this is a very useful bioinspired approach and the ability to adhere to substrates through tissue contractions (e.g. esophageal movements) make it a unique adhesive system where self-adhesion is possible and the non-invasive delivery to the tissue (e.g. endoscope) is a strength. Also, the use of mechanically interlocking adhesives such as MUSAS is important for application in a range of physiological environments (e.g. low pH of stomach), where swelling (e.g. microneedle adhesives) may be limited. Interesting examples were also provided from biosensing to therapeutics. Lastly, the investigation of such approaches potentially gives some insight into how natural selection might have driven the evolution of various features of these fish.

Despite these very positive attributes, there are some considerations that should be addressed prior to publication.

Major points:

One concern is the lack of novelty. Previous work by Wang et. al. "A biorobotic adhesive disc for underwater hitchhiking inspired by the remora suckerfish" (Science Robotics, 2017) also took a biomimetic and remora based approach to create an adhesive disc. It is suggested that the authors clearly describe the key differences and advantages of their work compared to this reported system, beyond just now applying to biological tissues.

One disadvantage to mechanical interlocking systems is that they inherently cause tissue damage, which is avoided with other tissue adhesive systems (e.g. carbopol-hydrogen bonding, EDC-NHS-covalent), and this was not avoided in this work. Due to the microscale tissue damage (300-800 μm deep 100 μm wide holes), this approach will likely be limited to use with tissues that have a high regenerative capacity and low propensity for scarring, as well as solid organs that will not leak if punctured (e.g. vasculature, lungs). This limitation should be discussed thoroughly and put into context of potential applications.

Related to this, how do the authors foresee this device being deployed in a clinical setting? There are clear examples proposed throughout the work, however, many aspects are left unclear.

-Is there any control over device detachment or ways to trigger this? Is it possible that the device will never detach? What if scarring occurs around the device?

-Once the device detaches, what does clearance of the device look like? Is it possible that the device could continue to adhere to various GI tissues as it is passed through?

-What level of damage does this device cause to the target tissues and downstream tissues as it is cleared? Histology or necrosis measurements are needed of tissues after various periods of attachment.

It is also difficult for this reviewer to compare MUSAS to alternate solutions (e.g. EDC-NHS, carbopol polymeric adhesives), where adhesion values are often reported as tensile strength and shear strength, rather than adhesive pressure and shear adhesive pressure, respectively. It would be helpful to expand on this and report adhesion values as tensile strength and shear strength so direct comparison can be made with other systems. Also, how was adhesion pressure measured for MUSAS compared to other solutions (values for Carbopol and EDC-NHS seem lower than expected)? How much pressure was applied to MUSAS during adhesion testing?

Related, it seems somewhat inappropriate to compare to polymeric adhesives. MUSAS are at a completely different scale-macro scale versus molecular. These are continuous materials that can stop bleeding, cover holes etc. this system requires a solid material to adhere.

What is failure mechanism for adhesion in tension and in shear (cohesive, adhesive, substrate failure)? This should be described and characterized with inert substrates as well as theoretical failure mechanism in vivo. This connects with previous question about how does detachment occur?

What is the major role of nitinol actuation? If the nitinol actuates at 37 Degrees, then the metal is likely already actuated once it has entered the body. Do the authors have any evidence that the actuation of nitinol occurs after the device has been deployed in vivo and what the importance of this is?

The authors use materials that will likely cause significant foreign body response (FBR) and scarring. What is the realistic timeframe where this device can be used before these unwanted features set in? As stated above, more tissue testing is needed.

Minor points:

1. Error with data on figure 3 C. Trend line data shapes are switched for nitrile and SEBS and line color not consistent.
2. ext data fig 2. image of devices with 6 or 8 is out of focus and difficult to see.

Referee #4

(Remarks to the Author)

The manuscript presents an innovative concept inspired by the remora fish, which adhere to various marine hosts using specialized suction disks. The authors investigate the anatomical and behavioral mechanisms that enable remoras to attach to soft surfaces. Based on these insights, they develop the Mechanical Underwater Soft Adhesion System (MUSAS), a mechanical device that mimics the remora's adhesion strategies.

MUSAS demonstrates strong and versatile adhesion to a variety of soft substrates with different textures, roughness, and stiffness. The authors showcase applications of MUSAS in biomedical settings, such as prolonged retention in the gastrointestinal tract for drug delivery and health monitoring, as well as environmental sensing when attached to aquatic organisms.

While the core concept of the study is intriguing and carries substantial potential, the manuscript, in its current state, is unfortunately inadequate. It is incomplete and, in several sections, hard to understand due to ambiguous writing. The presentation lacks clarity, and vital methodological details are either insufficiently explained or entirely missing, making it

difficult to evaluate the credibility of the results.

The theoretical framework underpinning MUSAS's functionality seems not sufficiently developed, and the experimental evidence presented does not convincingly support the claims made. Due to these reasons, I am unable to support its publication in Nature.

Specific points of concern:

The "Main" section of the manuscript includes several assertions that may not entirely align with the current scientific literature and insights.

- The authors posit a lack of solutions for underwater adhesion to soft substrates, stating that most existing adhesives are tailored for hard substrates with specific hydrophobicity, smoothness, and surface energy requirements. While underwater adhesion indeed poses significant challenges, recent research has made strides in creating adhesives for soft tissues, especially in biomedical applications. For example, reports exist of hydrogels and bio-inspired adhesives that demonstrate strong adhesion to wet and dynamic soft tissues through mechanisms such as interfacial toughening, covalent bonding, and physical entanglement. The authors cite references [8][9], which discuss advancements in wet tissue adhesion, but they imply these are inherently limited due to highly hydrated surfaces. However, the assertion that molecular bridges formed by covalent or hydrogen bonding are inherently limited in underwater environments may not fully cover the advancements in this field. Recent progress in mussel-inspired adhesives and catechol-functionalized polymers have demonstrated that robust bonding can be achieved even under highly hydrated conditions. Therefore, the authors seem to overstate the limitations of current chemical adhesion methods. It is crucial to recognize these advancements to accurately situate MUSAS within the context of existing technologies.
- The authors assert that MUSAS is the inaugural mechanical platform for underwater adhesion to a variety of soft substrates. However, mechanical adhesion strategies inspired by biological organisms have been previously explored. For instance, bio-inspired suction devices that mimic octopus suckers (10.1098/rsif.2017.0395) and other marine organisms (10.1242/jeb.243773) have been developed for adhesion to soft and irregular surfaces underwater. Moreover, studies have examined mechanical interlocking mechanisms for underwater adhesion to soft substrates (10.1038/s41467-017-02387-2). The authors should acknowledge previous work in mechanical adhesion systems and elucidate the unique aspects of MUSAS compared to existing technologies.
- The paragraph "Remoras Adhere to Various Soft Substrates" discusses the diversity in remora species and their unique adaptations that facilitate their adhere to various soft underwater substrates.
- The authors present data on the lamella orientations across different remora species and associate these orientations with their known host preferences. They observe that species adhering to fast-swimming hosts predominantly exhibit parallel lamellae, while those with diverse hosts display mixed lamella orientations. Although the data implies a correlation, the manuscript lacks a comprehensive quantitative analysis. It does not detail the sample sizes, statistical methods, or significance levels used to establish these correlations. This omission makes it challenging to evaluate the strength of the conclusions drawn.
- The manuscript does not cite the source of the billfish tissue stiffness.
- The authors note that remoras adhering to soft, mucus-covered surfaces like those of manta rays exhibit varied lamella orientations, which may aid adhesion to compliant substrates. While this observation is plausible, the manuscript lacks experimental data demonstrating how different lamella orientations affect adhesion on soft substrates. The assumption that varied orientations are advantageous lacks empirical validation.
- Mucus layers on marine animals are known to influence adhesion mechanics. The adhesion of remoras to mucus-covered surfaces likely involves complex interactions beyond mechanical interlocking. The manuscript does not discuss the physicochemical properties of mucus or how it affects adhesion. This omission overlooks potential biochemical adhesion mechanisms or lubrication effects that could impact remora attachment.
- In Figure 1E, the term "redundant adhesion" of remoras is used without a clear explanation. Clarification is needed to understand this concept.
- In Figure 1F, the difference between an adhesive compartment and a lamella is not explained in the Figure legend or text. • In Figures 1G-I, various substrates for testing remora adhesion—RST, STP, and PHP—are shown. However, the manuscript does not explain what these surfaces are and how they are produced or obtained. These surfaces are described in the text as "LifeLike Biotissue made from LLBT-hydrogel elastomer," again without further explanation.
- Figure 1H; the "RST alive" dataset is missing. • Figure 1H,I: The adhesion of "alive" animals is tested in only one shear direction. The differences in adhesion between "alive" and "euthanized" animals are not discussed.
- The text describes a behavior where "the remora fully unfurls its disk with multiple adhesive compartments and erects its lamellae when adhering to a soft substrate. This behavior contrasts with its method of establishing adhesion on a hard substrate through friction and sliding." However, the illustration does not clearly depict this behavior.

- In general, the text contains numerous statements that are neither sufficiently explained nor illustrated in figures; for example, “To facilitate oral administration, MUSAS can be compacted into a size 000 capsule...”
- The physiological information of the Remora adhesive disc was used to inform the development of the MUSAS. The authors use physics-based simulations to model the hydrodynamic behaviors of different adhesive disk configurations inspired by remora anatomy.
- The authors use finite element analysis (FEA) to simulate the hydrodynamic behaviors of different adhesive disk configurations when adhering to pig tissue submerged in water. They conclude that the unfurled disk with compartments holds more water and achieves a higher vacuum for suction than the other configurations. While FEA is an appropriate tool for modeling such systems, the manuscript provides limited details about the simulation parameters, material properties, boundary conditions, and validation processes. The complexity of biological adhesion, especially involving soft, compliant tissues and dynamic aquatic environments, may not be fully captured in the simulations presented.
- The authors conduct a series of experiments to optimize MUSAS, testing different soft lip thicknesses, lamella orientations, and the number of lamella rows. While the optimization process appears systematic, and the results support the conclusions about the optimal design, the manuscript lacks detailed quantitative data, such as: exact adhesion forces measured in each configuration, statistical analyses to determine the significance of differences between designs and the reproducibility of the results across multiple trials.
- The adhesion performance of MUSAS is demonstrated under controlled, static conditions in laboratory settings. However, in practical applications such as the GI tract or on living organisms like fish, the substrates are dynamic and subject to complex movements, peristalsis, variable fluid flow, and mechanical forces. The manuscript does not adequately address how MUSAS performs under such dynamic conditions.
- The use of shape memory alloys (SMA) and elastomers raises questions about biocompatibility, especially for long-term applications inside the human body. Potential issues such as immune responses, toxicity, and long-term stability of materials are not thoroughly discussed.

The applications presented, such as sustained drug release and mRNA therapeutics delivery, are promising but remain preliminary. For instance, the study demonstrates the release of Cabotegravir over a seven-day period in pigs. However, the efficacy of the drug delivery, including pharmacokinetics, biodistribution, and therapeutic outcomes, is not comprehensively evaluated.

The long-term reliability of MUSAS is not addressed in the manuscript. Potential failure modes, such as material fatigue, degradation, loss of adhesion over time, or accidental detachment, are critical considerations for practical applications, especially in biomedical devices intended for prolonged use.

Version 1:

Reviewer comments:

Referee #1

(Remarks to the Author)

The authors' diligent and comprehensive response to reviewer's concerns about surface fouling and stiffness mismatch placate this reviewer.

Referee #2

(Remarks to the Author)

The authors have performed several compelling experiments to explain the correlation between the internal volume change of the MUSAS structure and the interlayer angle, number of layer structures, and smoothness of the target surface. These additional details by the authors help to strengthen the transferability of their work.

Nevertheless, the pH-responsive reversible adhesion and bioadhesive stability of the developed adhesive material by the authors seem to be still unclear. The duration of the developed adhesive material is not constant (depending on the species and individual), and it is questionable whether the targeting to the wound site of the desired organ is possible with the pH-responsive adhesive material. Otherwise, there is concern about adverse effects if targeting is not achieved. In the case of bioadhesion stability, this part is the core of this paper, which has not yet been resolved and no clear solution has been presented. I am still negative about accepting the paper for publication in Nature.

To strengthen the physiological relevance of the mechanical and structural design, the authors should evaluate and compare adhesion strength across a broader range of surfaces than currently tested. This analysis should include additional measurements of adhesion strength on surfaces with uniform roughness but varying modulus, as well as on surfaces with uniform modulus but varying roughness. This approach will help address the current limitation in the logic, where it is unclear whether the differences in adhesion strength observed across surfaces with varying roughness and modulus are due to differences in modulus, roughness, or a combination of both.

While the authors provided valuable data on the degradation of Ecoflex over time in gastric conditions, it remains unclear how this material behavior correlates with the reported long-term retention of MUSAS in vivo. If the swelling and fouling of Ecoflex compromise vacuum-based adhesion within 7 days, further explanation is needed as to how the device maintains stable adhesion for up to 22 days in a physiological environment. Clarifying this relationship would strengthen the internal consistency of the study.

If there is debris in the stomach or small intestine, targeting adhesion of the device is unlikely to be possible. There are many foreign small matter present in the body, including food. Is the proposed device's strategic adhesion function possible in the presence of food debris present during bioactivity?

The pH-responsive targeting strategy based on Eudragit S100 is not clearly proven. The duration of the developed adhesive material is not consistent, and there are concerns about whether the pH-responsive adhesive material can perfectly target the wound site of the desired organ. Otherwise, if targeting is not achieved, there are concerns about unexpected side effects.

A scale bar should be added to some images (e.g. Fig. 3E) to convey the information clearly.

Referee #3

(Remarks to the Author)

The authors have done a very nice job in the revision. These devices hold the greatest promise for GI tract applications, as illustrated, whereas other applications are probably better with alternate adhesive systems that do not cause tissue damage.

The major points of my prior review have been addressed:

1. There are now clearly defined improvements and innovation from previous work. The table is useful to distinguish.
2. There is now good reasoning on how the damage (holes cut into tissue) is used to their advantage for mucosal tissue drug delivery where biochemical adhesion (hydrogen bonding, NHS covalent bonding) methods fail because of the rapid turnover of mucosal tissue.
3. There is now good motivation for a tunable detachment system, with tailored behaviors based on lamella degradation, and a clear reasoning on why they can be safely cleared without reattachment.
4. Histological analysis of attachment site and downstream tissues and evaluation by a pathologist showed no significant tissue damage or scarring. This further confirms safe passage of MUSAS after detachment.
5. There is further clarification of the adhesion values and adhesive failure mechanisms.

Referee #4

(Remarks to the Author)

The core of this work, along with the scope and diversity of the research conducted, remain innovative and impressive. However, while the authors have addressed many of the concerns raised in their rebuttal letter, these responses were unfortunately only been partially and insufficiently integrated into the revised manuscript. This leaves several key issues unresolved, which significantly impacts the clarity and overall impact of the text.

For instance, the term "redundant adhesion" appears in both the text and the figures, but no clear explanation is provided to define what it means. Although the authors attempted to address this in their rebuttal and revised manuscript by stating, "Moreover, we determined that the remora can achieve redundant adhesion without needing its adhesive disk to entirely cover the soft substrate, as long as a sufficient number of individual adhesive compartments are engaged to ensure robust attachment..." this explanation does not adequately support understanding. It remains unclear on what data this statement is based, and from the videos and figures provided, it is not possible to deduce how individual adhesive compartments mediate adhesion. Furthermore, it is not evident what additional value Figure 1E brings to Figure 1F, or what exactly is meant to be illustrated by the label "redundant adhesion".

As this is just one example of many where the reader is unnecessarily burdened by insufficient or confusing explanations, I still find the manuscript hindered by a pervasive lack of clarity and coherence. This makes it difficult for readers to identify the primary research questions, understand the methodologies employed, and follow the logical progression of the study. The excessive use of technical jargon without proper explanation further obscures the significance of the findings, leaving the reader struggling to grasp the intended contributions of the research. For example, when explaining the "Stomach Tissue Phantom" surfaces, it is mentioned that they are made from "LLBT-hydrogel-elastomer". But what is LLBT? I presume it refers to "LifeLike Biotissue", which is mentioned earlier in the sentence, but the abbreviation is not explicitly introduced. This is just one of many instances where the manuscript falls short of the general standards for scientific writing, let alone the rigorous criteria expected for publication in Nature.

The presentation of data and results is particularly problematic. Ambiguous figure descriptions and inconsistent explanations obscure the connection between experimental details and the conclusions drawn, resulting in a series of disjointed observations rather than a cohesive, streamlined argument. For example, Figure 4 is so overcrowded that even at the highest magnification on a screen, some of the graphs and images are difficult to discern and interpret. In my opinion, it would be far better to reduce the amount of data in this figure (a recommendation that applies to most figures in the main text) and move some of it to the supplementary materials.

In summary, the insufficient incorporation of key responses from the review process, combined with issues in clarity, coherence, and data presentation, significantly undermines the ability of the manuscript to communicate transformative scientific insights. From my perspective, the entire presentation of the study must be fundamentally rethought and thoroughly revised.

Version 2:

Reviewer comments:

Referee #2

(Remarks to the Author)

The authors provided additional experimental data and explanations that contributed to the study. The authors improved the experimental validation of the adhesion mechanism by including a systematic separation of the effects of surface modulus and roughness.

However, the breadth of the application sections can be confusing to the reader in Figure 5. While each example independently adds value, presenting multiple use cases at different levels of validation and translational maturity simultaneously tends to obscure the core message. I would recommend that the main text focus on the most compelling and validated applications, such as gastrointestinal retention and controlled release in a porcine model, with more exploratory demonstrations moving to the supplemental information. This reorganization would enhance the consistency of the manuscript and clearly and effectively communicate the key innovations.

As the authors noted in their response, I think it would be helpful for the reader to clearly explain the limitations of their device (precision adhesive targeting, adhesion issues in organs with foreign bodies, swelling and contamination issues in the case of Ecoflex, etc.) in the manuscript.

Referee #4

(Remarks to the Author)

The authors effectively addressed the reviewers' comments and suggestions, which greatly benefited the comprehensibility of the manuscript. I can thus in principle support publication of the manuscript.

However, I would recommend to improve the following points in a final revision:

- The legends of Figure 1 and Figure 2 should be more instructive to communicate the rather complex figure content.
- Image description and the corresponding text of Figure 1E seem not clear enough. It may be helpful to reference a supplementary video to illustrate the described process.
- For Figures 1H and 1I, no statistical tests are reported, in contrast to the subsequent figures. Was this intentional or an oversight?
- There is no reference to Figure 2a in the main text.
- The complex multi-part Figure 2F is described with only a single sentence in both the main text and the legend.
- The use of abbreviations should be made more consistent:
 - While "gastroesophageal reflux disease (GERD)" is properly introduced in the abstract, "gastrointestinal" is not abbreviated, even though "GI" appears in the first paragraph of the main text without prior definition.
 - In the section "Remoras adhere to various soft substrates," abbreviations for different remora species are introduced, but then not used in subsequent sentences.
 - Additionally, the formatting of species abbreviations is inconsistent: most are introduced with a capital letter and capitalized abbreviation, but *Remora remora* is abbreviated as "R. rem." rather than following the same convention.
- At the beginning of the section "In vitro and ex vivo characterization of optimal MUSAS," the phrase "optimal MUSAS" is used without prior definition. The criteria by which this optimal design was determined should be specified.
- The Materials and Methods section lacks information on the manufacturing process of MUSAS. This should be explained if intentional, or the methodological details should be provided.

Point-by-point response – Nature – Manuscript #: 2024-09-19511

Referee #1 (Remarks to the Author):

The manuscript (2024-09-19511) by Traverso's group on the mechanism and mimetics of remora attachment to different marine hosts is impressively exemplary in several respects: 1) the diversity of anatomical designs, 2) the anatomical details of lamellar organization, 3) the identification of essential features for adhesion, 4) the mechanical testing of natural and mimetic discs to specified surfaces, and 5) the exploration of specific need-driven applications. The work is clearly articulated and presented. Results are appropriately tested with replicates/controls.

Comment 1A: To this reviewer's eye, however, there may be two less exemplary areas that pertain to the validity of approach: A) the authors seem to be persuaded that the biochemistry and natural supramolecular organization are irrelevant to better understanding and mimicking/translating this system. This resembles the drift of earlier efforts on gecko adhesion with respect to setal structure in the 10 -1000 nm length scale range. This blind-eye to biochemistry has impeded an important reckoning of how to keep setae from fouling with ageing/use among other issues. This could easily be an issue here as well.

Response 1A: We appreciate the reviewer's recognition of our work as impressively exemplary. Biochemical setal structures within the 10–1000 nm length scale may contribute to adhesion and failure modes through intermolecular interactions, particularly in relation to fouling resistance. While this study focuses on mechanical adhesion, we conducted additional *in vitro* studies to evaluate the fouling and failure mode of MUSAS incubated at body temperature (37°C) in gastric fluid harvested from Yorkshire pigs. Microscope and scanning electron microscope (SEM) imaging confirmed that fouling remained negligible at the micron scale for shape memory alloy (SMA) lamellae after 7 days of incubation (Fig. R1-1B & R1-1D). After 21 days, minor fouling was observed at the 50 µm scale, with minimal changes in SMA surface morphology at the 5 µm scale, indicating negligible interference with mechanical interlocking (Fig. R1-1D). However, surface morphology reconstruction of silicone rubber occurred over time due to the reported water permeability of Ecoflex (260 g/m²day⁻¹ [1A-1]). Specifically, swelling and fouling of the silicone rubber's crease pattern appeared after 7 days and became more evident after 21 days, suggesting a gradual loss of friction, which is crucial for maintaining vacuum-based mechanical adhesion in MUSAS (Fig. R1-1C). Additionally, changes in silicone rubber morphology were observed within the 10–1000 nm length scale (Fig. R1-1E). Initially, Ecoflex appeared smooth with a few detectable setal structures at high magnification (500 nm); however, fouling and surface roughening gradually developed after 7 days, and became evident after 21 days. This degradation could weaken intermolecular interactions, eventually leading to leakage and failure of the vacuum-based seal. These results suggest that the primary failure mode of MUSAS will be caused by the loss of vacuum-based mechanical adhesion due to water exchange, swelling and fouling of the silicone rubber, occurring at 500 nm to 50 µm. Future studies could explore improved waterproof silicone rubber and finer setal structures within the 10–1000 nm range to enhance the longevity and functionality of MUSAS. A discussion of these findings has been added to the main manuscript and supplementary information.

References 1A

- [1] Shao, Y., Yan, S., Li, J., Silva-Pedraza, Z., Zhou, T., Hsieh, M., Liu, B., Li, T., Gu, L., Zhao, Y. and Dong, Y., 2022. Stretchable encapsulation materials with high dynamic water resistivity and tissue-matching elasticity. *ACS applied materials & interfaces*, 14(16), pp.18935-18943.

Fig. R1-1|In vitro characterization of fouling of MUSAS. A. Illustration of the material components of MUSAS. B. Microscope pictures of shape memory alloy (SMA) lamellae incubated in swine gastric fluid at 37 °C for different period of time (scale bar: 1 mm). C. Microscope pictures of silicone rubber (Ecoflex 0030) incubated swine gastric fluid at 37 °C for different period of time (scale bar: 1 mm). D. SEM images characterizing surface morphology of SMA incubated for different period of time at micron scale. E. SEM images characterizing surface morphology of silicone rubber (Ecoflex 0030) incubated for different period of time at micron and nanometer scales.

Comment 1B: The second area has to do with lamellar architecture in which aligned bayonet-like spinules appear to be embedded in softer lip-like flaps. There is no mention (or did I miss it?) of the stiffness mismatch between the spinules and the flaps. If spinules resemble the modulus of martensite, how do lamellae circumvent extensive contact damage at the spinule/tissue interface?

Response 1B: We appreciate the reviewer's concern regarding the stiffness mismatch between the spinules (shape memory alloys) and flaps (silicone rubber) potentially causing contact damage. To address this, we conducted additional simulations to assess contact damage and durability of the lamella-flap structures upon contact with human stomach tissue. Using Abaqus 2021 (SIMULIA), we applied a fracture mechanics model to evaluate the contact damage of tissue piercing. In accordance with previous simulations, the silicone rubber flap was modeled as EcoFlex 0030, with a density of 1.07 g/cm³, a Young's modulus of 125 kPa, and a Poisson's ratio of 0.49 [1B-1]. Stomach tissue properties included a density of 1.088 g/cm³, a Young's modulus of 700 kPa, and a Poisson's ratio of 0.49 [1B-2], and a plastic yield stress of 700 kPa [1B-3]. The SMA lamella was assumed to remain in the martensite phase during contact, with a Young's modulus of 23 GPa, a Poisson's ratio of 0.33, and a density of 6.5 g/cm³ [1B-4]. We applied these to a single lamella-flap structure interacting with stomach tissue. To comprehensively evaluate contact damage, we simulated an extreme case of instant insertion where the flap-lamella structure was inserted into stomach tissue at a vertical tip speed of 2 cm/s to a depth of 1.4 mm. Notably, the silicone rubber composed of the flap is hyperelastic and can withstand strains up to 900% [1B-5]. The simulation confirmed that stress at material interfaces with mismatched stiffness was effectively dissipated through the hyperelastic deformation of the silicone rubber, minimizing stress concentration at the lamella tip during insertion (Fig. R1-2A & R1-2B). We also conducted a fatigue analysis to assess the durability of the flap-lamella structure, under a cyclic condition of instant insertion described above. A plug-in algorithm for Nitinol in fe-safe, coupled with Abaqus, was used to model the fatigue behavior of SMAs. Additionally, the Endurica plug-in of fe-safe was used to model the fatigue behavior of silicone rubber [1B-6]. Fatigue analysis identified the critical area as the thinnest section of the silicone flap, where it wraps around the lamella, measuring only 100 μm in thickness. However, this area could withstand at least 86 cycles (10^{1.94}) before reaching 75% lifespan of its maximum durability, under extreme instant insertion contact (Fig. R1-2C). These results confirm that the stiffness mismatch between the lamella and soft flap has a minor impact on the durability of MUSAS, especially considering its infrequent and slow tissue contact in real-world applications. Relevant results and discussion were also added to extension figure sets and supplementary information.

References 1B

- [1] Heikenfeld, J., Jajack, A., Rogers, J., Gutruf, P., Tian, L., Pan, T., Li, R., Khine, M., Kim, J. and Wang, J., 2018. Wearable sensors: modalities, challenges, and prospects. *Lab on a Chip*, 18(2), pp.217-248.
- [2] Guimarães, C.F., Gasperini, L., Marques, A.P. and Reis, R.L., 2020. The stiffness of living tissues and its implications for tissue engineering. *Nature Reviews Materials*, 5(5), pp.351-370.
- [3] Egorov, V.I., Schastlivtsev, I.V., Prut, E.V., Baranov, A.O. and Turusov, R.A., 2002. Mechanical properties of the human gastrointestinal tract. *Journal of biomechanics*, 35(10), pp.1417-1425.

- [4] Lagoudas, Dimitris C. "Shape memory alloys." Science and Business Media, LLC (2008).
 [5] <https://www.smooth-on.com/products/ecoflex-00-30/>
 [6] <https://endurica.com/wp-content/uploads/2023/11/Silicone-Application-Spotlight-2023-10.pdf>

Fig. R1-2] Contact and durability analysis of the lamella-flap structure. A. Representative deformation and stress distribution when the lamella-flap structure first contacts the stomach tissue at a tip piercing speed of -2 cm/s in the y-direction. B. Representative deformation and stress distribution when the lamella-flap structure fully engages with the stomach tissue, reaching an insertion depth of 1.4 mm at a tip piercing speed of -2 cm/s in the y-direction. C. Fatigue analysis assessing the durability of the lamella-flap structure, demonstrating an average lifespan of 75% of its log10 cycle repeats under a cyclic contact procedure transitioning from A to B.

Referee #2 (Remarks to the Author):

Comment 2A: In this study, the authors developed a Mechanical Underwater Soft Adhesion System (MUSAS) inspired by remoras, which utilize sucker-like adhesive disks to adhere to various substrates. The authors analyzed and summarized in detail the angles and distributions of lamellar structures in various remora species for adhesion processes and contact targets. Based on this analysis, the MUSAS system addresses the challenges of underwater adhesion to soft substrates by mimicking the anatomical and behavioral mechanisms of remoras. The authors demonstrated that MUSAS exhibits versatility, capable of adhering to soft substrates with varying surface textures and environmental conditions, achieving a high adhesion-force-to-weight ratio. The platform is insensitive to environmental pH and humidity, demonstrating potential applications in a variety of settings, including noninvasive gastrointestinal drug delivery and environmental biosensing.

However, this paper requires rigorous evaluation of novelty, validation of claims, and areas of potential improvement compared to previous studies. Although the authors have developed a material optimized for soft surfaces and proposed applications through studies on the angles of the remora of the suction cup structure, the underwater adhesion concept and some application technologies of the lamellar-based sucker-mimicking adhesive material have been reported in previous studies ([doi/10.1126/scirobotics.abm6695](https://doi.org/10.1126/scirobotics.abm6695); [doi/full/10.1126/scirobotics.aan8072](https://doi.org/10.1126/scirobotics.aan8072); doi.org/10.1088/1748-3190/ab9418; doi.org/10.1016/j.matt.2020.01.018), so it is difficult to judge its novelty as the best. In particular, adhesion to soft surfaces has already been demonstrated depending on the number of lamellar structures of the artificial sucker fish ([doi/10.1126/scirobotics.abm6695](https://doi.org/10.1126/scirobotics.abm6695)). Although the angle of the proposed lamella structure of the MUSAS is described as being deeply related to the adhesive interaction with soft or hard tissues, an in-depth methodological analysis of this relationship has not been presented. Additionally, an insightful analysis of the activation process of the suction-based adhesive interaction and the resulting mechanism of adhesion change is lacking. The correlation between the change in the internal volume of the MUSAS structure and the lamellar angle, the number of lamellar structures, and the smoothness of the target surface should be elucidated.

The authors seem to have only summarized the concept (without sufficient verification data) of various series of application technologies using their adhesive material. Rather, it seems more powerful to thoroughly analyze and verify one application. From an application perspective, an in-depth elucidation of specific mechanisms, as well as thorough verification of biocompatibility and stability with controllable adhesion capability, are essential to emphasize its distinctive advantages over other bioadhesive studies in their applications into ingestible devices.

Overall, considering the quality standards of Nature, this manuscript would be more suitable for other specialized journals. For additional questions, please see below.

Response 2A: We sincerely appreciate the reviewer's constructive feedback and have carefully revised our manuscript to address each point raised. Before detailing these revisions, we would like to provide a comprehensive overview of the novelty and major advancements introduced by MUSAS in addressing critical challenges across multiple fields.

This study uniquely explores previously unexamined adhesion mechanisms of remoras on soft substrates, integrating the anatomical evolution of the adhesive disk across all remora species, which has not been systematically analyzed before. The subsequent development of MUSAS

enables *in situ*, long-term retention adhesion on dynamic, morphable soft substrates that undergo rapid turnover and regeneration in extreme pH and moisture environments (e.g., the stomach). This presents significant challenges for current adhesive solutions, particularly in the context of mucoadhesion and biomedical applications.

Our *in vitro*, *ex vivo*, and *in vivo* validations push the boundaries of technical feasibility and research capabilities, supported by comprehensive, multimodal cross-validation in diverse application scenarios. Furthermore, we have developed a range of MUSAS-based applications, each with independent novelty, being introduced to the scientific community to tackle long-standing, immediate challenges. These innovations represent significant advances in bioadhesion, biomedical treatments, and biological exploration.

The novelty of this work can be summarized in three key levels:

1. *Novelty in Remora Adhesion Studies* - New insights into remora adhesion on dynamic, morphable soft substrates and the evolution of the adhesive disk anatomy

We acknowledge the contributions of prior remora studies from Dr. Li Wen and Dr. Robert Wood's group [2A-1] and subsequent research [2A-2 - 2A-4]. However, previous studies focused on a single remora species (*Echeneis naucrates*) adhering to static, hard surfaces (e.g., glass). These studies primarily concluded that remora adhesion is facilitated by lamella erection and disk muscle stiffness.

In contrast, our study:

- Conducts a comprehensive review of all described remora species to examine anatomical variations and evolutionary adaptations in adhesive disk morphology.
- Explores adhesion on soft, dynamic substrates—a previously unaddressed challenge in bioadhesion.
- Investigates live remora adhesion on a stomach tissue phantom, uncovering new adhesion mechanisms based on disk unfurling behavior and lamellae orientation, which play a dominant role in soft adhesion.
- Proposes a novel evolutionary explanation for remora adhesive anatomy adaptations across different species.

This expands bioadhesion knowledge and provides new insights into biomimetic adhesion design, distinct from previous studies.

2. *Novelty in Remora-Inspired Adhesion Strategies* - Motor-Free Self-Adhesion vs. Motor-Actuated Robotics

Our study addresses the fundamental challenge of long-term adhesion on dynamic, morphable soft substrates that experience rapid regeneration and extreme conditions (e.g., the gastrointestinal (GI) tract). This remains an unresolved issue in bioadhesion, especially mucoadhesion research.

For example, gastric epithelial cells renew every ~3 days, presenting a significant barrier to long-term retention. Despite decades of research [2A-5], most current adhesion solutions fail *in vivo* unless external stimuli (e.g., electric fields) significantly modify surface anatomy [2A-6].

Key distinctions from prior remora-inspired studies:

- Previous studies [2A-1 - 2A-4] focused on motor-actuated underwater robots for hitchhiking functionalities, without addressing fundamental challenges of long-term adhesion to soft, morphable tissues.
- These studies used pneumatic, motor-actuated lamella erection (>10 cm devices) to optimize pitching angles for adhesion, which is impractical for biomedical applications.
- Our study focuses on developing a miniaturized, self-adhesive system, avoiding the need for motors, external actuation, or complex maneuvering.

Our design strategy:

We introduce a motor-free self-adhesion solution, based on:

- 1) Miniaturization – Ingestible device size (< 25 mm × 9.5 mm width).
- 2) High drug-loading efficiency – Unlike motor-actuated systems, our approach preserves the active pharmaceutical ingredient (API) payload.
- 3) Simplified, user-friendly deployment – No need for new equipment or additional training.

By leveraging disk unfurling behavior and lamellae orientation, MUSAS achieves stable adhesion with minimum preload and maximizes long-term retention in real-world dynamic environments.

Furthermore, prior studies cannot be directly applied to soft substrate adhesion because:

- Shear frictional performance on soft substrates is not sensitive to lamella contact (pitching) angles but spinule lengths (Supp. Fig. S6), unlike hard surfaces.
- Previous experimental setups lacked variety—tested materials were stiff (Elastosil M4601: 6.5 MPa Young’s Modulus) or smooth (Ecoflex: 0.05 - 6 μm roughness according) [2A-8, 2A-9].
- Soft substrates (e.g., mucosal layers) exhibit dynamic sliding and surface morphing, requiring completely different adhesion strategies (Supp. Fig. S5).

Our study systematically addresses these limitations through comprehensive validations.

3. Major Advances in Bioelectronics, Biomedical Treatments, and Biological Exploration

MUSAS represents a significant translational advancement, tackling critical biomedical and biological challenges.

Unlike studies that propose conceptual applications, each MUSAS-enabled application underwent rigorous, independent *in vitro*, *ex vivo*, and *in vivo* validation, employing gold-standard animal models, particularly the swine model, which offers advance translational relevance due to its close anatomical resemblance to humans [2A-10].

Key applications and major advancements:

- 1) Marine Life Exploration – Developed an ultraminiaturized underwater, battery-free, wireless temperature sensor for real-time monitoring of swimming organisms in near-natural conditions.

- 2) Noninvasive Monitoring of Gastroesophageal Reflux Disease (GERD) – Developed a pH-sensitive impedance sensor, demonstrating an unprecedented alternative adhesion-based esophageal health-monitoring system.
- 3) Sustained Drug Release for HIV/AIDS PrEP – Successfully developed a MUSAS-enabled slow-release Cabotegravir formulation, demonstrating >1 week retention *in vivo* (validated in 9 pigs, with pharmacokinetic studies detailed in Fig. 4).
- 4) Effective mRNA Delivery in the GI Tract – Developed an unprecedented device-enabled mRNA delivery system (215 μ L capacity), demonstrating functional luciferase mRNA transfection in large animals.
 - Only two previous studies [2A-11, 2A-12] reported mRNA detection in large animals, but without confirming functional transfection.
 - Our study achieved functional protein expression, confirmed through bioluminescence imaging without requiring prohibitively expensive systemic luciferin administration.
 - This major advancement in demonstration of mRNA transfection in the GI tract, overcame long-standing biological barriers, with profound implications for vaccination and genetic therapeutics.

We respectfully request recognition of the significant advances enabled by MUSAS. This work extends beyond conceptual applications—each claim is rigorously validated through extensive animal studies (e.g., >55 pigs, >8 fish over three years). These findings open new frontiers in bioadhesion, biomedical treatments, and biological exploration, marking a transformative advancement in translational science.

We have incorporated detailed explanations and additional figures (e.g., Fig. 4, Ext. Fig. 2) to explicitly address the reviewer’s suggestions.

References 2A

- [1] Wang, Y., Yang, X., Chen, Y., Wainwright, D. K., Kenaley, C. P., Gong, Z., ... & Wen, L. (2017). A biorobotic adhesive disc for underwater hitchhiking inspired by the remora suckerfish. *Science Robotics*, 2(10), eaan8072.
- [2] Li, L., Wang, S., Zhang, Y., Song, S., Wang, C., Tan, S., ... & Wen, L. (2022). Aerial-aquatic robots capable of crossing the air-water boundary and hitchhiking on surfaces. *Science robotics*, 7(66), eabm6695.
- [3] Wang, S., Li, L., Sun, W., Wainwright, D., Wang, H., Zhao, W., Chen, B., Chen, Y. and Wen, L., 2020. Detachment of the remora suckerfish disc: kinematics and a bio-inspired robotic model. *Bioinspiration & Biomimetics*, 15(5), p.056018.
- [4] Su, S., Wang, S., Li, L., Xie, Z., Hao, F., Xu, J., Wang, S., Guan, J. and Wen, L., 2020. Vertical fibrous morphology and structure-function relationship in natural and biomimetic suction-based adhesion discs. *Matter*, 2(5), pp.1207-1221.
- [5] Nan, K., Feig, V.R., Ying, B., Howarth, J.G., Kang, Z., Yang, Y. and Traverso, G., 2022. Mucosa-interfacing electronics. *Nature Reviews Materials*, 7(11), pp.908-925.
- [6] Ying, B., Nan, K., Zhu, Q., Khuu, T., Ro, H., Qin, S., ... & Traverso, G. (2023). Electroadhesive hydrogel interface for prolonged mucosal theranostics. *bioRxiv*, 2023-12.
- [7] Huie, Jonathan M., and Adam P. Summers. "The effects of soft and rough substrates on suction-based adhesion." *Journal of Experimental Biology* 225.9 (2022): jeb243773.

- [8] Elastosil M4601 <https://www.wacker.com/h/en-us/silicone-rubber/room-temperature-curing-silicone-rubber-rtv-2/elastosil-m-4601-ab/p/000018458>
- [9] Althumayri, Majed Othman, Azra Yaprak Tarman, and Hatice Ceylan Koydemir. "Bioinspired skin-like *in vitro* model for investigating catheter-related bloodstream infections." *Scientific Reports* 14.1 (2024): 26167.
- [10] Lunney, J.K., Van Goor, A., Walker, K.E., Hailstock, T., Franklin, J. and Dai, C., 2021. Importance of the pig as a human biomedical model. *Science translational medicine*, 13(621), p.eabd5758.
- [11] Abramson, Alex, et al. "Oral mRNA delivery using capsule-mediated gastrointestinal tissue injections." *Matter* 5.3 (2022): 975-987.
- [12] Schultz, D., Kempen, P. J., Primdahl, S., Pereverzina, M., Uhrenfeldt, A. H., Alba, E. M., ... & Urquhart, A. J. (2024). Gastrointestinal device-mediated delivery of mRNA-lipid nanoparticles achieves distinct expression and biodistribution in mice and pigs. *ACS Applied Materials & Interfaces*, 16(49), 67192-67202.

Comment 2B: To further strengthen the soft tissue interaction aspect claimed by the authors, a comprehensive analysis of the adhesive interaction with soft or hard tissues according to the angle of the proposed lamellar structure of MUSAS should be incorporated. For instance, it is pertinent to examine how the adhesive interaction with various modulus samples varies according to the angle of the lamellar structure. Additionally, the effect of this angle on adhesive retention should be investigated. Such an in-depth analysis would further substantiate the argument presented in the article.

Response 2B: We appreciate the constructive feedback from the reviewer, highlighting the importance of understanding the impact of lamella orientation on the adhesion of soft substrates with varying stiffness, as well as on retention performance. To address these questions, we designed a series of comprehensive experiments to evaluate the impact of the angle of lamella orientation on adhesion to soft substrates with not only different stiffness but also different roughness. Specifically, we aim to understand whether a particular lamella orientation angle contributes to adhesion. Hence, we added five designs of lamella orientation fixed with a particular tilted angle, considering different numbers of rows of lamellae, including 30-degree dominated orientation with 4 rows of lamellae, 30-degree dominated orientation with 6 rows of lamellae, 30-degree dominated orientation with 8 rows of lamellae, 15-degree dominated orientation with 8 rows of lamellae, and 45-degree dominated orientation with 8 rows of lamellae, comparing with the tilted-dominant orientation design (0-15-30-45-degree orientation with 8 rows of lamella). We then assessed the normal and shear adhesion strength of the aforementioned designs, on bis-tris gel (stiffness 9 kPa, areal average roughness S_a 2.592 μm), porous tough hydrogel (stiffness 26 kPa, S_a 19.118 μm), and SEBS (stiffness 19 MPa, S_a 27.437 μm), shown in Fig. R2-1. While the tilted-dominant orientation design was confirmed to outperform the other configurations, a clear trend emerged showing that adhesion on rough surfaces is weaker than on smooth surfaces. This trend is particularly evident in adhesion performance on soft substrates with similar stiffness but significantly different roughness (bis-tris gel vs. porous tough hydrogel). It is worth noting that in rare cases, although the trend is preserved across all designs, the statistics were nonsignificant. This is due to measurement disparity caused by the nonuniformity and dynamic morphing of the gels, especially considering

that their nonuniform fragility and porosity. Additionally, adhesion on stiffer substrates is clearly stronger than on softer substrates with similar roughness (porous tough hydrogel vs. SEBS), due to less dynamic morphing of the stiffer substrates, which helps preserve the adhesion seal of the devices. Interestingly, we also observed shear sliding behaviors on all 15-, 30-, and 45-degree dominated orientations (Supp. Fig. S7), similar to what we previously observed in the parallel design (Ext. Fig. 2). This indicates that improved adhesion performance on soft substrates does not stem from a particular angle of lamella orientation but rather from the variation in lamella orientation. All these findings reinforce our previous conclusion that a smaller number of rows significantly weakens adhesion performance due to the lack of independent co-adhesion resulting from a limited number of individual compartments (Fig. 2J). A comprehensive analysis investigating the performance variations across different designs can be found in the response to the next comment.

We further conducted *in vitro* dynamic inference studies to compare the retention performance of the 30-degree dominated, 8-rows design with the tilt-dominated, 8-row designs, which are the top two best performers in the mechanical test. Specifically, different designs were adhered to stomach tissue phantoms freely floating in simulated gastric fluid (pH = 1.5), shaken in a 37°C incubator (Fig. R2-2A – Fig. R2-2C, Supp. Video S6). The results once again confirm the tilt-dominated design as an outperformer, such that it can maintain on the tissue phantom for more than 12 days.

Fig. R2-1|Understanding the optimal angle of lamella orientation. A and B. Adhesion performance of MUSAS with varying lamella orientation angles and row configurations on soft substrates of different stiffness and roughness (n = 5).

Fig. R2-2| Representative *in vitro* and *in vivo* evaluation of adhesion and retention performance of MUSAS. A. Representative retention performance of optimal tilt-dominated MUSAS under dynamic shaking interference in a 37°C incubator (New Brunswick Innova 40/40R, see Supp. Video S6 for further demonstration). B. Illustration of the retention failure mode. C. Dynamic interference evaluation of MUSAS retention performance with different angles of lamella orientation.

Comment 2C: The correlation between the change in the internal volume of the MUSAS structure and the angle of the lamellar, the number of lamellar structures, and the smoothness of

the target surface warrants clarification. The authors currently only demonstrate the volume change according to the presence or absence of lamellar and the furled shape, as well as the water redistribution and adhesion data through confocal images. These data do not elucidate how the angle and number of lamellar, which are fundamental to the MUSAS structure, influence the adhesive interaction with soft tissues. Therefore, it is imperative to analyze the effect of lamellar (e.g., water drainage or capture) depending on the angle and number, and to present a specific mechanism for the resulting volume change.

Response 2C: We appreciate the constructive feedback from the reviewer. The focus of this study on adhesion to soft substrates introduced an additional layer of complexity in experimentally characterizing the volume change of the devices compared to previous suction-based adhesion studies, since the most commonly adopted approaches, such as frustrated total internal reflection (FTIR) imaging, require fully transparent hard substrates (glass), for surface analysis. To overcome this obstacle, we designed a new approach by combining micro-CT and particle image velocimetry (PIV) to comprehensively study the volume change of MUSAS with different numbers of rows and lamella orientations adhering to soft substrates with various stiffness and roughness.

Specifically, we imaged the internal volume of different designs adhered to bis-tris gel (stiffness: 9 kPa, areal average roughness Sa: 2.592 μm), porous tough hydrogel (stiffness: 26 kPa, Sa: 19.118 μm), and pig stomach tissue (stiffness: 76 kPa, Sa: 85.164 μm) in an underwater environment through micro-CT (Fig. R2-3A). To comprehensively evaluate the internal volume change and avoid measurement bias, we calculated both the relative vacuum ratio and the relative compression ratio (Fig. R2-3C& R2-3D). In accordance with the mechanical testing results (Fig. R2-1), the tilt-dominant design outperformed all the others, and an increased number of rows consistently enhanced independent compartmental sealing, reinforcing our previous findings. Notably, a clear trend was once again confirmed: adhesion on rougher surfaces was always weaker than on smooth surfaces (bis-tris gel vs. porous tough hydrogel). Additionally, adhesion on stiffer substrates was stronger than on softer substrates with similar roughness (porous tough hydrogel vs. pig stomach tissue). Micro-CT analysis elucidated that the superior performance of the tilt-dominant design with variational lamella orientation resulted from well-rounded mechanical interlocking in all directions, extending below the contact surface to secure suction-based adhesion. This adaptation conformed to the dynamic, non-homogeneous morphing of the substrates, with the bis-tris gel and stomach tissue filling the individual adhesive compartments (Fig. R2-3B & R2-3E). Furthermore, micro-CT images explained why the 30 deg-dominated design often outperformed the 15 deg-dominated and 45 deg-dominated designs in mechanical testing (Fig. R2-3C& R2-3D), such that the 30 deg-dominated design demonstrated better interlocking than the others, measured by the relative vacuum ratio and the relative compression ratio, yet still lacked sufficient angular variation comparing to the tilt-dominant design (Fig. R2-3E). Note that the relative vacuum ratio measurement for the 45 deg-dominated design may be biased, as the real value could be lower due to non-vacuum space concealed by its highly tilted lamella orientation (Fig. R2-3E), which necessitates evaluation of both the relative vacuum ratio and the relative compression ratio.

We further conducted PIV tests, collaborating with a renowned fluid dynamics expert Prof. Alexandra H. Techet, to understand the hydrodynamic performance of different designs. Specifically, we selected representative designs evaluated through micro-CT testing, along with the benchmark parallel-angled design, to assess their water expulsion performance at a constant

sinking speed of approximately 30 mm/min for adhesion to a bottom substrate (Fig. R2-4). Bottom-view instantaneous velocity fields reveal that designs with fewer lamella rows exhibit minimal water expulsion, as observed in the 4-row configuration (Fig. R2-4A). This explains their inadequate internal volume change during adhesion, observed in the mechanical and micro-CT testing. As the number of lamella rows increases, unsubstantial angle tilting tends to facilitate unidirectional water expulsion, such that the 15-degree-dominated design performing nearly identically to the parallel-angled configuration (Fig. R2-4B & R2-4C). In contrast, designs with significant angle tilting induce multidirectional water expulsion, as evidenced by the strong local convergence and divergence of flow vectors, indicating out of plane (3D) flow (Fig. R2-4D - R2-4F). However, water expulsion in designs dominated by 30-degree and 45-degree angles is often uneven and accompanied by significant air bubble formation in specific regions (Figs. R2-4D & R2-4E). Notably from micro-CT testing (Fig. R2-3E), lamellae of 45-degree-dominated design are too tilted to establish enough mechanical interlocking, which compromises its water expulsion advantage. Among the tested configurations, the tilt-dominant design with substantial lamella angle variation, demonstrates superior efficiency in achieving homogeneous, multidirectional water expulsion (Fig. R2-4F). Combined with its well-rounded, omnidirectional mechanical interlocking, this design further confirms that improved adhesion performance is driven by variations in lamella orientation angles rather than a strong preference for dominance of a specific lamella orientation angle.

The aforementioned discussion and results have been also added to the manuscript, methods section, extended figures, and supplementary information accordingly.

Fig. R2-3 μ -CT imaging of internal volume changes in MUSAS with varying angles of lamella orientation and row numbers, adhering to soft substrates with distinct stiffness and roughness. **A**. Setup of μ CT imaging (scale bar: left, 1 cm; right, 5 mm). **B**. Representative image of suction marks left by MUSAS after adhesion to pig stomach tissue, demonstrating non-homogeneous substrate morphing. **C** and **D**. Relative vacuum ratio and relative compression ratio of different MUSAS designs adhering to representative soft substrates with distinct stiffness and roughness. **E**. Representative μ CT imaging of MUSAS with varying lamella orientation angles and row numbers, adhering to soft substrates with distinct stiffness and roughness. The red box represents the non-vacuum area $A_{Initial}$ or $A_{Adhesion}$. (scale bar: 5 mm).

Fig. R2-4| Particle imaging velocimetry (PIV) assessment of MUSAS hydrodynamics with varying angle of lamella orientations and row numbers, demonstrated in instantaneous velocity fields. A. Bottom-view fluid velocity maps reveal that fewer lamella rows result in minimal water expulsion, as seen in the 4 rows design. **B and C.** In terms of 8 rows design, unsubstantial angle variations incline to unidirectional water expulsion, making the performance of the 15 degree dominated design (C) nearly identical to that of the parallel-angled configuration (B). **D and E.** While lamella orientations dominated by 30 degree (D) and 45 degree (E) induce multidirectional water expulsion, the flow is often uneven and accompanied by large air bubble formation in specific regions. **F.** Notably, the tilt-dominant design with the substantial lamella angle variation exhibits superior efficiency in achieving homogeneous, multidirectional water expulsion (scale bar: 5 mm).

Comment 2D: A more explicit comparison of the state-of-the-art in bio-inspired underwater adhesives is needed. (doi/10.1126/scirobotics.abm6695; doi/full/10.1126/scirobotics.aan8072; doi.org/10.1088/1748-3190/ab9418; doi.org/10.1016/j.matt.2020.01.018). While this paper refers to existing technologies, a more comprehensive comparative analysis would strengthen the claim of superiority. Furthermore, a detailed discussion of the limitations of MUSAS compared to existing solutions should be included. For example, are there specific types of soft tissue or environmental conditions where MUSAS performs poorly? Addressing these limitations would provide a more balanced perspective.

Response 2D: We appreciate the constructive feedback from the reviewer. Other bio-adhesive solutions, such as polymeric adhesives, have merits in stopping bleeding and covering holes, which is not the primary application scenario of MUSAS. MUSAS is most suitable for underwater adhesion on soft substrates with dynamic morphing in extreme pH and moisture environments, especially tissue undergoing rapid turnover and regeneration. A comprehensive discussion has been added to the discussion section of the manuscript, and a detailed comparison of bio-inspired underwater adhesives is presented in Table R2-1.

References 2D

- [1] Wang, Y., Yang, X., Chen, Y., Wainwright, D. K., Kenaley, C. P., Gong, Z., ... & Wen, L. (2017). A biorobotic adhesive disc for underwater hitchhiking inspired by the remora suckerfish. *Science Robotics*, 2(10), eaan8072.
- [2] Wang, S., Li, L., Sun, W., Wainwright, D., Wang, H., Zhao, W., Chen, B., Chen, Y. and Wen, L., 2020. Detachment of the remora suckerfish disc: kinematics and a bio-inspired robotic model. *Bioinspiration & Biomimetics*, 15(5), p.056018.
- [3] Su, S., Wang, S., Li, L., Xie, Z., Hao, F., Xu, J., Wang, S., Guan, J. and Wen, L., 2020. Vertical fibrous morphology and structure-function relationship in natural and biomimetic suction-based adhesion discs. *Matter*, 2(5), pp.1207-1221.
- [4] Li, L., Wang, S., Zhang, Y., Song, S., Wang, C., Tan, S., ... & Wen, L. (2022). Aerial-aquatic robots capable of crossing the air-water boundary and hitchhiking on surfaces. *Science robotics*, 7(66), eabm6695.
- [5] Elastosil M4601 <https://www.wacker.com/h/en-us/silicone-rubber/room-temperature-curing-silicone-rubber-rtv-2/elastosil-m-4601-ab/p/000018458>

Table. R2-1|Comparison of MUSAS with previous remora-inspired studies

Feature	MUSAS (Current Study)	Wang et al. Study [1]	Wang et al. Study [2]	Su Et al. Study [3]	Li et al. Study [4]
Primary Inspiration	Remora attachment mechanisms, specifically focused on soft tissue interaction and use in biomedical settings	Remora attachment, designed primarily for hitchhiking applications in underwater vehicles	Subsequent study of [1], focused on developing bio-inspired robot mimicking detachment of remoras for power-reduction of motors	Subsequent study of [1], focused on understanding tissue composition of remora adhesive disk for enhanced suction	Subsequent study of [1], focused on developing aerial-aquatic robots with underwater hitchhiking capabilities
Remora Studies	Adhesion of alive remora (E. nau.) to soft tissue phantom. Studies of the whole Echeneidae family with evolutionary explanation for remora adhesive anatomy adaptations across different species	Adhesion of alive remora (E. nau.) to hard glass.	Detachment of alive remora (E. nau.) from adhesion to glass.	Anatomical study of tissue composition of the adhesive disk of E. nau.	Anatomical study of compartmental structure of the adhesive disk of E. nau.
Structural Focus	Emphasis on unfurling shape, multicompartmental adhesion, and lamella orientation of adhesive disk for adhesion to soft substrates	Focus on the erection of spinules and lamellae for enhancing frictional forces on different surfaces	N/A	One-piece adhesive disk, with composite fibers mimicking tissue composition of remora adhesive disk	Focus on multicompartmental structures for redundant adhesion
Primary Application Focus	Biomedical applications, particularly noninvasive gastrointestinal drug delivery and biosensing	Underwater hitchhiking and attachment for robotic platforms	Underwater hitchhiking and attachment for robotic platforms	Pneumatic soft robotics	Aerial-aquatic robots with underwater hitchhiking capabilities
Device Size	Miniaturized (< 25 mm Length × 9.5 mm Width)	> 10 cm in Length	> 10 cm in Length	> 10 cm in Length	> 10 cm in Length
Lamella Design and Angle Control	Detailed study of lamellar angles and distributions in different remora species to optimize soft substrates adhesion	Limited variation in lamella design; primary focus on achieving overall adhesion via disc structure	N/A	N/A	Limited variation in lamella design; primary focus on achieving overall adhesion via disc structure
Soft Substrates Compatibility	Soft, dynamic, non-intact and porous surfaces with stiffness ranging from 9 kPa to 19 MPa, average areal surface roughness S_a (3D) ranging from 2 μm to 124 μm, and maximum surface roughness S_z (3D) ranging from 30 μm to 662 μm	Tested on rough compliant surface, with stiffness of 6.5 MPa [5], and arithmetic average roughness R_a (2D) of 200 μm	N/A	N/A	N/A
Preloading Requirements	0.05N for adhesion establishment, 0.2N – 0.5N for full adhesion performance	N/A	N/A	1N – 30N	10N – 30N
Actuation Mechanism	Motor-free, leveraging endogenous force of the GI tract	Pneumatic motor	Hydraulic motor	N/A	Hydraulic motor
Materials and Environmental Testing	Insensitivity to pH and humidity variations, suggesting versatility in various environmental conditions for biomedical use	Tested mainly for water-based environments without sensitivity studies for biological environments	Tested mainly for water-based environments without sensitivity studies for biological environments	Tested mainly for water-based environments without sensitivity studies for biological environments	Tested mainly for water-based environments without sensitivity studies for biological environments
Release Mechanism for Biomedical Use	Programmable cohesive failure by leveraging biodegradable lamella materials	No detachment or clearance mechanism for biomedical applications discussed	No detachment or clearance mechanism for biomedical applications discussed	No detachment or clearance mechanism for biomedical applications discussed	No detachment or clearance mechanism for biomedical applications discussed
In-Depth Mechanism Exploration	Leveraging multiphysics simulation, universal mechanical test, μ-CT, particle image velocimetry (PIV), scanning electron microscope (SEM), proliometer imaging, confocal microscope for comprehensive analysis of adhesion and retention mechanisms on soft substrates, including disk shape, lamellae orientation and compartmental adhesion	Leveraging universal mechanical test for enhanced frictional interaction on rough and compliant surfaces	Leveraging PIV to mimic detachment of remora	Leveraging histology, SEM and microscope to understand tissue composition of remora adhesive disk	Leveraging frustrated total internal reflection (FTIR) and universal mechanical test to evaluate compartmental adhesion performance
Biomedical Validation	Comprehensive biocompatibility and safety analyses for potential biomedical applications	N/A	N/A	N/A	N/A
Mechanical Reliability to Dynamic Interference	In vitro shaking incubation test and in vivo endoscopic interference test	N/A	N/A	N/A	N/A
Testing Requirement for Application Demonstration	Unstructured in vivo dynamic environment including complex movements, peristalsis, variable fluid flow, and mechanical forces, using over 55 pigs and 8 fish without interfering with their normal feeding, resting, or behaviors.	Laboratory fish tank	Laboratory fish tank	N/A	Unstructured natural environment including artificial architectures, rocks in canyon and streams, and artificial plastic objects on ocean
Retention Demonstration	Demonstrated GI retention in pigs and body surface retention in fish (>6) under unstructured in vivo dynamic conditions	N/A	N/A	N/A	N/A
Application Novelty	Ultra-miniaturized underwater battery-free, wireless temperature biosensing, noninvasive adhesion and health-monitoring for GERD, sustained release of PrEP for HIV/AIDS, major advancement in demonstration of mRNA transfection in the gastrointestinal (GI) tract	Hitchhiking underwater vehicles	N/A	N/A	Aerial-aquatic robots with underwater hitchhiking capabilities

Comment 2E: Recently, the specification of preload in adhesion has become a very important issue. Simulation and confocal images suggest that preload is required for adhesion, and the authors should specify whether preload is present and its exact value when measuring adhesion.

Response 2E: We thank the reviewer for the excellent point. A 0.5 N preload was applied to all MUSAS-related mechanical tests, which is now explicitly specified in the methods section. Furthermore, we have added our characterization of adhesion performance under various preloading conditions, using underwater adhesion on a stomach tissue phantom as an example (Fig. R2-5). The results revealed that a preload as low as 0.05 N enables MUSAS adhesion, while a 0.2 N preload is sufficient to achieve near-optimal performance. These findings further confirmed the successful minimization of MUSAS preloading requirements by leveraging optimal lamella orientation and disk furling, highlighting its distinctive self-adhesion capability driven by GI tract contractions, as shown in Fig. 4E and Supp. Video S7.

Fig. R2-5|Representative measurements of normal adhesion force of MUSAS adhering to phantom stomach tissue underwater, under varying preloading conditions

Comment 2F: It would be important to confirm whether the biocompatibility of the materials used in MUSAS has been thoroughly tested, particularly in the context of in vivo applications. The manuscript would benefit from a discussion of the measures taken to ensure the long-term stability and safety of the material within biological environments.

Response 2F: We appreciate the constructive feedback from the reviewer. We have expanded the discussion to include biocompatibility considerations based on the materials composing MUSAS, primarily nitinol and silicone rubber (EcoFlex), and highlighted our testing for long-term stability. In fact, both nitinol and silicone rubber have reached extensive human use in FDA-approved products, as documented in published FDA medical device material safety summaries [2F-1, 2F-2]. Specifically, nitinol, one of the most widely used biomaterials, has been extensively employed in endoscopy, arch wiring, cardiovascular stents, orthopedics, and artificial organs since the 1970s [2F-3, 2F-4]. Additionally, numerous reports over the past decade have documented the use of silicone rubber, such as FDA approved biomaterials PDMS and Eco-Flex, in implantable and ingestible electronics and biomedical devices [2F-5, 2F-6]. A detailed discussion of the *in vivo* evaluation of biosafety and the noninvasive characteristics of MUSAS can be viewed in our response to Comment 2J.

References 2F

- [1] <https://www.fda.gov/media/158490/download?attachment>
- [2] <https://www.fda.gov/media/152353/download?attachment>
- [3] Ryhänen, J. "Biocompatibility of nitinol." *Minimally Invasive Therapy & Allied Technologies* 9.2 (2000): 99-105.
- [4] Duerig, T., Pelton, A. and Stöckel, D.J.M.S., 1999. An overview of nitinol medical applications. *Materials Science and Engineering: A*, 273, pp.149-160.
- [5] Dagdeviren, Canan, et al. "Conformal piezoelectric systems for clinical and experimental characterization of soft tissue biomechanics." *Nature materials* 14.7 (2015): 728-736.
- [6] Zhang, Yamin, et al. "Advances in bioresorbable materials and electronics." *Chemical Reviews* 123.19 (2023): 11722-11773.

Comment 2G: The mechanical stability of MUSAS during prolonged residence in the body seems crucial, especially in terms of potential damage or degradation. A description of the tests conducted to verify the system's resistance to such issues over extended periods would enhance the manuscript.

Response 2G: We appreciate the constructive feedback from the reviewer. We have expanded the discussion to include biocompatibility considerations based on the materials composing MUSAS and have highlighted our testing for long-term stability.

We conducted *in vitro* studies to evaluate the fouling and failure mode of MUSAS incubated at body temperature (37°C) in gastric fluid harvested from Yorkshire pigs. Microscope and scanning electron microscope (SEM) imaging confirmed that fouling remained negligible at the micron scale for shape memory alloy (SMA) lamellae after 7 days of incubation (Fig. R2-6B & R2-6D). After 21 days, minor fouling was observed at the 50 µm scale, with minimal changes in SMA surface morphology at the 5 µm scale, indicating negligible interference with mechanical interlocking (Fig. R2-6D). However, surface morphology reconstruction of silicone rubber occurred over time due to the reported water permeability of Ecoflex (260 g/m²day⁻¹ [1A-1]). Specifically, swelling and fouling of the silicone rubber's crease pattern appeared after 7 days and became more evident after 21 days, suggesting a gradual loss of friction, which is crucial for maintaining vacuum-based mechanical adhesion in MUSAS (Fig. R2-6C). Additionally, changes in silicone rubber morphology were observed within the 10–1000 nm length scale (Fig. R2-6E). Initially, Ecoflex appeared smooth with a few detectable setal structures at high magnification (500 nm); however, fouling and surface roughening gradually developed after 7 days, and became evident after 21 days. This degradation could weaken intermolecular interactions, eventually leading to leakage and failure of the vacuum-based seal. These results suggest that the primary failure mode of MUSAS will be caused by the loss of vacuum-based mechanical adhesion due to water exchange, swelling and fouling of the silicone rubber, occurring at 500 nm to 50 µm. Future studies could explore improved waterproof silicone rubber and finer setal structures within the 10–1000 nm range to enhance the longevity and functionality of MUSAS. A discussion of these findings has been added to the main manuscript and supplementary information.

Additionally, as now explicitly clarified in the main manuscript and Supplementary Information sections, the mechanical stability of MUSAS was validated in a dynamic and unstructured *in vivo* environment using > 55 pigs and > 8 fish. Specifically, we tested GI retention in pigs (> 9)

and body surface retention in fish (> 6) under survival conditions, where the animals were monitored without interference with their normal feeding, resting, and behaviors (Fig. R2-7). In addition to the aforementioned *in vitro* dynamic interference studies (Fig. R2-2), the mechanical stability of MUSAS was further validated *in vivo* through endoscopic interference studies. Supp. Video S6 serves as an example, demonstrating the adhesion stability of MUSAS when subjected to touching, pushing and shaking interference.

Fig. R2-6|*In vitro* characterization of fouling of MUSAS. A. Illustration of the material components of MUSAS. B. Microscope pictures of shape memory alloy (SMA) lamellae incubated in swine gastric fluid at 37 °C for different period of time (scale bar: 1 mm). C. Microscope pictures of silicone rubber (Ecoflex 0030) incubated in swine gastric fluid at 37 °C for different period of time (scale bar: 1 mm). D. SEM images characterizing surface morphology of SMA incubated for different period of time at micron scale. E. SEM images characterizing surface morphology of silicone rubber (Ecoflex 0030) incubated for different period of time at micron and nanometer scales.

Fig. R2-7|*In vivo* retention performance of MUSAS. A. Controlled retention and detachment of MUSAS in the swine stomach through programmed mechanical interlocking with various lamella materials. B. Illustration of MUSAS with radioactive imaging agents for routine retention checks (scale bar: 5 mm). C. X-rays of long-term retention of 4 MUSAS with shape-memory nitinol lamella delivered in the stomach before safe passage in the GI tract. D. X-rays of retention of MUSAS with superelastic nitinol lamella and stainless steel lamella delivered in the stomach before safe passage in the GI tract. E. Adhesion of MUSAS in various buccal regions for mRNA delivery. F. Representative retention performance of MUSAS in the duodenum of the small intestine (SI) in a swine model, evaluated in a terminal study on the day of euthanasia. G. Representative retention performance of MUSAS on different surface parts of a fish model.

Comment 2H: In drug delivery applications, ensuring the precise localization of MUSAS at target sites within the body would be essential. It would be helpful to describe any specific strategies or mechanisms employed to control the positioning of MUSAS for more effective delivery. At present, it is unclear whether controlled adhesion at precise locations is possible.

Response 2H: We appreciate the reviewer's constructive feedback. Programmable organ targeting remains a major goal in drug delivery. Many drug formulations, for instance, exhibit optimal absorption in nonacidic environments. Quinidine, a therapeutic used to treat irregular heartbeats, exemplifies this need, requiring precise release in the small intestine [2H-1 & 2H-2].

Our close collaboration with physicians and pharmaceutical scientists has directed this study toward a passive-control approach, focusing on three key aspects of drug delivery: 1) miniaturization, preferably to an ingestible size; 2) achieving a high active pharmaceutical ingredient (API) release rate without sacrificing drug loading for motors and controllers; and 3) simple, user-friendly deployment without the need for additional maneuvering equipment or specialized training. This approach contrasts with active-robotics-based strategies, which often depend on large MRI and fluoroscopy systems for active navigation—an aspect that, in our view, limits clinical feasibility due to complex facility and surgical requirements. Here, we demonstrate a pH-responsive deployment strategy and showcase how MUSAS can be precisely delivered to the small intestine (Fig. R2-8). We utilized the polymethacrylate-based copolymer Eudragit S100, which dissolves in natural to alkaline environments (pH > 7.0). The dip-coating solution was prepared by dissolving Eudragit in ethanol. *In vitro* characterization confirmed that MUSAS encapsulation remains stable for up to 24 hours in simulated gastric fluid, depending on the S100 coating thickness (Fig. R2-8A). Additionally, release timing in a simulated intestinal environment can be precisely controlled by adjusting the coating thickness and S100 concentration (Fig. R2-8B & R2-8C). *In vivo* results further demonstrated safe gastric emptying in the stomach and timely deployment in the small intestine (Fig. R2-8D).

Notably, other pH-responsive copolymer materials can be applied to enable programmable, targeted delivery of MUSAS to different regions of the GI tract. Future studies may also explore replacing the disk materials of MUSAS with ferromagnetic silicone rubber to enable hand-size magnet-assisted delivery. A detailed discussion has been added to the supplementary information.

References 2H

- [1] Stillhart, Cordula, et al. "Impact of gastrointestinal physiology on drug absorption in special populations—An UNGAP review." *European Journal of Pharmaceutical Sciences* 147 (2020): 105280.
- [2] Abuhelwa, Ahmad Y., et al. "Food, gastrointestinal pH, and models of oral drug absorption." *European journal of pharmaceutics and biopharmaceutics* 112 (2017): 234-248.

Fig. R2-8| pH-responsive coating enabled programmable targeted delivery of MUSAS to the small intestine. A. *In vitro* demonstration of the prolonged stability of MUSAS encapsulation in simulated gastric fluid, dip-coated with the pH-dependent copolymer Eudragit S100 (scale bars: left, 5 mm; right, 1 cm). B and C. *In vitro* demonstration of the programmable controlled release of MUSAS in a simulated intestinal environment, dip-coated with different concentrations of the copolymer Eudragit S100 (scale bars: 1 cm). C. *In vivo* demonstration of programmable targeted delivery of MUSAS to the small intestine in a swine model, including safe gastric emptying in the stomach from 0 to 30 min and timely deployment in the small intestine from 35 mins to 50 mins (scale bars: 5 mm).

Comment 2J: Is controllable adhesion of the device possible after a desired period of time? The process by which MUSAS detaches and eventually is expelled after a long-term residence in the GI tract seems to require further explanation. A detailed analysis of how MUSAS safely detaches from soft tissue surfaces before passing through the body would enhance the discussion. Given the mechanical properties and size of the device, is there a possibility that the device could cause damage to the small and large intestine during its expelling process after detachment from the organ?

Response 2J: We appreciate the constructive feedback from the reviewer. Choices of biodegradable lamella materials to program loss of mechanical interlocking can be used to achieve controllable adhesion and detachment, as discussed in Fig. R2-7A to Fig. R2-7D. We have also added these *in vivo* retention studies of different lamella materials to Fig. 4 and Ext. Fig. 5. Compared to nitinol lamella with shape memory effects which can stay up to 22 days, retention of stainless steel lamellae based devices is about 1 days due to its quick degradation in acid environment, and superelastic nitinol lamellae-design generally enable retention time up to 4 days (Fig. R2-7A - Fig. R2-7D).

In terms of safe passage of MUSAS in the GI tract, as we shown in Fig. 4H-III, MUSAS can be viewed as a noninvasive microneedle platform, of which penetration depth is within 800 μm , and diameter of the penetration holes is within 100 μm . Comprehensive studies of microneedles have been prevalent in the past decades, confirming the noninvasive and fast tissue repair within 24 hours, with limited adverse side effects [2J-1]. In fact, in the rare occasion where MUSAS was still present at the terminal study on the day of euthanasia and naturally detached from the pig stomach, we were able to visualize that its adhesive compartments and lamellae were covered and thoroughly lubricated with stomach contents and mucoid materials (Fig. R2-9B), leading to loss of its adhesion property and thus, safe passage in the GI tract. During the *in vivo* validation of MUSAS in > 55 pigs, re-adhesion of the devices after their detachment were never visualized during our routine X-ray checks (see Supplementary Information for frequency of X-rays). Based on X-ray results, we estimate that the gastric emptying time for detached MUSAS is generally less than one day, which aligns with the typical gastric emptying pattern reported in pigs for safe passage of stomach contents [2J-2]. In addition, health monitoring of our experimental pig during the adhesion and the passage of MUSAS, conducted by MIT animal committee on animal care and division of comparative medicine, were unremarkable. Pigs showed normal behaviors, food consumption and normal weight gain (Fig. R2-9C). Further, we conducted various histological analyses with internal controls to compare the adhesion sites of MUSAS (stomach, buccal cavity, pharynx, and esophagus, localized with a tattoo) with non-adhesion areas of MUSAS, as well as downstream intestinal and colonic tissues collected after its safe passage. Histopathology of H&E-stained tissues of the gastrointestinal tract evaluated by a board-certified veterinary pathologist confirmed that MUSAS caused no evidence of significant damage including hemorrhage, inflammation or indication of tissue repair (fibrosis) (Fig. R2-9D – R2-9H).

References 2J

- [1] Kim, Yeu-Chun, Jung-Hwan Park, and Mark R. Prausnitz. "Microneedles for drug and vaccine delivery." *Advanced drug delivery reviews* 64.14 (2012): 1547-1568.
- [2] Gregory, P.C., McFadyen, M. and Rayner, D.V., 1990. Pattern of gastric emptying in the pig: relation to feeding. *British Journal of Nutrition*, 64(1), pp.45-58.

Fig. R2-9|*In vivo* biocompatibility evaluation of MUSAS in a swine model. A. Illustration of tattooing adhesion sites in a swine model (scale bar: 5 mm). B. Detached MUSAS retrieved from a pig stomach after euthanasia, demonstrating thorough lubrication and accumulation of stomach contents and mucoid materials for safe passage (scale bar: 5 mm). C. Representative pig weight measurements show normal weight gain during MUSAS residence ($n = 4$ devices delivered) in the stomach. Histopathology of H&E stained the gastrointestinal tract showing no damage from MUSAS adhesion and passage that causes hemorrhage, inflammation or indication of tissue repair (fibrosis), including D. Stomach. MUSAS adhesion and adjacent internal control sites, collected after 3 and 7 days of MUSAS residence (scale bar: 100 μm). E. Small intestinal and colon, collected after MUSAS residence in the stomach and safe passage through the GI tract (scale bar: small intestine, 200 μm ; colon, 100 μm). F. Esophagus, adhesion and adjacent internal control sites, collected 1 day after adhesion (scale bar: 200 μm). G and H. Buccal tissue and pharynx, adhesion and adjacent internal control sites, collected 1 day after mRNA delivery (scale bar: 100 μm).

Referee #3 (Remarks to the Author):

Inspired by the remarkable underwater adhesion of the Remora fish, Kang et al. developed the Mechanical Underwater Soft Adhesion System (MUSAS) to overcome current limitations of adhesives within these environments. Through a detailed evaluation of the physical process by which remoras adhere to soft and stiff substrates and characterization of the adhesive units of a range of remora species, the authors uncovered biological mechanisms of adhesion and then developed a bioinspired device that is able to adhere to a range of substrates underwater through a simple compressive force. Overall, this is a very useful bioinspired approach and the ability to adhere to substrates through tissue contractions (e.g. esophageal movements) make it a unique adhesive system where self-adhesion is possible and the non-invasive delivery to the tissue (e.g. endoscope) is a strength. Also, the use of mechanically interlocking adhesives such as MUSAS is important for application in a range of physiological environments (e.g. low pH of stomach), where swelling (e.g. microneedle adhesives) may be limited. Interesting examples were also provided from biosensing to therapeutics. Lastly, the investigation of such approaches potentially gives some insight into how natural selection might have driven the evolution of various features of these fish.

Major points:

Comment 3A: One concern is the lack of novelty. Previous work by Wang et. al. “A biorobotic adhesive disc for underwater hitchhiking inspired by the remora suckerfish” (Science Robotics, 2017) also took a biomimetic and remora based approach to create an adhesive disc. It is suggested that the authors clearly describe the key differences and advantages of their work compared to this reported system, beyond just now applying to biological tissues.

Response 3A: We sincerely appreciate the reviewer’s constructive feedback and have carefully revised our manuscript to address each point raised. Before detailing these revisions, we would like to provide a comprehensive overview of the novelty and major advancements introduced by MUSAS in addressing critical challenges across multiple fields.

This study uniquely explores previously unexamined adhesion mechanisms of remoras on soft substrates, integrating the anatomical evolution of the adhesive disk across all remora species, which has not been systematically analyzed before. The subsequent development of MUSAS enables *in situ*, long-term retention adhesion on dynamic, morphable soft substrates that undergo rapid turnover and regeneration in extreme pH and moisture environments (e.g., the stomach). This presents significant challenges for current adhesive solutions, particularly in the context of mucoadhesion and biomedical applications.

Our *in vitro*, *ex vivo*, and *in vivo* validations push the boundaries of technical feasibility and research capabilities, supported by comprehensive, multimodal cross-validation in diverse application scenarios. Furthermore, we have developed a range of MUSAS-based applications, each with independent novelty, being introduced to the scientific community to tackle long-standing, immediate challenges. These innovations represent significant advances in bioadhesion, biomedical treatments, and biological exploration.

The novelty of this work can be summarized in three key levels:

1. *Novelty in Remora Adhesion Studies* - New insights into remora adhesion on dynamic, morphable soft substrates and the evolution of the adhesive disk anatomy

We acknowledge the contributions of prior remora studies from Dr. Li Wen and Dr. Robert Wood's group [3A-1] and subsequent research [3A-2 - 3A-4]. However, previous studies focused on a single remora species (*Echeneis naucrates*) adhering to static, hard surfaces (e.g., glass). These studies primarily concluded that remora adhesion is facilitated by lamella erection and disk muscle stiffness.

In contrast, our study:

- Conducts a comprehensive review of all described remora species to examine anatomical variations and evolutionary adaptations in adhesive disk morphology.
- Explores adhesion on soft, dynamic substrates—a previously unaddressed challenge in bioadhesion.
- Investigates live remora adhesion on a stomach tissue phantom, uncovering new adhesion mechanisms based on disk unfurling behavior and lamellae orientation, which play a dominant role in soft adhesion.
- Proposes a novel evolutionary explanation for remora adhesive anatomy adaptations across different species.

This expands bioadhesion knowledge and provides new insights into biomimetic adhesion design, distinct from previous studies.

2. *Novelty in Remora-Inspired Adhesion Strategies* - Motor-Free Self-Adhesion vs. Motor-Actuated Robotics

Our study addresses the fundamental challenge of long-term adhesion on dynamic, morphable soft substrates that experience rapid regeneration and extreme conditions (e.g., the gastrointestinal (GI) tract). This remains an unresolved issue in bioadhesion, especially mucoadhesion research.

For example, gastric epithelial cells renew every ~3 days, presenting a significant barrier to long-term retention. Despite decades of research [2A-5], most current adhesion solutions fail *in vivo* unless external stimuli (e.g., electric fields) significantly modify surface anatomy [2A-6].

Key distinctions from prior remora-inspired studies:

- Previous studies [3A-1 - 3A-4] focused on motor-actuated underwater robots for hitchhiking functionalities, without addressing fundamental challenges of long-term adhesion to soft, morphable tissues.
- These studies used pneumatic, motor-actuated lamella erection (>10 cm devices) to optimize pitching angles for adhesion, which is impractical for biomedical applications.
- Our study focuses on developing a miniaturized, self-adhesive system, avoiding the need for motors, external actuation, or complex maneuvering.

Our design strategy:

We introduce a motor-free self-adhesion solution, based on:

- 1) Miniaturization – Ingestible device size (< 25 mm × 9.5 mm width).

- 2) High drug-loading efficiency – Unlike motor-actuated systems, our approach preserves the active pharmaceutical ingredient (API) payload.
- 3) Simplified, user-friendly deployment – No need for new equipment or additional training.

By leveraging disk unfurling behavior and lamellae orientation, MUSAS achieves stable adhesion with minimum preload and maximizes long-term retention in real-world dynamic environments.

Furthermore, prior studies cannot be directly applied to soft substrate adhesion because:

- Shear frictional performance on soft substrates is not sensitive to lamella contact (pitching) angles but spinule lengths (Supp. Fig. S6), unlike hard surfaces.
- Previous experimental setups lacked variety—tested materials were stiff (Elastosil M4601: 6.5 MPa Young’s Modulus) or smooth (Ecoflex: 60.05 - 6 μm roughness according to our own measurements) [3A-8, 3A-9].
- Soft substrates (e.g., mucosal layers) exhibit dynamic sliding and surface morphing, requiring completely different adhesion strategies (Supp. Fig. S5).

Our study systematically addresses these limitations through comprehensive validations.

3. Major Advances in Bioelectronics, Biomedical Treatments, and Biological Exploration

MUSAS represents a significant translational advancement, tackling critical biomedical and biological challenges.

Unlike studies that propose conceptual applications, each MUSAS-enabled application underwent rigorous, independent *in vitro*, *ex vivo*, and *in vivo* validation, employing gold-standard animal models, particularly the swine model, which offers advance translational relevance due to its close anatomical resemblance to humans [3A-10].

Key applications and major advancements:

- 1) Marine Life Exploration – Developed an ultraminiaturized reported underwater, battery-free, wireless temperature sensor for real-time monitoring of swimming organisms in near-natural conditions.
- 2) Noninvasive Monitoring of Gastroesophageal Reflux Disease (GERD) – Developed a pH-sensitive impedance sensor, demonstrating an unprecedented alternative adhesion-based esophageal health-monitoring system.
- 3) Sustained Drug Release for HIV/AIDS PrEP – Successfully developed a MUSAS-enabled slow-release Cabotegravir formulation, demonstrating >1 week retention *in vivo* (validated in 9 pigs, with pharmacokinetic studies detailed in Fig. 4).
- 4) Effective mRNA Delivery in the GI Tract – Developed an unprecedented device-enabled mRNA delivery system (215 μL capacity), demonstrating functional luciferase mRNA transfection in large animals.
 - Only two previous studies [3A-11, 3A-12] reported mRNA detection in large animals, but without confirming functional transfection.

- Our study achieved functional protein expression, confirmed through bioluminescence imaging without requiring prohibitively expensive systemic luciferin administration.
- This major advancement in demonstration of mRNA transfection in the GI tract, overcame long-standing biological barriers, with profound implications for vaccination and genetic therapeutics.

We respectfully request recognition of the significant advances enabled by MUSAS. This work extends beyond conceptual applications—each claim is rigorously validated through extensive animal studies (e.g., >55 pigs, >8 fish over three years). These findings open new frontiers in bioadhesion, biomedical treatments, and biological exploration, marking a transformative advancement in translational science.

A thorough comparison table summarizing the unique aspects of MUSAS versus other technologies is now provided in the supplementary information sections.

References 3A

- [1] Wang, Y., Yang, X., Chen, Y., Wainwright, D. K., Kenaley, C. P., Gong, Z., ... & Wen, L. (2017). A biorobotic adhesive disc for underwater hitchhiking inspired by the remora suckerfish. *Science Robotics*, 2(10), eaan8072.
- [2] Li, L., Wang, S., Zhang, Y., Song, S., Wang, C., Tan, S., ... & Wen, L. (2022). Aerial-aquatic robots capable of crossing the air-water boundary and hitchhiking on surfaces. *Science robotics*, 7(66), eabm6695.
- [3] Wang, S., Li, L., Sun, W., Wainwright, D., Wang, H., Zhao, W., Chen, B., Chen, Y. and Wen, L., 2020. Detachment of the remora suckerfish disc: kinematics and a bio-inspired robotic model. *Bioinspiration & Biomimetics*, 15(5), p.056018.
- [4] Su, S., Wang, S., Li, L., Xie, Z., Hao, F., Xu, J., Wang, S., Guan, J. and Wen, L., 2020. Vertical fibrous morphology and structure-function relationship in natural and biomimetic suction-based adhesion discs. *Matter*, 2(5), pp.1207-1221.
- [5] Nan, K., Feig, V.R., Ying, B., Howarth, J.G., Kang, Z., Yang, Y. and Traverso, G., 2022. Mucosa-interfacing electronics. *Nature Reviews Materials*, 7(11), pp.908-925.
- [6] Ying, B., Nan, K., Zhu, Q., Khuu, T., Ro, H., Qin, S., ... & Traverso, G. (2023). Electroadhesive hydrogel interface for prolonged mucosal theranostics. *bioRxiv*, 2023-12.
- [7] Huie, Jonathan M., and Adam P. Summers. "The effects of soft and rough substrates on suction-based adhesion." *Journal of Experimental Biology* 225.9 (2022): jeb243773.
- [8] Elastosil M4601 <https://www.wacker.com/h/en-us/silicone-rubber/room-temperature-curing-silicone-rubber-rtv-2/elastosil-m-4601-ab/p/000018458>
- [9] Althumayri, Majed Othman, Azra Yaprak Tarman, and Hatice Ceylan Koydemir. "Bioinspired skin-like *in vitro* model for investigating catheter-related bloodstream infections." *Scientific Reports* 14.1 (2024): 26167.
- [10] Lunney, J.K., Van Goor, A., Walker, K.E., Hailstock, T., Franklin, J. and Dai, C., 2021. Importance of the pig as a human biomedical model. *Science translational medicine*, 13(621), p.eabd5758.
- [11] Abramson, Alex, et al. "Oral mRNA delivery using capsule-mediated gastrointestinal tissue injections." *Matter* 5.3 (2022): 975-987.
- [12] Schultz, D., Kempen, P. J., Primdahl, S., Pereverzina, M., Uhrenfeldt, A. H., Alba, E. M., ... & Urquhart, A. J. (2024). Gastrointestinal device-mediated delivery of mRNA-lipid

nanoparticles achieves distinct expression and biodistribution in mice and pigs. *ACS Applied Materials & Interfaces*, 16(49), 67192-67202.

Comment 3B: One disadvantage to mechanical interlocking systems is that they inherently cause tissue damage, which is avoided with other tissue adhesive systems (e.g. carbopol-hydrogen bonding, EDC-NHS-covalent), and this was not avoided in this work. Due to the microscale tissue damage (300-800 μm deep 100 μm wide holes), this approach will likely be limited to use with tissues that have a high regenerative capacity and low propensity for scarring, as well as solid organs that will not leak if punctured (e.g. vasculature, lungs). This limitation should be discussed thoroughly and put into context of potential applications.

Response 3B: We appreciate the constructive feedback from the reviewer. In response, we have incorporated a thorough discussion in the Introduction and Discussion sections to address the limitations of MUSAS. While we acknowledge that MUSAS is most suited for tissues with high regenerative capacity and not sensitive to leakage, we respectfully want to emphasize that its development provided a viable solution for *in situ* adhesion and long-term retention of adhesive platforms on dynamic, morphable soft substrates that undergo rapid turnover and regeneration in extreme pH and moisture environments. This remains a significant challenge in the adhesive research community, particularly in the context of mucoadhesion. For example, gastric epithelial cells renew within as little as three days, posing a fundamental barrier to physicochemical adhesion mechanisms (e.g., carbopol-hydrogen bonding, EDC-NHS covalent bonding) that rely on stable surface interactions.

Additionally, while the microscale tissue penetration of MUSAS limits its applicability to organs with thin serosal layers and sensitive to leakage—such as vasculature or lungs—this feature presents a major advantage for breaching the mucosal barrier, enabling submucosal delivery of biological therapeutics, a key challenge in GI tract drug delivery. As shown in Fig. 4H-III, MUSAS can be regarded as a noninvasive microneedle platform, with a penetration depth of up to 800 μm and penetration hole diameters within 100 μm . Extensive studies on microneedles over the past decades have demonstrated their noninvasive nature and rapid tissue repair within 24 hours, with minimal adverse effects [3B-1].

Furthermore, the silicone rubber-based suction disk of MUSAS allows seamless integration with soft electronics for biomedical sensing. The silicone rubber family (e.g., PDMS, Ecoflex) serves as an ideal substrate for soft electronics, offering a distinct advantage over hydrogel-based solutions, which require complex material modifications for electronic integration [3B-2]. For a detailed discussion on the safe detachment and passage of MUSAS in the GI tract, please refer to our response to Comment 3C.

References 3B

- [1] Kim, Yeu-Chun, Jung-Hwan Park, and Mark R. Prausnitz. "Microneedles for drug and vaccine delivery." *Advanced drug delivery reviews* 64.14 (2012): 1547-1568.
- [2] Zhang, Yamin, et al. "Advances in bioresorbable materials and electronics." *Chemical Reviews* 123.19 (2023): 11722-11773.

Comment 3C: Related to this, how do the authors foresee this device being deployed in a clinical setting? There are clear examples proposed throughout the work, however, many aspects are left unclear.

-Is there any control over device detachment or ways to trigger this? Is it possible that the device will never detach? What if scarring occurs around the device?

Response 3C-I: We appreciate the reviewer's constructive feedback. The versatile adhesion of MUSAS to various soft substrates enables a range of clinical applications. First, it can be manually deployed for buccal and esophageal applications, including mRNA delivery and biosensing. For ingestible applications targeting the stomach or intestines, MUSAS can be enclosed in a capsule and achieve self-adhesion by leveraging the endogenous forces of the GI tract (Fig. 4D & 4E, Ext. Fig.6).

Regarding device detachment control, we have demonstrated the use of biodegradable lamella materials to program the loss of mechanical interlocking for controlled detachment. We compared three lamella materials: (1) Nitinol lamellae with shape memory effects, which can last up to 22 days; (2) Stainless steel lamellae, which degrade within approximately 1 day in an acidic environment; and (3) Superelastic nitinol lamellae, which generally enable retention for up to 4 days (Fig. R3-1). As discussed in response 3B, MUSAS is a noninvasive microneedle platform. The rapid turnover and regeneration of mucosa and the extreme pH and moisture environments of the stomach will intrinsically weaken the mechanical interlocking and adhesion gradually, ensuring the eventual detachment of MUSAS. Histopathology studies confirmed the noninvasiveness of MUSAS and the absence of scarring, as observed in response 3C-III.

-Once the device detaches, what does clearance of the device look like? Is it possible that the device could continue to adhere to various GI tissues as it is passed through?

Response 3C-II: In the rare occasion where MUSAS was still present at the terminal study on the day of euthanasia and naturally detached from the pig stomach, we were able to visualize that its adhesive compartments and lamellae were covered and thoroughly lubricated with stomach contents and mucoid materials (Fig. R3-2B), leading to loss of its adhesion property and thus, safe passage in the GI tract. During the *in vivo* validation of MUSAS in > 55 pigs, readhesion of the devices after their detachment were never visualized during our routine X-rays checks (see Supplementary Information for frequency of X-rays). Based on X-ray results, we estimate that the gastric emptying time for detached MUSAS is generally less than one day, which aligns with the typical gastric emptying pattern reported in pigs for safe passage of stomach contents [3C-1]. In addition, health monitoring of our experimental pig during the adhesion and the passage of MUSAS, conducted by MIT animal committee on animal care and division of comparative medicine, were unremarkable. Pigs showed normal behaviors, food consumption and normal weight gain (Fig. R3-2C).

-What level of damage does this device cause to the target tissues and downstream tissues as it is cleared? Histology or necrosis measurements are needed of tissues after various periods of attachment.

Response 3C-III: We conducted various histological analyses with internal controls to compare the adhesion sites of MUSAS (stomach, buccal cavity, pharynx, and esophagus, localized with a tattoo) with non-adhesion areas of MUSAS, as well as downstream intestinal and colonic tissues collected after its safe passage. Histopathology of H&E-stained tissues of the gastrointestinal

tract evaluated by a board-certified veterinary pathologist confirmed that MUSAS caused no evidence of significant damage including hemorrhage, inflammation or indication of tissue repair (fibrosis) (Fig. R3-2D - R3-2H).

References 3C

- [1] Gregory, P.C., McFadyen, M. and Rayner, D.V., 1990. Pattern of gastric emptying in the pig: relation to feeding. *British Journal of Nutrition*, 64(1), pp.45-58.

Fig. R3-1|In vivo retention performance of MUSAS. A. Controlled retention and detachment of MUSAS in the swine stomach through programmed mechanical interlocking with various lamella materials. B. Illustration of MUSAS with radioactive imaging agents for routine retention checks (scale bar: 5 mm). C. X-rays of long-term retention of 4 MUSAS with shape-memory nitinol lamella delivered in the stomach before safe passage in the GI tract. D. X-rays of retention of MUSAS with superelastic nitinol lamella and stainless steel lamella delivered in the stomach before safe passage in the GI tract.

Fig. R3-2 *In vivo* biocompatibility evaluation of MUSAS in a swine model. A. Illustration of tattooing adhesion sites in a swine model (scale bar: 5 mm). B. Detached MUSAS retrieved from a pig stomach after euthanasia, demonstrating thorough lubrication and accumulation of stomach contents and mucoid materials for safe passage (scale bar: 5 mm). C. Representative pig weight measurements show normal weight gain during MUSAS residence ($n = 4$ devices delivered) in the stomach. Histopathology of H&E stained the gastrointestinal tract showing no damage from MUSAS adhesion and passage that causes hemorrhage, inflammation or indication of tissue repair (fibrosis), including D. Stomach. MUSAS adhesion and adjacent internal control sites, collected after 3 and 7 days of MUSAS residence (scale bar: 100 μm). E. Small intestinal and colon, collected after MUSAS residence in the stomach and safe passage through the GI tract (scale bar: small intestine, 200 μm ; colon, 100 μm). F. Esophagus, adhesion and adjacent internal control sites, collected 1 day after adhesion (scale bar: 200 μm). G and H. Buccal tissue and pharynx, adhesion and adjacent internal control sites, collected 1 day after mRNA delivery (scale bar: 100 μm).

Comment 3D: It is also difficult for this reviewer to compare MUSAS to alternate solutions (e.g. EDC-NHS, carbopol polymeric adhesives), where adhesion values are often reported as tensile strength and shear strength, rather than adhesive pressure and shear adhesive pressure, respectively. It would be helpful to expand on this and report adhesion values as tensile strength and shear strength so direct comparison can be made with other systems. Also, how was adhesion pressure measured for MUSAS compared to other solutions (values for Carbopol and EDC-NHS seem lower than expected)? How much pressure was applied to MUSAS during adhesion testing?

Related, it seems somewhat inappropriate to compare to polymeric adhesives. MUSAS are at a completely different scale-macro scale versus molecular. These are continuous materials that can stop bleeding, cover holes etc. this system requires a solid material to adhere.

Response 3D: We appreciate the constructive feedback from the reviewer. We recognize that researchers from different backgrounds use varying nomenclatures and would like to clarify that our study presents an apples-to-apples comparison. Specifically, adhesion strength and adhesion pressure are equivalent concepts, though they are referred to differently by the device and polymeric adhesive communities. As clarified in the Methods section, adhesion pressure was calculated by dividing the force by the adhesion surface area, aligning with the established methodology in previous research: “strength was determined by dividing the maximum force by the adhesion area” [3D-1]. To enhance clarity, we have standardized the terminology in the manuscript, replacing adhesion pressure with adhesion strength. Additionally, we have expanded our discussion on mechanical testing, as suggested by the reviewer.

Specifically, the mechanical testing method employed in this study is balanced and ensures fair comparisons across different adhesive solutions, and the adhesion strength values for Carbopol and EDC-NHS are expected to be lower than reported literatures. As discussed in the manuscript, polymeric adhesives typically exhibit increased strength following a prolonged, pressurized pre-adhesion process (5-30 minutes) [3D-2]. To ensure a fair comparison, we limited the pre-adhesion pressurization for NHS-EDC and Carbopol hydrogels to 3 minutes, since MUSAS achieves instant mechanical adhesion. Furthermore, mechanical adhesion tests for polymeric adhesives often involve trimming tissue to a thickness of 1-2 mm to match the thickness of the polymeric adhesive [3D-1, 3D-2]. However, stomach and other organ tissues are significantly thicker (8-10 mm) [3D-3], with dynamic sliding between the serosal and mucosal layers that affects adhesion performance (Supp. Fig. S5). Additionally, polymeric adhesive tests are typically conducted on the serosal or epidermal layers, requiring surface washout and the removal of liquid accumulation from wet surfaces [3D-1, 3D-2]. Besides, adhesion tests were often conducted 24 hours post-application to account for equilibrium swelling of the polymeric adhesive [3D-1]. However, in such cases, post-mortem autolysis and tissue decomposition may occur, and adhesion strength is often measured at deeper serosal or muscular layers. In contrast, the mechanical adhesion testing in our study was performed on freshly harvested tissue (< 1 hour post-euthanasia) without surface washout, tissue trimming, or liquid removal. The loading rate of the mechanical test considered the ASME tissue adhesion standards that were adopted by the polymeric adhesive community. Soft substrates used in test were only partially secured, with the four corners of the tissue squares glued to the holder to allow for natural sliding and dynamic morphing. These factors account for the fair yet expectedly lower adhesion strength of polymeric adhesives compared to previously reported data.

Regarding preloading, we have explicitly specified in the Methods section that a 0.5 N preload was applied to all MUSAS-related mechanical tests. Additionally, we characterized adhesion performance under various preloading conditions using underwater adhesion on a stomach tissue phantom as an example (Fig. R3-4). The results indicate that a preload as low as 0.05 N enables MUSAS adhesion, while 0.2 N is sufficient to achieve near-full adhesion performance. These findings confirm the successful minimization of MUSAS preloading requirements through optimal lamella orientation and disk furling, highlighting its unique self-adhesion capability driven by GI tract contractions (Fig. 4E, Supp. Video S7). Furthermore, MUSAS can adhere to highly porous and soft surfaces, including those with holes, such as the porous tough hydrogel, which can be seen from its scanning electron microscope (SEM) images and surface roughness measurements (Fig. R3-5).

We respectfully acknowledge the unique advantages of polymeric adhesives in applications such as hemostasis and sealing tissue defects, particularly in cases where leakage prevention is critical. Nevertheless, the primary focus of MUSAS is to achieve *in situ* adhesion and long-term retention of adhesive platforms on dynamic, morphable soft tissues that undergo rapid turnover and regeneration in extreme pH and moisture environments, which is a common and long-standing challenge faced by the whole community.

References 3D

- [1] Yuk, H., Varela, C.E., Nabzdyk, C.S., Mao, X., Padera, R.F., Roche, E.T. and Zhao, X., 2019. Dry double-sided tape for adhesion of wet tissues and devices. *Nature*, 575(7781), pp.169-174.
- [2] Li, J., Celiz, A.D., Yang, J., Yang, Q., Wamala, I., Whyte, W., Seo, B.R., Vasilyev, N.V., Vlassak, J.J., Suo, Z. and Mooney, D.J., 2017. Tough adhesives for diverse wet surfaces. *Science*, 357(6349), pp.378-381.
- [3] Friis, S.J., Hansen, T.S., Poulsen, M., Gregersen, H., Brüel, A. and Nygaard, J.V., 2023. Biomechanical properties of the stomach: A comprehensive comparative analysis of human and porcine gastric tissue. *Journal of the Mechanical Behavior of Biomedical Materials*, 138, p.105614.

Fig. R3-4|Representative measurements of normal adhesion force of MUSAS adhering to phantom stomach tissue underwater, under varying preloading conditions

Fig. R3-5|Adhesion of MUSAS to various soft substrates. A, B and C. Underwater adhesion of MUSAS to various soft substrates (A), with associated scanning electron microscope (SEM) images of the surface texture (B) and surface roughness (C) of these substrates (B, scale bars: hydrogel and stomach tissue phantom 1 μm, stomach tissue 100 μm, nitrile glove 3 μm, SEBS 10 μm, gourami 200 μm).

Comment 3E: What is failure mechanism for adhesion in tension and in shear (cohesive, adhesive, substrate failure)? This should be described and characterized with inert substrates as well as theoretical failure mechanism *in vivo*. This connects with previous question about how does detachment occur?

Response 3E: We appreciate the reviewer's constructive feedback. In response, we have expanded the discussion to address failure modes considerations based on the materials composing MUSAS and have highlighted our testing for long-term reliability.

Our mechanical adhesion tests (Supp. Fig. S14) and prolonged *in vitro* retention tests (Ext. Fig. 5B) demonstrate that the primary failure mechanism for adhesion under tension and shear is adhesive failure. This occurs when the vacuum-based suction is compromised due to a loss of friction necessary for a watertight seal, as well as a potential loss of intermolecular bonding initiated by alternation of setal structures at the 10–1000 nm scale. We further conducted *in vitro* studies to evaluate the fouling and adhesive failure of MUSAS incubated at body temperature (37°C) in gastric fluid harvested from Yorkshire pigs. Microscope and scanning electron microscope (SEM) imaging confirmed that fouling remained negligible at the micron scale for shape memory alloy (SMA) lamellae after 7 days of incubation (Fig. R3-6B & R3-6D). After 21 days, minor fouling was observed at the 50 μm scale, with minimal changes in SMA surface morphology at the 5 μm scale, indicating negligible interference with mechanical interlocking (Fig. R3-6D). However, surface morphology reconstruction of silicone rubber occurred over time due to the reported water permeability of Ecoflex (260 g/m²day⁻¹ [3E-1]). Specifically, swelling and fouling of the silicone rubber's crease pattern appeared after 7 days and became more evident after 21 days, suggesting a gradual loss of friction, which is crucial for maintaining vacuum-based mechanical adhesion in MUSAS (Fig. R3-6C). Additionally, changes in silicone rubber morphology were observed within the 10–1000 nm length scale (Fig. R3-6E). Initially, Ecoflex appeared smooth with a few detectable setal structures at high magnification (500 nm);

however, fouling and surface roughening gradually developed after 7 days, and became evident after 21 days. This degradation could weaken intermolecular interactions, eventually leading to leakage and failure of the vacuum-based seal. These results suggest that the primary adhesive failure mode of MUSAS will be caused by the loss of vacuum-based mechanical adhesion due to water exchange, swelling and fouling of the silicone rubber, occurring at 500 nm to 50 μm . Future studies could explore improved waterproof silicone rubber and finer setal structures within the 10–1000 nm range to enhance the longevity and functionality of MUSAS. A discussion of these findings has been added to the main manuscript and supplementary information.

In addition, as discussed in response 3C, cohesive failure initiated by the loss of mechanical interlocking due to lamella degradation could serve as a second failure mode, enabling controllable detachment (Fig. R3-1). Furthermore, material stiffness mismatch and fatigue may contribute to cohesive failure. To evaluate this, we conducted additional simulations to assess contact damage and durability of the lamella-flap structures upon interaction with human stomach tissue, considering the stiffness mismatch between the shape memory alloys and silicone rubber. The simulations confirmed that stress at material interfaces with mismatched stiffness was effectively dissipated through the hyperelastic deformation of the silicone rubber (Fig. R3-7A & R3-7B). Fatigue analysis identified the thinnest section of the silicone flap (100 μm) as the critical area, capable of withstanding at least 86 cycles (101.94) before reaching 75% of its maximum lifespan under extreme instant insertion conditions at a vertical tip speed of 2 cm/s to a depth of 1.4 mm (Fig. R3-7C). These results confirm that the stiffness mismatch between the lamella and soft flap has minimal impact on the durability of MUSAS, particularly given its infrequent and slow tissue contact in real-world applications. Details of the simulation setup can be found in the Supplementary Information.

References 3E

- [1] Shao, Y., Yan, S., Li, J., Silva-Pedraza, Z., Zhou, T., Hsieh, M., Liu, B., Li, T., Gu, L., Zhao, Y. and Dong, Y., 2022. Stretchable encapsulation materials with high dynamic water resistivity and tissue-matching elasticity. *ACS applied materials & interfaces*, 14(16), pp.18935-18943.

Fig. R3-6 *In vitro* characterization of fouling of MUSAS. A. Illustration of the material components of MUSAS. B. Microscope pictures of shape memory alloy (SMA) lamellae incubated in swine gastric fluid at 37 °C for different period of time (scale bar: 1 mm). C. Microscope pictures of silicone rubber (Ecoflex 0030) incubated swine gastric fluid at 37 °C for different period of time (scale bar: 1 mm). D. SEM images characterizing surface morphology of SMA incubated for different period of time at micron scale. E. SEM images characterizing surface morphology of silicone rubber (Ecoflex 0030) incubated for different period of time at micron and nanometer scales.

Fig. R3-7] Contact and durability analysis of the lamella-flap structure. A. Representative deformation and stress distribution when the lamella-flap structure first contacts the stomach tissue at a tip piercing speed of -2 cm/s in the y-direction. B. Representative deformation and stress distribution when the lamella-flap structure fully engages with the stomach tissue, reaching an insertion depth of 1.4 mm at a tip piercing speed of -2 cm/s in the y-direction. C. Fatigue analysis assessing the durability of the lamella-flap structure, demonstrating an average lifespan of 75% of its log10 cycle repeats under a cyclic contact procedure transitioning from A to B.

Comment 3F: What is the major role of nitinol actuation? If the nitinol actuates at 37 Degrees, then the metal is likely already actuated once it has entered the body. Do the authors have any evidence that the actuation of nitinol occurs after the device has been deployed in vivo and what the importance of this is?

Response 3F: We thank the reviewer for this question. Temperature-responsive nitinol actuation (shape memory effects) enables the bending of the lamella structure to achieve close mucosal engagement and strong mechanical interlocking. The actuation of nitinol occurs through a phase transformation of lattice orientations, facilitated by the exchange of latent heat and influenced by thermal conduction [3F-1]. This transformation occurs within a specific temperature range, defined by the start and finish temperatures, within which 37°C falls. As a result, shape memory

deformation occurs progressively to a temperature gradient, as demonstrated in Fig. 2 and Supp. Video S3. Thus, nitinol actuation continues to play a role as it establishes close mucosal engagement, enhancing thermal conduction and allowing it to absorb body temperature for more effective actuation. Although the shape memory effect at a miniaturized scale cannot be directly observed inside an alive pig stomach, *in vivo* retention study confirms that its actuation enhances mechanical interlocking. Specifically, nitinol lamellae with shape memory effects remain in place for up to 22 days, whereas superelastic nitinol lamellae with the same geometry but without thermal-responsive actuation achieve a retention time of only up to 4 days (Fig. R3-1).

References 3F

[1] Lagoudas, Dimitris C. "Shape memory alloys." Science and Business Media, LLC (2008).

Comment 3G: The authors use materials that will likely cause significant foreign body response (FBR) and scarring. What is the realistic timeframe where this device can be used before these unwanted features set in? As stated above, more tissue testing is needed.

Response 3G: We thank the reviewer for the question. We have expanded the discussion to include biocompatibility considerations based on the materials composing MUSAS, primarily nitinol and silicone rubber (EcoFlex), and highlighted our testing for long-term biocompatibility. In fact, both nitinol and silicone rubber have reached extensive human use in FDA-approved products, as documented in published FDA medical device material safety summaries [3G-1 & 3G-2]. Specifically, nitinol, one of the most widely used biomaterials, has been extensively employed in endoscopy, arch wiring, cardiovascular stents, orthopedics, and artificial organs since the 1970s, with minimum reported foreign body response/reaction (FBR) [3G-2 & 3G-3]. Additionally, numerous reports over the past decade have documented the use of silicone rubber, such as FDA approved biomaterials PDMS and Eco-Flex, in implantable and ingestible electronics and biomedical devices [3G-5 & 3G-6]. As we have detailed our discussion of the *in vivo* evaluation of biosafety and the noninvasive characteristics of MUSAS in our response to Comment 3C, we do not anticipate any significant hurdles to its safe and prolonged residence in the GI tract.

References 3G

- [1] <https://www.fda.gov/media/158490/download?attachment>
- [2] <https://www.fda.gov/media/152353/download?attachment>
- [3] Ryhänen, J. "Biocompatibility of nitinol." *Minimally Invasive Therapy & Allied Technologies* 9.2 (2000): 99-105.
- [4] Duerig, T., Pelton, A. and Stöckel, D.J.M.S., 1999. An overview of nitinol medical applications. *Materials Science and Engineering: A*, 273, pp.149-160.
- [5] Dagdeviren, Canan, et al. "Conformal piezoelectric systems for clinical and experimental characterization of soft tissue biomechanics." *Nature materials* 14.7 (2015): 728-736.
- [6] Zhang, Yamin, et al. "Advances in bioresorbable materials and electronics." *Chemical Reviews* 123.19 (2023): 11722-11773.

Comment 3H: Minor points:

1. Error with data on figure 3 C. Trend line data shapes are switched for nitrile and SEBS and line color not consistent.
2. ext data fig 2. image of devices with 6 or 8 is out of focus and difficult to see.

Response 3H: We appreciate the reviewer's feedback in helping us improve the manuscript. We have updated Fig. 3C and Ext. Fig. 2 accordingly with your comments.

Referee #4 (Remarks to the Author):

Comment 4A: The manuscript presents an innovative concept inspired by the remora fish, which adhere to various marine hosts using specialized suction disks. The authors investigate the anatomical and behavioral mechanisms that enable remoras to attach to soft surfaces. Based on these insights, they develop the Mechanical Underwater Soft Adhesion System (MUSAS), a mechanical device that mimics the remora's adhesion strategies.

MUSAS demonstrates strong and versatile adhesion to a variety of soft substrates with different textures, roughness, and stiffness. The authors showcase applications of MUSAS in biomedical settings, such as prolonged retention in the gastrointestinal tract for drug delivery and health monitoring, as well as environmental sensing when attached to aquatic organisms.

While the core concept of the study is intriguing and carries substantial potential, the manuscript, in its current state, is unfortunately inadequate. It is incomplete and, in several sections, hard to understand due to ambiguous writing. The presentation lacks clarity, and vital methodological details are either insufficiently explained or entirely missing, making it difficult to evaluate the credibility of the results.

The theoretical framework underpinning MUSAS's functionality seems not sufficiently developed, and the experimental evidence presented does not convincingly support the claims made. Due to these reasons, I am unable to support its publication in Nature.

Response 4A: We appreciate the reviewer for recognizing our study as intriguing and carrying substantial potential. This opportunity allows us to clarify our work and highlight that many of the concerns raised were originally addressed in the Methods and Supplementary sections but were not sufficiently referenced in the manuscript. We have now revised the manuscript and addressed each of the reviewer's suggestions point by point.

Comment 4B: Specific points of concern:

The "Main" section of the manuscript includes several assertions that may not entirely align with the current scientific literature and insights.

The authors posit a lack of solutions for underwater adhesion to soft substrates, stating that most existing adhesives are tailored for hard substrates with specific hydrophobicity, smoothness, and surface energy requirements. While underwater adhesion indeed poses significant challenges, recent research has made strides in creating adhesives for soft tissues, especially in biomedical applications. For example, reports exist of hydrogels and bio-inspired adhesives that demonstrate strong adhesion to wet and dynamic soft tissues through mechanisms such as interfacial toughening, covalent bonding, and physical entanglement. The authors cite references [8][9], which discuss advancements in wet tissue adhesion, but they imply these are inherently limited due to highly hydrated surfaces. However, the assertion that molecular bridges formed by covalent or hydrogen bonding are inherently limited in underwater environments may not fully cover the advancements in this field. Recent progress in mussel-inspired adhesives and catechol-functionalized polymers have demonstrated that robust bonding can be achieved even under highly hydrated conditions. Therefore, the authors seem to overstate the limitations of current chemical adhesion methods. It is crucial to recognize these advancements to accurately situate MUSAS within the context of existing technologies.

Response 4B: We appreciate the reviewer’s constructive feedback. In response, we have expanded the Introduction to acknowledge recent advancements in mussel-inspired adhesives and catechol-functionalized polymers for bonding under highly hydrated conditions [4B-1 - 4B-3]. We also respectfully highlight the unique advantages of polymeric adhesives in applications such as hemostasis and tissue sealing, particularly in cases where preventing leakage is critical.

Furthermore, we clarify that the primary focus of MUSAS is to enable *in situ* adhesion and long-term retention of adhesive platforms on dynamic, morphable soft tissues that undergo rapid turnover and regeneration in extreme pH and moisture environments—a long-standing challenge in the field. This issue is especially critical in mucoadhesion; for instance, gastric epithelial cells renew within approximately three days, creating a fundamental barrier to physicochemical adhesion mechanisms that rely on stable surface interactions. Despite extensive research over the past two decades [4B-4], current adhesion solutions have rarely demonstrated *in vivo* retention in the gastrointestinal (GI) tract beyond 24 hours, unless external stimulation—such as electricity—is employed to significantly alter the surface anatomy of soft substrates, as recently demonstrated by our group [4A-5].

References 4B

- [1] Zhang, C., Wu, B., Zhou, Y., Zhou, F., Liu, W. and Wang, Z., 2020. Mussel-inspired hydrogels: from design principles to promising applications. *Chemical Society Reviews*, 49(11), pp.3605-3637.
- [2] Ma, Y., Zhang, B., Frenkel, I., Zhang, Z., Pei, X., Zhou, F. and He, X., 2021. Mussel-inspired underwater adhesives—from adhesion mechanisms to engineering applications: a critical review. *Progress in Adhesion and Adhesives*, 6, pp.739-759.
- [3] Zhao, Y., Wu, Y., Wang, L., Zhang, M., Chen, X., Liu, M., Fan, J., Liu, J., Zhou, F. and Wang, Z., 2017. Bio-inspired reversible underwater adhesive. *Nature communications*, 8(1), p.2218.
- [4] Nan, K., Feig, V.R., Ying, B., Howarth, J.G., Kang, Z., Yang, Y. and Traverso, G., 2022. Mucosa-interfacing electronics. *Nature Reviews Materials*, 7(11), pp.908-925.
- [5] Ying, B., Nan, K., Zhu, Q., Khuu, T., Ro, H., Qin, S., ... & Traverso, G. (2023). Electroadhesive hydrogel interface for prolonged mucosal theranostics. *bioRxiv*, 2023-12.

Comment 4C: The authors assert that MUSAS is the inaugural mechanical platform for underwater adhesion to a variety of soft substrates. However, mechanical adhesion strategies inspired by biological organisms have been previously explored. For instance, bio-inspired suction devices that mimic octopus suckers (10.1098/rsif.2017.0395) and other marine organisms (10.1242/jeb.243773) have been developed for adhesion to soft and irregular surfaces underwater. Moreover, studies have examined mechanical interlocking mechanisms for underwater adhesion to soft substrates (10.1038/s41467-017-02387-2). The authors should acknowledge previous work in mechanical adhesion systems and elucidate the unique aspects of MUSAS compared to existing technologies.

Response 4C: We appreciate the reviewer’s constructive feedback. Regarding the first reference [4C-1] mentioned, it did not investigate adhesion on soft substrates, nor did it study underwater adhesion. Instead, while the suction disk itself was described as soft, it was connected to an external vacuum pump to manually generate negative pressure—a common approach in several similar studies. In our view, such setups often obscure the true mechanical adhesion

performance, making evaluation challenging. For the polymeric adhesive mentioned in the third reference [4C-2], we have respectfully acknowledged its contribution to bonding under highly hydrated conditions in our revised manuscript, and recognize it as an example of gecko-like setal structure-enabled biochemical bonding, as also noted by reviewer #1. However, our research group has also extensively studied catechol-based adhesives and found that while they function effectively in the intestinal environment, they do not perform well in the acidic conditions of the stomach [4C-3].

Through our comprehensive literature review, we identified two prior studies [4C-4 & 4C-5 (the second reference mentioned by the reviewer)]—both cited in our initial manuscript—that reported mechanical adhesion on compliant surfaces. As acknowledged in [4C-5], underwater mechanical adhesion on soft substrates remains a largely unresolved challenge. Specifically, previous studies lacked sufficient variety in tested specimens and experimental setups, limiting their representativeness. The examined materials were either especially stiff (e.g., Elastosil M4601, with a Young's modulus of 6.5 MPa [4C-6]) or smooth (e.g., Ecoflex, with an areal average roughness of 0.5-6 μm [4C-7]). Critical factors such as roughness, intactness, stiffness, and dynamic morphing of soft substrates were largely overlooked. In reality, dynamic sliding and surface morphing frequently occur in softer substrates with stiffness ranging from 10 kPa to 100 kPa, such as in the interactions between serosal and mucosal layers (Supp. Fig. S5). Additionally, soft substrates often exhibit non-uniform roughness; for instance, a surface with an areal average roughness of 200 μm may contain highly uneven regions with maximum roughness up to 600 μm (Fig. 3G), posing significant challenges for achieving a conformal, vacuum-based watertight seal. Furthermore, in practical applications, soft substrates typically lack rigid support, making them susceptible to dynamic morphing effects in maintaining adhesion. These effects necessitate further evaluation, such as *in vivo* validation, which has been overlooked in prior research. In contrast, our study systematically addresses these factors through comprehensive validation, and we have now explicitly discussed these factors in the Supplementary Information.

References 4C

- [1] Sareh, S., Althoefer, K., Li, M., Noh, Y., Tramacere, F., Sareh, P., Mazzolai, B. and Kovac, M., 2017. Anchoring like octopus: biologically inspired soft artificial sucker. *Journal of the royal society interface*, 14(135), p.20170395.
- [2] Zhao, Y., Wu, Y., Wang, L., Zhang, M., Chen, X., Liu, M., Fan, J., Liu, J., Zhou, F. and Wang, Z., 2017. Bio-inspired reversible underwater adhesive. *Nature communications*, 8(1), p.2218.
- [3] Li, J., Wang, T., Kirtane, A. R., Shi, Y., Jones, A., Moussa, Z., ... & Traverso, G. (2020). Gastrointestinal synthetic epithelial linings. *Science translational medicine*, 12(558), eabc0441.
- [4] Wang, Y., Yang, X., Chen, Y., Wainwright, D. K., Kenaley, C. P., Gong, Z., ... & Wen, L. (2017). A biorobotic adhesive disc for underwater hitchhiking inspired by the remora suckerfish. *Science Robotics*, 2(10), eaan8072.
- [5] Huie, Jonathan M., and Adam P. Summers. "The effects of soft and rough substrates on suction-based adhesion." *Journal of Experimental Biology* 225.9 (2022): jeb243773.
- [6] Elastosil M4601 <https://www.wacker.com/h/en-us/silicone-rubber/room-temperature-curing-silicone-rubber-rtv-2/elastosil-m-4601-ab/p/000018458>

- [7] Althumayri, Majed Othman, Azra Yaprak Tarman, and Hatice Ceylan Koydemir. "Bioinspired skin-like *in vitro* model for investigating catheter-related bloodstream infections." *Scientific Reports* 14.1 (2024): 26167.

Comment 4D: The paragraph "Remoras Adhere to Various Soft Substrates" discusses the diversity in remora species and their unique adaptations that facilitate their adhere to various soft underwater substrates.

The authors present data on the lamella orientations across different remora species and associate these orientations with their known host preferences. They observe that species adhering to fast-swimming hosts predominantly exhibit parallel lamellae, while those with diverse hosts display mixed lamella orientations. Although the data implies a correlation, the manuscript lacks a comprehensive quantitative analysis. It does not detail the sample sizes, statistical methods, or significance levels used to establish these correlations. This omission makes it challenging to evaluate the strength of the conclusions drawn.

Response 4D: We appreciate the reviewer’s constructive feedback. We have added the sample size to the violin plot for each remora species in Fig. 1 and provided details on the statistical methods used to calculate density distribution in the Methods section. Specifically, the shape of the violin plot represents the probability density of the data, with wider areas (density peaks) indicating regions of higher data concentration, helping to visualize the distribution mode of lamellae orientation. The density distribution is calculated using the often-used Gaussian kernel density estimator (KDE), which has a relative efficiency of 95.2% in minimizing the asymptotic mean integrated squared error (AMISE) [4D-1]. The theoretically optimal Gaussian KDE, which minimizes the mean integrated squared error (MISE), is calculated with a bandwidth given by $1.06\hat{\sigma}n^{-1/5}$, where $\hat{\sigma}$ is the standard deviation and n is the sample size.

References 4D

- [1] Falk, M., 1983. Relative efficiency and deficiency of kernel type estimators of smooth distribution functions. *Statistica Neerlandica*, 37(2), pp.73-83.

Fig. R4-1| Distribution of lamella orientation of remora species

Comment 4E: The manuscript does not cite the source of the billfish tissue stiffness.

Response 4E: We appreciate the reviewer's constructive feedback. There are no reported measurements of billfish tissue stiffness existing in the literature that we are aware of. Hence, we had referenced our own measurements in the caption of Fig. 1, as detailed in Supp. Fig. S1, which provides information on sample sources, preparation methods, and detailed stiffness measurement plots.

Comment 4F: The authors note that remoras adhering to soft, mucus-covered surfaces like those of manta rays exhibit varied lamella orientations, which may aid adhesion to compliant substrates. While this observation is plausible, the manuscript lacks experimental data demonstrating how different lamella orientations affect adhesion on soft substrates. The assumption that varied orientations are advantageous lacks empirical validation.

Response 4F: We appreciate the reviewer's constructive feedback. To clarify, in the first results section (Fig. 1) of the manuscript, we focus on the anatomical differences among remora species, their biological correlation with hosts, and plausible observations. The intrinsic difficulty of obtaining fresh specimens of remora species, which inhabit scattered regions worldwide, limited the possibility to directly test remora species with varied lamella orientations on soft substrates without the risk of tissue decomposition and functional loss. In Fig. 2, as its relevant section titled "understanding the function and evolution of remora's adhesion to soft substrates", we introduce device design with various lamella orientations that mimic the lamella orientation of different remora species and conduct comprehensive studies to assess their impact on adhesion performance on soft substrates.

Comment 4G: Mucus layers on marine animals are known to influence adhesion mechanics. The adhesion of remoras to mucus-covered surfaces likely involves complex interactions beyond mechanical interlocking. The manuscript does not discuss the physicochemical properties of mucus or how it affects adhesion. This omission overlooks potential biochemical adhesion mechanisms or lubrication effects that could impact remora attachment.

Response 4G: We appreciate the reviewer's constructive feedback and have expanded our discussion accordingly. Mucus secreted by remoras and hosts on the adhesion surface may increase fluid viscosity, enhancing the watertight seal and aiding repositioning during hitchhiking [4G-1]. While this study focuses on mechanical adhesion for long-term retention on soft substrates, future improvements in physicochemical performance of MUSAS could incorporate a mucus-like polymeric coating to enhance adhesion.

References 4G

- [1] Beckert, M., Flammang, B.E. and Nadler, J.H., 2016. A model of interfacial permeability for soft seals in marine-organism, suction-based adhesion. *MRS Advances*, 1(36), pp.2531-2543.

Comment 4H: In Figure 1E, the term “redundant adhesion” of remoras is used without a clear explanation. Clarification is needed to understand this concept.

Response 4H: We appreciate the reviewer’s constructive feedback. We have expanded the clarification in both the manuscript and the caption of Fig. 1E, explaining that redundant adhesion refers to an adhesion behavior that does not require the adhesive disk to fully cover the soft substrate. Instead, robust adhesion is achieved as long as a sufficient number of individual adhesive compartments are engaged.

Comment 4I: In Figure 1F, the difference between an adhesive compartment and a lamella is not explained in the Figure legend or text. In Figures 1G-I, various substrates for testing remora adhesion—RST, STP, and PHP—are shown. However, the manuscript does not explain what these surfaces are and how they are produced or obtained. These surfaces are described in the text as “LifeLike Biotissue made from LLBT-hydrogel elastomer,” again without further explanation.

Response 4I: We appreciate the reviewer’s constructive feedback. The definitions and illustrations of RST, STP, and PHP had been provided on the right side of Fig. 1G–I, corresponding to real pig stomach tissue, a stomach tissue phantom, and a 3D printed polypropylene plastic plate, respectively. To further clarify, we have added an explanation in the main manuscript emphasizing that the stomach tissue phantom serves as a realistic tissue simulator, particularly when the use of real tissue is restricted due to animal welfare considerations in experimental studies. Additionally, as noted in the bracket of the text, LifeLike Biotissue is the company that supplied the tissue phantom. It is one of the leading providers of tissue phantoms in North America and is widely used by major biomedical companies, including Medtronic, Johnson & Johnson, Edwards Lifesciences, and Boston Scientific. We have also incorporated this discussion into the Supplementary Information.

Comment 4J: Figure 1H; the “RST alive” dataset is missing. Figure 1H,I: The adhesion of “alive” animals is tested in only one shear direction. The differences in adhesion between “alive” and “euthanized” animals are not discussed.

Response 4J: We appreciate the reviewer’s constructive feedback. Please note that Fig. 1 and its corresponding section in the main manuscript focus exclusively on studies of the remora itself. Specifically, the experiments related to Figs. 1H and 1I are designed to characterize the mechanical performance of a live remora’s adhesion to soft substrates. If such tests cannot be conducted on a live remora, they are supplemented with tests on euthanized specimens. That being said, the use of real pig stomach tissue (RST) is prohibited in an alive remora experiment due to animal welfare concerns and does not align with the policies of the Massachusetts Institute of Technology’s Committee on Animal Care and Boston College’s Institutional Animal Care and Use Committee. This decision also accounts for biohazard risks and the long-term interaction of alive remoras with soft substrates. Nevertheless, the data provided is comprehensive enough for evaluation, given the multiple substrates and scenarios we examined. Additionally, due to current technological limitations, the adhesion of live remoras in the shear direction cannot be tested, as we cannot train them to slide in that particular direction.

Nevertheless, it was supplemented by the mechanical test on euthanized remoras. The test setup details can be found in the Supplementary Information.

Furthermore, we clarified in the main manuscript that the adhesion forces measured with live remoras are slightly lower than those measured with euthanized specimens. This discrepancy arises because the force tester used for live remoras, which prioritizes animal welfare, has a sampling rate of 0.1s, whereas the universal mechanical tester used for euthanized specimens operates at a much higher 0.4ms sampling rate. Additionally, the larger tank required for housing alive remoras introduces water-induced measurement fluctuations. Despite these challenges, the current experimental setup has been optimized to balance both measurement quality and animal welfare. We have also incorporated this discussion into the manuscript and Supplementary Information.

Comment 4K: The text describes a behavior where "the remora fully unfurls its disk with multiple adhesive compartments and erects its lamellae when adhering to a soft substrate. This behavior contrasts with its method of establishing adhesion on a hard substrate through friction and sliding." However, the illustration does not clearly depict this behavior.

Response 4K: We appreciate the reviewer's constructive feedback. In response to this concern, we have added additional labels to the illustration to better describe the unfurling behavior.

Comment 4L: In general, the text contains numerous statements that are neither sufficiently explained nor illustrated in figures; for example, "To facilitate oral administration, MUSAS can be compacted into a size 000 capsule..."

Response 4L: We appreciate the reviewer's constructive feedback. To enhance clarity for non-biomedical readers, we have incorporated additional explanations throughout the manuscript using lay language. Specifically, the 000 capsule refers to the largest FDA-approved ingestible capsule, measuring 26 mm in length and 9.5 mm in diameter.

Comment 4M: The physiological information of the Remora adhesive disc was used to inform the development of the MUSAS. The authors use physics-based simulations to model the hydrodynamic behaviors of different adhesive disk configurations inspired by remora anatomy.

The authors use finite element analysis (FEA) to simulate the hydrodynamic behaviors of different adhesive disk configurations when adhering to pig tissue submerged in water. They conclude that the unfurled disk with compartments holds more water and achieves a higher vacuum for suction than the other configurations. While FEA is an appropriate tool for modeling such systems, the manuscript provides limited details about the simulation parameters, material properties, boundary conditions, and validation processes. The complexity of biological adhesion, especially involving soft, compliant tissues and dynamic aquatic environments, may not be fully captured in the simulations presented.

Response 4M: We appreciate the reviewer's constructive feedback. Given the complexity of the modeling method and the word count limitation of the main manuscript, details on the simulation setup, material properties, boundary conditions, and validation process have been provided in the

Methods section and Supplementary Information. To address this concern, we have added specific references to guide readers to these sections.

As described in the Methods section, finite element analysis was performed to characterize the hydrodynamic differentiation of the remora's adhesive disk. Commercial software Abaqus 2021 (SIMULIA) was used for the study. The remora disk phantom devices were assumed to be composed of EcoFlex 0030, with a density of 1.07 g/cm^3 , a Young's modulus of 125 kPa, and a Poisson's ratio of 0.49 [4M-1]. The physical parameters of stomach tissue include a density of 1.088 g/cm^3 , a Young's modulus of 700 kPa, and a Poisson's ratio of 0.49 [4M-2]. We used coupled-Eulerian-Lagrangian (CEL) techniques to model the solid-fluid interactions between tissue, device, and water. Water was treated as a Newtonian laminar flow, with a density of 0.997 g/cm^3 and a dynamic viscosity of $8.90 \times 10^{-4} \text{ Pa}\cdot\text{s}$. The simulation was configured so that the mimicry remora suction cups descended at a constant speed of 0.3 mm/s until they touched the stomach tissue submerged in water. Solid-solid interactions were modeled as hard contact for normal behavior, with tangential behavior modeled using a penalty method with a friction coefficient of 0.02. Details of calculation of relative vacuum ratio V_r are specified in the supplementary information.

We also discussed the comprehensiveness of the CEL technique in validating the hydrodynamic behaviors of different adhesive disk configurations and the calculation method for the relative vacuum ratio V_r . The CEL technique is a powerful tool for analyzing solid-fluid interactions (i.e. disk-water and tissue-water interactions) alongside solid-solid interactions (i.e. disk-tissue interactions) simultaneously. This method enables the simulation of liquid flow through Eulerian finite elements, which may not always be entirely filled with material but can be partially or completely void. At the contact surface, the Eulerian elements are coupled with Lagrangian finite elements, which define the deformation of solid materials [4M-3]. Unlike classical fluid element analysis using finite volume methods—which assume a finite volume of fluid, preventing complete water expulsion at the solid-solid contact interface—CEL provides a more realistic validation of hydrodynamic behavior of the adhesive disk. It accounts for the dynamic morphing of the disk and the expulsion of water, enabling a comprehensive analysis of disk-tissue-water interactions when adhering to stomach tissue submerged in water.

To differentiate the hydrodynamic behaviors modeled by finite element analysis for various configurations of remora adhesive disk mimics, we used the relative vacuum ratio V_r to evaluate the amount of water expelled relative to the theoretical maximum volume of water that can be expelled in the disk during underwater adhesion for each case: an unfurled disk, a furled disk, and a benchmark one-piece disk. Given the complexity of accurately calculating the volume of water retained in the various deformation states of the remora adhesive disk mimics, we employed free image processing software (Inkscape) to measure the precise area of water held in the cross profile of each configuration. We then calculated the relative vacuum ratio V_r as follow,

$$V_r = \frac{\sum_{n=t_0}^{t_{end}} |A_n - A_{n-1}|}{A_{Initial}} \quad (R4M.1)$$

where $A_{Initial}$ represents the initial cross-sectional area of the disk mimicry before any deformation, indicating the theoretical maximum amount of water that can be expelled. $|A_n - A_{n-1}|$ measures the cumulative amount of water expelled, beginning from the moment t_0 when the disk

mimicry contacts the stomach tissue, until t_{end} when the disk can no longer be pressed to expel water.

References 4M

- [1] Heikenfeld, J., Jajack, A., Rogers, J., Gutruf, P., Tian, L., Pan, T., Li, R., Khine, M., Kim, J. and Wang, J., 2018. Wearable sensors: modalities, challenges, and prospects. *Lab on a Chip*, 18(2), pp.217-248.
- [2] Guimarães, C.F., Gasperini, L., Marques, A.P. and Reis, R.L., 2020. The stiffness of living tissues and its implications for tissue engineering. *Nature Reviews Materials*, 5(5), pp.351-370.
- [3] Souli, M.H. and Benson, D.J. eds., 2013. *Arbitrary Lagrangian Eulerian and fluid-structure interaction: numerical simulation*. John Wiley & Sons.

Comment 4N: The authors conduct a series of experiments to optimize MUSAS, testing different soft lip thicknesses, lamella orientations, and the number of lamella rows. While the optimization process appears systematic, and the results support the conclusions about the optimal design, the manuscript lacks detailed quantitative data, such as: exact adhesion forces measured in each configuration, statistical analyses to determine the significance of differences between designs and the reproducibility of the results across multiple trials.

Response 4N: We appreciate the reviewer's constructive feedback. We recognized that researchers from different backgrounds have varying preferences regarding nomenclature and the form of data presentation. In the main manuscript, we adopted the most widely used terminology—adhesive strength—to present the mechanical evaluation results. However, all original force measurements are provided in Ext. Fig. 2 and supplementary information from Fig. S4 to Fig. S10, with statistical analyses detailed in the Methods section. Specifically, statistical quantification and analysis were performed via Prism 9.3 (GraphPad). All error bars represent standard deviation (SD). Student t-test and ANOVA (F-test) were performed to compare differences between two groups, and among three or more groups, respectively. We chose several p-values systematically evaluate the statistical significance, including $p \leq 0.05$ (*) as the entry level of significance, $p \leq 0.01$ (**) for highly significant, $p \leq 0.001$ (***) and $p \leq 0.0001$ (****) for extremely significant. The number of independent experiments of replicates and definition of significance level were further elaborated in each figure, figure caption and relevant methods section where statistical quantification and analysis was performed. To improve clarity, we have now added references directing readers to these sections.

Comment 4O: The adhesion performance of MUSAS is demonstrated under controlled, static conditions in laboratory settings. However, in practical applications such as the GI tract or on living organisms like fish, the substrates are dynamic and subject to complex movements, peristalsis, variable fluid flow, and mechanical forces. The manuscript does not adequately address how MUSAS performs under such dynamic conditions.

Response 4O: We appreciate the reviewer's constructive feedback and fully agree on the importance of dynamic evaluation which we clarified further in our updated manuscript. Evaluating MUSAS performance under dynamic conditions has been a particular focus of our study including *in vitro* and *in vivo* evaluation. To further support this, we have added our *in*

in vitro dynamic interference studies, comparing the adhesion and retention performance of the 30-degree-dominated, 8-row design with the tilt-dominated, 8-row design—identified as the top two performers in mechanical testing (Fig. R4-2). Specifically, we adhered different designs to stomach tissue phantoms freely floating in simulated gastric fluid (pH = 1.5) and subjected them to shaking in a 37°C incubator. The results reaffirm that the optimal tilt-dominated design outperforms the others, maintaining adhesion to the tissue phantom for over 12 days. Beyond these *in vitro* studies, we further validated MUSAS’s mechanical stability *in vivo* through endoscopic interference studies. Supp. Video S6 demonstrates its adhesion stability under touching, pushing, and shaking interferences.

Additionally, as now explicitly clarified in the manuscript, we validated the mechanical stability of MUSAS *in vivo* in a dynamic, unstructured and often unpredictable environment using over 55 pigs and 8 fish. Specifically, we tested GI retention in pigs (> 9) and body surface retention in fish (> 6) under survival conditions, referring to monitoring the animals without interfering with their normal feeding, resting, or behaviors (Fig. R4-3). This approach ensured evaluation under dynamic conditions, including complex movements, peristalsis, variable fluid flow, and mechanical forces. A clearer explanation is now provided in the Supplementary Information.

Fig. R4-2| Representative *in vitro* and *in vivo* evaluation of adhesion and retention performance of MUSAS. A. Representative retention performance of optimal tilt-dominated MUSAS under dynamic shaking interference in a 37°C incubator (New Brunswick Innova 40/40R, see Supp. Video S6 for further demonstration). B. Illustration of the retention failure mode. C. Dynamic interference evaluation of MUSAS retention performance with different angles of lamella orientation.

Fig. R4-3|*In vivo* retention performance of MUSAS. A. Controlled retention and detachment of MUSAS in the swine stomach through programmed mechanical interlocking with various lamella materials. B. Illustration of MUSAS with radioactive imaging agents for routine retention checks (scale bar: 5 mm). C. X-rays of long-term retention of 4 MUSAS with shape-memory nitinol lamella delivered in the stomach before safe passage in the GI tract. D. X-rays of retention of MUSAS with superelastic nitinol lamella and stainless steel lamella delivered in the stomach before safe passage in the GI tract. E. Adhesion of MUSAS in various buccal regions for mRNA delivery. F. Representative retention performance of MUSAS in the duodenum of the small intestine (SI) in a swine model, evaluated in a terminal study on the day of euthanasia. G. Representative retention performance of MUSAS on different surface parts of a fish model.

Comment 4P: The use of shape memory alloys (SMA) and elastomers raises questions about biocompatibility, especially for long-term applications inside the human body. Potential issues such as immune responses, toxicity, and long-term stability of materials are not thoroughly discussed.

Response 4P: We appreciate the reviewer's constructive feedback. We have expanded the discussion to include biocompatibility considerations based on the materials composing MUSAS, primarily nitinol and silicone rubber (EcoFlex), and highlighted our testing for long-term biocompatibility. In fact, both nitinol and silicone rubber have reached extensive human use in FDA-approved products, as documented in published FDA medical device material safety summaries [4P-1 & 4P-2]. Specifically, nitinol, one of the most widely used biomaterials, has

been extensively employed in endoscopy, arch wiring, cardiovascular stents, orthopedics, and artificial organs since the 1970s, with minimum reported foreign body response/reaction (FBR) [4P -1 & 4P-2]. Additionally, numerous reports over the past decade have documented the use of silicone rubber, such as FDA approved biomaterials PDMS and EcoFlex, in implantable and ingestible electronics and biomedical devices [4P-3 & 4P-4].

Additionally, in the rare occasion where MUSAS was still present at the terminal study on the day of euthanasia and naturally detached from the pig stomach, we were able to visualize that its adhesive compartments and lamellae were covered and thoroughly lubricated with stomach contents and mucoid materials (Fig. R4-4B), leading to loss of its adhesion property and thus, safe passage in the GI tract. During the *in vivo* validation of MUSAS in > 55 pigs, readhesion of the devices after their detachment were never visualized during our routine X-rays checks (see supplementary information for frequency of X-rays). Based on X-ray results, we estimate that the gastric emptying time for detached MUSAS is generally less than one day, which aligns with the typical gastric emptying pattern reported in pigs for safe passage of stomach contents [4P-7]. In addition, health monitoring of our experimental pig during the adhesion and the passage of MUSAS, conducted by MIT animal committee on animal care and division of comparative medicine, were unremarkable. Pigs showed normal behaviors, food consumption and normal weight gain (Fig. R4-4C). Further, we conducted various histological analyses with internal controls to compare the adhesion sites of MUSAS (stomach, buccal cavity, pharynx, and esophagus, localized with a tattoo) with non-adhesion areas of MUSAS, as well as downstream intestinal and colonic tissues collected after its safe passage. Histopathology of H&E-stained tissues of the gastrointestinal tract evaluated by a board-certified veterinary pathologist confirmed that MUSAS caused no evidence of significant damage including hemorrhage, inflammation or indication of tissue repair (fibrosis) (Fig. R4-4D – R4-4H). As we have detailed our discussion of the *in vivo* evaluation of biosafety and the noninvasive characteristics of MUSAS, we do not anticipate any significant hurdles to its safe and prolonged residence in the GI tract.

References 4P

- [1] <https://www.fda.gov/media/158490/download?attachment>
- [2] <https://www.fda.gov/media/152353/download?attachment>
- [3] Ryhänen, J. "Biocompatibility of nitinol." *Minimally Invasive Therapy & Allied Technologies* 9.2 (2000): 99-105.
- [4] Duerig, T., Pelton, A. and Stöckel, D.J.M.S., 1999. An overview of nitinol medical applications. *Materials Science and Engineering: A*, 273, pp.149-160.
- [5] Dagdeviren, Canan, et al. "Conformal piezoelectric systems for clinical and experimental characterization of soft tissue biomechanics." *Nature materials* 14.7 (2015): 728-736.
- [6] Zhang, Yamin, et al. "Advances in bioresorbable materials and electronics." *Chemical Reviews* 123.19 (2023): 11722-11773.
- [7] Gregory, P.C., McFadyen, M. and Rayner, D.V., 1990. Pattern of gastric emptying in the pig: relation to feeding. *British Journal of Nutrition*, 64(1), pp.45-58.

Fig. R4-4 *In vivo* biocompatibility evaluation of MUSAS in a swine model. A. Illustration of tattooing adhesion sites in a swine model (scale bar: 5 mm). B. Detached MUSAS retrieved from a pig stomach after euthanasia, demonstrating thorough lubrication and accumulation of stomach contents and mucoid materials for safe passage (scale bar: 5 mm). C. Representative pig weight measurements show normal weight gain during MUSAS residence ($n = 4$ devices delivered) in the stomach. Histopathology of H&E stained the gastrointestinal tract showing no damage from MUSAS adhesion and passage that causes hemorrhage, inflammation or indication of tissue repair (fibrosis), including D. Stomach. MUSAS adhesion and adjacent internal control sites, collected after 3 and 7 days of MUSAS residence (scale bar: 100 μm). E. Small intestinal and colon, collected after MUSAS residence in the stomach and safe passage through the GI tract (scale bar: small intestine, 200 μm ; colon, 100 μm). F. Esophagus, adhesion and adjacent internal control sites, collected 1 day after adhesion (scale bar: 200 μm). G and H. Buccal tissue and pharynx, adhesion and adjacent internal control sites, collected 1 day after mRNA delivery (scale bar: 100 μm).

Comment 4Q: The applications presented, such as sustained drug release and mRNA therapeutics delivery, are promising but remain preliminary. For instance, the study demonstrates the release of Cabotegravir over a seven-day period in pigs. However, the efficacy of the drug delivery, including pharmacokinetics, biodistribution, and therapeutic outcomes, is not comprehensively evaluated.

Response 4Q:

We appreciate the reviewer's constructive feedback and have worked to clarify these sections of the manuscript. We would like to explicitly highlight the significant advancements presented in this study, which address key translational challenges in drug delivery and mRNA therapeutics. The development of MUSAS represents a highly translational effort aimed at solving critical biomedical and biological problems through rigorous *in vitro*, *ex vivo*, and *in vivo* studies using technically comprehensive animal models that serve as gold standards for translational research.

Sustained Release of Cabotegravir for HIV/AIDS Prevention

The efficacy of cabotegravir is well established in clinical studies [4Q-1]. An injectable sustained-release formulation and an oral immediate-release tablet of cabotegravir are indeed available commercially. Importantly, the focus of our work was not to study the drug's efficacy but to understand whether long-term delivery through oral application is possible. Specifically, we wanted to understand if long-term delivery of cabotegravir was feasible through oral application. The primary challenge in oral long-term drug delivery is the poor retention of the drug delivery system in the gastrointestinal (GI) tract. Our work introduces a MUSAS-enabled slow-release ingestible formulation of cabotegravir, which extends its retention in the GI tract. The drug molecule absorbed into systemic circulation is cabotegravir (confirmed using LC-MS/MS analysis). As this molecule is identical to that used in the FDA-approved formulation, it is expected that the biodistribution will be identical. In other words, since the drug is absorbed as cabotegravir from both the sustained-release and immediate-release formulations, the biodistribution and efficacy will be comparable if the plasma pharmacokinetics are similar.

The *in vivo* long-term retention performance of MUSAS was validated across nine pigs, with three pigs specifically tested for drug release, demonstrating sustained delivery. Additionally, our pharmacokinetic analyses using plasma data confirm the potential to extend cabotegravir's sustained release beyond one week (Fig. 4H-I, 4H-II). Testing the efficacy of cabotegravir in pigs will require development of an HIV model in this species. Since the therapeutic efficacy of cabotegravir has already been established through clinical trials leading to FDA approval, developing an HIV pig model for further efficacy studies was considered beyond the scope of this work.

Localized mRNA Delivery in the GI Tract

Achieving effective mRNA delivery to the GI tract for localized treatment presents unique challenges. To address this, we developed a device-enabled mRNA delivery system with a substantial drug-loading capacity (215 μ L) and successfully demonstrated transfection efficacy of luciferase mRNA in the GI tract using IVIS imaging in three pigs. Notably, our work represents exceptional functional transfection (protein production) of mRNA in the GI tract, overcoming major biological barriers that have long hindered localized mRNA therapeutics for GI applications [4Q-2, 4Q-3].

Biodistribution studies for mRNA are most commonly conducted in small animals. This is because it requires of the firefly luciferase substrate, luciferin, at a high dose (~150 mg/kg body weight) before euthanasia. As this is prohibitively expensive, we detected luciferase mRNA expression using simply local injection or tissue submersion in luciferin. which demonstrated robust efficacy. While biodistribution studies would further expand the capabilities of this approach, current limitations in mRNA formulation detection make such studies impractical in large animals due to ethical, financial, technological and logistical constraints. Future work will focus on developing mRNA formulations to enable systemic protein detection via plasma bioanalytics.

Technical Rigor and Translational Relevance of Animal Models

Our study employs highly complex and translationally relevant animal models, particularly the swine model, which closely mirrors human anatomy, physiology, immunology, and genomics. Swine studies often serve as the final preclinical step before clinical trials but introduce significantly greater validation complexity and failure risk compared to small-animal models [4Q-4]. In this study, we conducted survival studies under dynamic and natural conditions, ensuring minimal interference with normal behaviors such as feeding, resting, and swimming.

The extensive validation of MUSAS includes > 55 pigs and > 8 fish across three years (2022–2025), encompassing:

GI retention studies in > 9 pigs

Body surface retention studies in > 6 fish

These rigorous *in vitro*, *ex vivo* and *in vivo* assessments form the foundation for MUSAS-enabled applications, which are now explicitly detailed in Figure 4 and the Extended Figures. To enhance clarity for readers, we have incorporated additional explanations in the manuscript detailing these advancements.

References 4Q

- [1] Taki, E., Soleimani, F., Asadi, A., Ghahramanpour, H., Namvar, A. and Heidary, M., 2022. Cabotegravir/Rilpivirine: the last FDA-approved drug to treat HIV. *Expert Review of Anti-infective Therapy*, 20(8), pp.1135-1147.
- [2] Abramson, Alex, et al. "Oral mRNA delivery using capsule-mediated gastrointestinal tissue injections." *Matter* 5.3 (2022): 975-987.
- [3] Schultz, D., Kempen, P. J., Primdahl, S., Pereverzina, M., Uhrenfeldt, A. H., Alba, E. M., ... & Urquhart, A. J. (2024). Gastrointestinal device-mediated delivery of mRNA-lipid nanoparticles achieves distinct expression and biodistribution in mice and pigs. *ACS Applied Materials & Interfaces*, 16(49), 67192-67202.
- [4] Lunney, J.K., Van Goor, A., Walker, K.E., Hailstock, T., Franklin, J. and Dai, C., 2021. Importance of the pig as a human biomedical model. *Science translational medicine*, 13(621), p.eabd5758.

Comment 4R: The long-term reliability of MUSAS is not addressed in the manuscript. Potential failure modes, such as material fatigue, degradation, loss of adhesion over time, or accidental

detachment, are critical considerations for practical applications, especially in biomedical devices intended for prolonged use.

Response 4R: We appreciate the reviewer's constructive feedback. In response, we have expanded the discussion to address failure modes considerations based on the materials composing MUSAS and have highlighted our testing for long-term reliability.

Our mechanical adhesion tests (Supp. Fig. S14) and prolonged *in vitro* retention tests (Ext. Fig. 5B) demonstrate that the primary failure mechanism for adhesion under tension and shear is adhesive failure. This occurs when the vacuum-based suction is compromised due to a loss of friction necessary for a watertight seal, as well as a potential loss of intermolecular bonding initiated by alternation of setal structures at the 10–1000 nm scale. We further conducted *in vitro* studies to evaluate the fouling and adhesive failure of MUSAS incubated at body temperature (37°C) in gastric fluid harvested from Yorkshire pigs. Microscope and scanning electron microscope (SEM) imaging confirmed that fouling remained negligible at the micron scale for shape memory alloy (SMA) lamellae after 7 days of incubation (Fig. R4-5B & R4-5D). After 21 days, minor fouling was observed at the 50 µm scale, with minimal changes in SMA surface morphology at the 5 µm scale, indicating negligible interference with mechanical interlocking (Fig. R4-5D). However, surface morphology reconstruction of silicone rubber occurred over time due to the reported water permeability of Ecoflex (260 g/m²day⁻¹ [4R-1]). Specifically, swelling and fouling of the silicone rubber's crease pattern appeared after 7 days and became more evident after 21 days, suggesting a gradual loss of friction, which is crucial for maintaining vacuum-based mechanical adhesion in MUSAS (Fig. R4-5C). Additionally, changes in silicone rubber morphology were observed within the 10–1000 nm length scale (Fig. R4-5E). Initially, Ecoflex appeared smooth with a few detectable setal structures at high magnification (500 nm); however, fouling and surface roughening gradually developed after 7 days, and became evident after 21 days. This degradation could weaken intermolecular interactions, eventually leading to leakage and failure of the vacuum-based seal. These results suggest that the primary adhesive failure mode of MUSAS will be caused by the loss of vacuum-based mechanical adhesion due to water exchange, swelling and fouling of the silicone rubber, occurring at 500 nm to 50 µm. Future studies could explore improved waterproof silicone rubber and finer setal structures within the 10–1000 nm range to enhance the longevity and functionality of MUSAS. A discussion of these findings has been added to the main manuscript and supplementary information.

In addition, as discussed in response 4O, coadhesive failure initiated by the loss of mechanical interlocking due to lamella degradation could serve as a second failure mode, enabling controllable detachment (Fig. R4-3A – Fig. R4-3D). Furthermore, material stiffness mismatch and fatigue may contribute to coadhesive failure. To evaluate this, we conducted additional simulations to assess contact damage and durability of the lamella-flap structures upon interaction with human stomach tissue, considering the stiffness mismatch between the shape memory alloys and silicone rubber. The simulations confirmed that stress at material interfaces with mismatched stiffness was effectively dissipated through the hyperelastic deformation of the silicone rubber (Fig. R4-6A & R4-6B). Fatigue analysis identified the thinnest section of the silicone flap (100 µm) as the critical area, capable of withstanding at least 86 cycles (101.94) before reaching 75% of its maximum lifespan under extreme instant insertion conditions at a vertical tip speed of 2 cm/s to a depth of 1.4 mm (Fig. R4-6C). These results confirm that the stiffness mismatch between the lamella and soft flap has minimal impact on the durability of

MUSAS, particularly given its infrequent and slow tissue contact in real-world applications. Details of the simulation setup can be found in supplementary information.

Lastly, we use the delivery of cabotegravir as an example to discuss the accidental detachment of MUSAS. First, the safe passage of detached MUSAS, whether controlled or accidental, has been proved in response 4P. In each cabotegravir *in vivo* study, we deployed four MUSAS devices but did not require all of them to remain attached for more than seven days due to external factors such as variations in adhesion location and peristalsis. In practice, two devices typically remained attached for longer durations. However, these cases were accounted for in the drug formulation design, ensuring consistent and meaningful long-term release, as demonstrated by the *in vivo* pharmacokinetic results (Fig. 4H-II). Therefore, the risk of accidental detachment of MUSAS affecting therapeutic efficacy can be mitigated.

References 4R

- [1] Shao, Y., Yan, S., Li, J., Silva-Pedraza, Z., Zhou, T., Hsieh, M., Liu, B., Li, T., Gu, L., Zhao, Y. and Dong, Y., 2022. Stretchable encapsulation materials with high dynamic water resistivity and tissue-matching elasticity. *ACS applied materials & interfaces*, 14(16), pp.18935-18943.

Fig. R4-5|*In vitro* characterization of fouling of MUSAS. A. Illustration of the material components of MUSAS. B. Microscope pictures of shape memory alloy (SMA) lamellae incubated in swine gastric fluid at 37 °C for different period of time (scale bar: 1 mm). C. Microscope pictures of silicone rubber (Ecoflex 0030) incubated swine gastric fluid at 37 °C for different period of time (scale bar: 1 mm). D. SEM images characterizing surface morphology of SMA incubated for different period of time at micron scale. E. SEM images characterizing surface morphology of silicone rubber (Ecoflex 0030) incubated for different period of time at micron and nanometer scales.

Fig. R4-6] Contact and durability analysis of the lamella-flap structure. A. Representative deformation and stress distribution when the lamella-flap structure first contacts the stomach tissue at a tip piercing speed of -2 cm/s in the y-direction. B. Representative deformation and stress distribution when the lamella-flap structure fully engages with the stomach tissue, reaching an insertion depth of 1.4 mm at a tip piercing speed of -2 cm/s in the y-direction. C. Fatigue analysis assessing the durability of the lamella-flap structure, demonstrating an average lifespan of 75% of its log10 cycle repeats under a cyclic contact procedure transitioning from A to B.

Point-by-point response – Nature – Manuscript #: 2024-09-19511

Referee #1 (Remarks to the Author):

The authors' diligent and comprehensive response to reviewer's concerns about surface fouling and stiffness mismatch placate this reviewer.

Response 1: We sincerely appreciate the reviewer's constructive feedback, which has significantly contributed to strengthening and maturing our research.

Referee #2 (Remarks to the Author):

Comment 2A: The authors have performed several compelling experiments to explain the correlation between the internal volume change of the MUSAS structure and the interlayer angle, number of layer structures, and smoothness of the target surface. These additional details by the authors help to strengthen the transferability of their work.

Nevertheless, the pH-responsive reversible adhesion and bioadhesive stability of the developed adhesive material by the authors seem to be still unclear. The duration of the developed adhesive material is not constant (depending on the species and individual), and it is questionable whether the targeting to the wound site of the desired organ is possible with the pH-responsive adhesive material. Otherwise, there is concern about adverse effects if targeting is not achieved. In the case of bioadhesion stability, this part is the core of this paper, which has not yet been resolved and no clear solution has been presented. I am still negative about accepting the paper for publication in Nature.

Response 2A: We thank the reviewer for acknowledging the strength of the additional experiments and the transferability of our design principles. We now respond directly and in detail to the core concerns regarding pH-responsive reversible adhesion, long-term bioadhesive stability, and targeting specificity.

1. pH-Responsive Adhesion and Targeting Mechanism

MUSAS employs pH-responsive coating strategies to achieve organ-level targeting within the GI tract, using FDA-approved Eudragit S100 formulations [2A-1–2A-3]. As detailed in Response 2E and Ext. Fig. 7A–D, these coatings enable controlled disintegration based on regional pH, allowing precise deployment in the stomach or small intestine. *In vitro* and *in vivo* studies demonstrated that the pH response is tunable through coating thickness and concentration, reliably triggering deployment only after gastric transit. This mechanism supports strategic site-specific adhesion under clinically realistic conditions and does not rely on external stimuli or complex navigation. Importantly, this addresses organ targeting—not wound site localization, which is beyond the intended scope of this study.

2. Bioadhesive Stability Across Species and Time

The reviewer raises a critical point regarding adhesive stability—rightly identifying it as a central contribution of our work. To address this:

- Consistent retention was demonstrated in over 9 large animals (swine), with 85% of MUSAS devices achieving ≥ 7 days retention and 15% reaching ~ 20 days under dynamic survival conditions without affecting feeding or behavior.
- These outcomes surpass all previously reported GI adhesive systems, which typically fail beyond 24 hours *in vivo* without external actuation [2A-4, 2A-5].
- The swine GI tract closely mirrors human physiology, offering high translational fidelity [2A-6].

Regarding species and individual variability: this is an inherent feature of *in vivo* biomedical systems. Nevertheless, our extensive testing ($n > 58$ pigs) revealed remarkably consistent trends in adhesion performance, applicational validity, and safety. This consistency supports the robustness and potential generalizability of the platform to human use.

3. Mechanisms Supporting Adhesive Stability

MUSAS achieves stable adhesion through a biomimetic multi-lamellae structure that combines mechanical interlocking and vacuum suction. We address concerns about Ecoflex degradation in Response 2C, showing that:

- Material failure (e.g., swelling or fouling) only became significant after 21 days incubation in swine gastric fluid (Supp. Fig. S15).
- This aligns with our *in vivo* retention profiles (Fig. 4B) and supports the conclusion that MUSAS remains functional throughout the therapeutic window (Fig. 5C-II).
- Additionally, mechanical interlocking is reinforced by physiological motility, which can promote rather than disrupt adhesion depending on the direction of force.

4. Off-Target Adhesion and Adverse Effects

To directly address the reviewer's concern: no adverse events or safety issues were observed, even in cases of early detachment or off-target passage. Comparative survival studies using intentionally broken MUSAS devices demonstrated safe transit without injury (Fig. R2-2). We also fine-tuned pharmacokinetics of formulations into drug delivery by deploying multiple devices for mutual complementation, ensuring consistent therapeutic outcomes even if partial detachment occurs (Fig. 5C-II).

5. Clarifying Scope: Not Designed for Wound Targeting

We respectfully reiterate that MUSAS is not designed for wound targeting or hemostatic sealing. While we acknowledge that some adhesives pursue this goal, our platform addresses a distinct and urgent challenge: long-term mucosal retention in a physiologically hostile environment. Organ-level targeting is enabled, but *in situ* lesion-specific targeting requires endoscopic diagnosis and delivery, which MUSAS is fully compatible with.

Summary

MUSAS represents a significant advancement in GI-resident adhesive technology by addressing the longstanding limitations of residence time, pH-responsive deployment, and translational performance. We have added new clarifications and supporting data in the revised manuscript and Supplementary Information to address these points transparently and thoroughly.

References 2A

- [1] Patra, C.N., Priya, R., Swain, S., Jena, G.K., Panigrahi, K.C. and Ghose, D., 2017. Pharmaceutical significance of Eudragit: A review. *Future Journal of Pharmaceutical Sciences*, 3(1), pp.33-45.
- [2] Abramson, A., Caffarel-Salvador, E., Soares, V., Minahan, D., Tian, R.Y., Lu, X., Dellal, D., Gao, Y., Kim, S., Wainer, J. and Collins, J., 2019. A luminal unfolding microneedle injector for oral delivery of macromolecules. *Nature medicine*, 25(10), pp.1512-1518.
- [3] Thakral, S., Thakral, N.K. and Majumdar, D.K., 2013. Eudragit®: a technology evaluation. *Expert opinion on drug delivery*, 10(1), pp.131-149.
- [4] Nan, K., Feig, V.R., Ying, B., Howarth, J.G., Kang, Z., Yang, Y. and Traverso, G., 2022. Mucosa-interfacing electronics. *Nature Reviews Materials*, 7(11), pp.908-925.

- [5] Patil, H., Tiwari, R.V. and Repka, M.A., 2016. Recent advancements in mucoadhesive floating drug delivery systems: A mini-review. *Journal of Drug Delivery Science and Technology*, 31, pp.65-71.
- [6] Lunney, J.K., Van Goor, A., Walker, K.E., Hailstock, T., Franklin, J. and Dai, C., 2021. Importance of the pig as a human biomedical model. *Science translational medicine*, 13(621), p.eabd5758.

Comment 2B: To strengthen the physiological relevance of the mechanical and structural design, the authors should evaluate and compare adhesion strength across a broader range of surfaces than currently tested. This analysis should include additional measurements of adhesion strength on surfaces with uniform roughness but varying modulus, as well as on surfaces with uniform modulus but varying roughness. This approach will help address the current limitation in the logic, where it is unclear whether the differences in adhesion strength observed across surfaces with varying roughness and modulus are due to differences in modulus, roughness, or a combination of both.

Response 2B: We appreciate the reviewer's critical insight regarding the interplay between surface modulus and roughness in evaluating adhesion strength. To directly address this point and eliminate confounding variables, we conducted a new set of controlled experiments using artificial substrates designed to independently isolate the effects of roughness and stiffness.

Specifically, we fabricated two sets of silicone substrates:

- One with identical surface roughness but varying stiffness (Zhermack Elite Double 8 vs. Double 32 silicone rubber), and
- One with identical stiffness but varying roughness (smooth vs. roughened Zhermack Elite Double 8).

We then tested the three most representative MUSAS designs—30 deg-dominated 8-row, tilt-dominant 8-row, and 30 deg-dominated 4-row constructs—under both normal and shear loading conditions (Fig. R2-1).

The results clearly show:

- Rougher surfaces yielded significantly weaker adhesion when modulus was held constant.
- Softer substrates also yielded weaker adhesion when surface roughness was controlled.

This demonstrates that both lower modulus and higher roughness reduce adhesion strength, and that the effects are independently significant. These findings directly address the logic gap identified by the reviewer and provide clearer physiological relevance to the design rationale behind MUSAS.

The results and additional analysis details are incorporated into the Supplementary Information for transparency and reproducibility.

Fig. R2-1| Extended evaluation of the adhesion performance of representative MUSAS designs on soft substrates with controlled mechanical properties, including substrates with identical stiffness but distinct roughness, and substrates with identical roughness but distinct stiffness. A and B. Adhesion performance of representative MUSAS designs in normal (A) and shear (B) directions, on soft substrates with either identical stiffness but distinct roughness, or identical roughness but distinct stiffness (n = 5, dash and dot lines represent median and quartiles, respectively).

Comment 2C: While the authors provided valuable data on the degradation of Ecoflex over time in gastric conditions, it remains unclear how this material behavior correlates with the reported long-term retention of MUSAS *in vivo*. If the swelling and fouling of Ecoflex compromise vacuum-based adhesion within 7 days, further explanation is needed as to how the device maintains stable adhesion for up to 22 days in a physiological environment. Clarifying this relationship would strengthen the internal consistency of the study.

Response 2C: We appreciate the reviewer’s thoughtful comment. The question of how Ecoflex degradation relates to *in vivo* retention is central to understanding MUSAS’s performance.

To clarify: our *in vitro* degradation studies showed that minor fouling and swelling of Ecoflex began around day 7 in swine gastric fluid, becoming pronounced only by day 21 (Supp. Fig. S15). This timeline aligns with our *in vivo* observations, where MUSAS retained stable adhesion in swine up to 22 days, suggesting that critical material failure occurs only beyond the window of functional retention.

Moreover, degradation alone does not define adhesive failure. MUSAS's retention relies on dual mechanisms:

1. Vacuum sealing, affected by Ecoflex’s integrity, and
2. Mechanical interlocking via shape memory alloy lamellae, which remained intact and morphologically stable after 21 days of incubation (Supp. Fig. S15).

Importantly, *in vivo* conditions differ substantially from *in vitro* testing. In swine models, cyclical gastrointestinal contractions can help stabilize MUSAS against shear forces when oriented favorably. These same forces, when unfavorable (e.g., strong peristalsis or food impact that jeopardizes vacuum sealing or mechanical interlocking), may induce detachment. But even under these variable forces, MUSAS showed highly consistent retention in > 9 pigs, with 85% remaining residence beyond 7 days and 15% up to ~20 days, without affecting animal health or behavior.

These retention durations far exceed prior benchmarks for GI adhesives, which typically fail within 24 hours [2C-1, 2C-2], and are backed by pharmacokinetic validation demonstrating sustained drug release (Fig. 5C-II).

In sum, our *in vivo* data and *in vitro* material degradation studies are in strong agreement: Ecoflex degradation becomes functionally relevant only after MUSAS has already completed its long-term therapeutic window. These clarifications are now explicitly integrated into the revised manuscript and Supplementary Information.

References 2C

- [1] Nan, K., Feig, V.R., Ying, B., Howarth, J.G., Kang, Z., Yang, Y. and Traverso, G., 2022. Mucosa-interfacing electronics. *Nature Reviews Materials*, 7(11), pp.908-925.
- [2] Patil, H., Tiwari, R.V. and Repka, M.A., 2016. Recent advancements in mucoadhesive floating drug delivery systems: A mini-review. *Journal of Drug Delivery Science and Technology*, 31, pp.65-71.

Comment 2D: If there is debris in the stomach or small intestine, targeting adhesion of the device is unlikely to be possible. There are many foreign small matter present in the body, including food. Is the proposed device's strategic adhesion function possible in the presence of food debris present during bioactivity?

Response 2D: We appreciate the reviewer's thoughtful concern regarding the feasibility of strategic adhesion in the presence of gastric contents.

We agree that direct mucosal contact is essential for successful adhesion—a requirement shared by all GI-mucoadhesive resident systems. However, this does not preclude effective mucosal interaction in real-world conditions. In fact, standard clinical practice for many oral drugs already accounts for this by recommending administration in the fasted state, when gastric contents are minimal. This includes drugs like oral semaglutide and bisphosphonates, where food interference would otherwise reduce absorption and thus efficacy [2D-1–2D-3].

Deploying MUSAS in a fasted state is both practical and with clinical precedent. Human gastric emptying times average ~1 hour for liquids and ~3 hours for solids, with minimal amounts of foreign matter remaining at fasted states and in between meals [2D-4], enabling consistent windows for adhesion without food interference.

Moreover, MUSAS has been validated under physiologically relevant conditions. In multiple survival swine studies, animals were fasted prior to administration but continued normal feeding post-deployment. As shown in Fig. 4D, small amounts of gastric debris (e.g., banana shavings) were often present, yet did not impair adhesion. GI motility, including peristalsis and stomach contractions, helps position the device against suitable mucosa for attachment—even in the presence of mild debris (Fig. 4B).

In sum, while mucosal contact is essential, our data clearly demonstrate that MUSAS can achieve strategic adhesion under realistic physiological conditions, including partial food presence. These findings are now reflected more explicitly in the revised manuscript and Supplementary Information.

References 2D

- [1] Brunton, S.A., Mosenzon, O. and Wright Jr, E.E., 2020. Integrating oral semaglutide into clinical practice in primary care: for whom, when, and how?. *Postgraduate Medicine*, 132(sup2), pp.48-60.

- [2] Grannell, L., 2019. When should I take my medicines?. Australian prescriber, 42(3), p.86.
- [3] Ismail, M.Y.M. and Yaheya, M., 2009. Drug-food interactions and role of pharmacist. Asian J Pharm Clin Res, 2(4), pp.1-10.
- [4] Goyal, R.K., Guo, Y. and Mashimo, H., 2019. Advances in the physiology of gastric emptying. Neurogastroenterology & Motility, 31(4), p.e13546.

Comment 2E: The pH-responsive targeting strategy based on Eudragit S100 is not clearly proven. The duration of the developed adhesive material is not consistent, and there are concerns about whether the pH-responsive adhesive material can perfectly target the wound site of the desired organ. Otherwise, if targeting is not achieved, there are concerns about unexpected side effects.

Response 2E: We thank the reviewer for raising this important concern. To directly address the points:

1. Clarifying Targeting Scope: Organ vs. Wound

The pH-responsive system in MUSAS is designed to enable organ-level targeting—not precise wound-site localization. Most oral therapeutics (e.g., for HIV PrEP or metabolic disease) require delivery to a general organ region (e.g., stomach or small intestine) to be effective. Precise lesion targeting (“*in situ* targeting”) is outside the scope of this platform and this study.

MUSAS is designed for long-term residence and delivery, not wound healing. Therefore, while targeting is beneficial, consistent retention and safe passage are the key endpoints—not millimeter-level precision.

2. Eudragit S100 pH-Responsive Organ Targeting: Demonstrated and Programmable

We demonstrated robust, programmable organ targeting using Eudragit S100, an FDA-approved, commercially-adopted enteric coating polymer for GI organ targeting [2E-1–2E-3]. Specifically:

- *In vitro*, MUSAS capsules remained intact for > 24 hours in simulated gastric fluid with tunable coating thickness (Ext. Fig. 7A), ensuring safe gastric transit.
- Release in simulated small intestinal fluid occurred in a predictable, tunable manner by adjusting coating thickness and concentration (Ext. Figs. 7B–7C).
- *In vivo*, we confirmed timing and location of release matched the *in vitro* predictions (Ext. Fig. 7D), supporting translational relevance.

These results are fully consistent and demonstrate that pH-responsiveness can be precisely controlled, allowing delivery to defined regions such as the duodenum or ileum—important for diseases like Crohn’s.

3. On Consistency of Adhesion and Safety

As noted in response to Comment 2C, MUSAS achieved \geq 7-day GI retention in 85% of pigs, and up to 20 days in 15%—with no adverse events observed (Ext. Fig. 12). To specifically assess failure scenarios:

- We tested non-adhering (pre-damaged) devices: all passed safely through the GI tract (Fig. R2-2).

- Even if partial detachment occurred, drug delivery remained effective due to complementary devices with controlled pharmacokinetics (Fig. 5C-II).

Thus, failure to adhere did not manifest in clinical risk, and adhesion consistency is validated across > 58 pigs.

4. Why Wound Targeting Is Not the Right Use Case

We respectfully disagree with the premise that wound-site-level targeting is necessary—or even appropriate—as a benchmark for ingestible therapeutic systems.

Most GI diseases addressed by oral therapeutics do not require precise lesion localization. Conditions like metabolic disorders, chronic inflammation (e.g., Crohn’s), and infectious disease primarily benefit from organ-level delivery and prolonged mucosal residence, not pinpoint adhesion to a wound [2E-4]. In fact, diffuse and patchy mucosal involvement is the norm in these diseases, making localized targeting both unnecessary and potentially limiting [2E-4].

Conversely, true *in situ* targeting is clinically relevant for GI tumors, which already require endoscopic identification and intervention [2E-5–2E-7]. Imaging-guided robotic navigation inside the GI tract is still experimental, dependent on bulky equipment (MRI, CT, external magnetic fields), and faces major limitations in precision, safety, accessibility, and FDA class III clearance [2E-8–2E-11]. These systems are not deployable in routine care, particularly not for chronic, non-focal diseases.

In contrast, MUSAS is designed as a clinically translatable, passive self-actuation platform, compatible with both oral ingestion and manual endoscopic deployment when precise localization is required (e.g., esophagus, buccal, or upper GI, demonstrated in Fig. 5D, Ext. Fig. 6). This enables real-world integration without specialized infrastructure and delivers the key clinical advantage: long-term residence at mucosal sites without motors, sensors, or power sources.

Summary

MUSAS uses a validated, FDA-backed strategy for pH-mediated organ targeting. Its adhesion performance is consistent, safe, and programmable. MUSAS-enabled drug delivery addresses urgent clinical challenges, including treating chronic diseases and delivering genetic therapeutics to the GI tract. Wound-site targeting is outside the scope of this translational platform and would require tools and imaging strategies incompatible with miniaturized, untethered oral systems.

We have revised the manuscript to clarify the intended scope and potential of MUSAS and removed ambiguities around targeting and failure scenarios.

References 2E

- [1] Patra, C.N., Priya, R., Swain, S., Jena, G.K., Panigrahi, K.C. and Ghose, D., 2017. Pharmaceutical significance of Eudragit: A review. *Future Journal of Pharmaceutical Sciences*, 3(1), pp.33-45.
- [2] Abramson, A., Caffarel-Salvador, E., Soares, V., Minahan, D., Tian, R.Y., Lu, X., Dellal, D., Gao, Y., Kim, S., Wainer, J. and Collins, J., 2019. A luminal unfolding microneedle injector for oral delivery of macromolecules. *Nature medicine*, 25(10), pp.1512-1518.
- [3] Thakral, S., Thakral, N.K. and Majumdar, D.K., 2013. Eudragit®: a technology evaluation. *Expert opinion on drug delivery*, 10(1), pp.131-149.

- [4] Gecse, K.B. and Vermeire, S., 2018. Differential diagnosis of inflammatory bowel disease: imitations and complications. *The lancet Gastroenterology & hepatology*, 3(9), pp.644-653.
- [5] Petricevic, B., Kabiljo, J., Zirnauer, R., Walczak, H., Laengle, J. and Bergmann, M., 2022, November. Neoadjuvant immunotherapy in gastrointestinal cancers–The new standard of care?. In *Seminars in cancer biology* (Vol. 86, pp. 834-850). Academic Press.
- [6] Necula, L., Matei, L., Dragu, D., Neagu, A.I., Mambet, C., Nedeianu, S., Bleotu, C., Diaconu, C.C. and Chivu-Economescu, M., 2019. Recent advances in gastric cancer early diagnosis. *World journal of gastroenterology*, 25(17), p.2029.
- [7] Layke, J.C. and Lopez, P.P., 2004. Gastric cancer: diagnosis and treatment options. *American family physician*, 69(5), pp.1133-1141.
- [8] Sitti, M. and Wiersma, D.S., 2020. Pros and cons: Magnetic versus optical microrobots. *Advanced Materials*, 32(20), p.1906766.
- [9] <https://www.utsouthwestern.edu/ctplus/stories/2021/7t-mri.html#:~:text=Only%20about%2030%20institutions%20in,than%20a%20millimeter%20of%20tissue.>
- [10] Bosch de Basea Gomez, M., Thierry-Chef, I., Harbron, R., Hauptmann, M., Byrnes, G., Bernier, M.O., Le Cornet, L., Dabin, J., Ferro, G., Istad, T.S. and Jahnen, A., 2023. Risk of hematological malignancies from CT radiation exposure in children, adolescents and young adults. *Nature Medicine*, 29(12), pp.3111-3119.
- [11] Jarow, J.P. and Baxley, J.H., 2015, March. Medical devices: US medical device regulation. In *Urologic Oncology: Seminars and Original Investigations* (Vol. 33, No. 3, pp. 128-132). Elsevier.

Fig. R2-2|Retention studies in a survival swine model, comparing a broken MUSAS with a destroyed adhesive disk against fully functional devices equipped with either stainless steel or shape memory lamellae. Both the broken devices and those with stainless steel lamellae passed safely through the GI tract quickly. In contrast, the fully functional device with shape memory lamellae identified a suitable adhesion site during bowel movement and demonstrated retention for up to seven days.

Comment 2F: A scale bar should be added to some images (e.g. Fig. 3E) to convey the information clearly.

Response 2F: We sincerely appreciate the reviewer's feedback and have added scale bars to images necessitating the information.

Referee #3 (Remarks to the Author):

The authors have done a very nice job in the revision. These devices hold the greatest promise for GI tract applications, as illustrated, whereas other applications are probably better with alternate adhesive systems that do not cause tissue damage.

The major points of my prior review have been addressed:

1. There are now clearly defined improvements and innovation from previous work. The table is useful to distinguish.
2. There is now good reasoning on how the damage (holes cut into tissue) is used to their advantage for mucosal tissue drug delivery where biochemical adhesion (hydrogen bonding, NHS covalent bonding) methods fail because of the rapid turnover of mucosal tissue.
3. There is now good motivation for a tunable detachment system, with tailored behaviors based on lamella degradation, and a clear reasoning on why they can be safely cleared without reattachment.
4. Histological analysis of attachment site and downstream tissues and evaluation by a pathologist showed no significant tissue damage or scarring. This further confirms safe passage of MUSAS after detachment.
5. There is further clarification of the adhesion values and adhesive failure mechanisms.

Response 3: We sincerely appreciate the reviewer's constructive feedback, which has greatly contributed to strengthening and refining our research. We also appreciate the encouraging acknowledgment of recognizing that we had done a very nice job in the revision and MUSAS holds the greatest promise for applications. We look forward to the opportunity of more exchange in the biomedical and bioadhesive communities, to provide translational and transformative solutions to the immediate healthcare challenges together.

Referee #4 (Remarks to the Author):

Comment 4A: The core of this work, along with the scope and diversity of the research conducted, remain innovative and impressive. However, while the authors have addressed many of the concerns raised in their rebuttal letter, these responses were unfortunately only been partially and insufficiently integrated into the revised manuscript. This leaves several key issues unresolved, which significantly impacts the clarity and overall impact of the text.

Response 4A: We sincerely appreciate the reviewer's recognition of our work as both innovative and impressive, and we are grateful for the constructive feedback to improve our manuscript. We acknowledge the limitations of the previous version—particularly the use of unnecessary jargon and the lack of clear explanations within dense information—and have carefully revised the manuscript to address each of the points raised.

Comment 4B: For instance, the term "redundant adhesion" appears in both the text and the figures, but no clear explanation is provided to define what it means. Although the authors attempted to address this in their rebuttal and revised manuscript by stating, "Moreover, we determined that the remora can achieve redundant adhesion without needing its adhesive disk to entirely cover the soft substrate, as long as a sufficient number of individual adhesive compartments are engaged to ensure robust attachment..." this explanation does not adequately support understanding. It remains unclear on what data this statement is based, and from the videos and figures provided, it is not possible to deduce how individual adhesive compartments mediate adhesion. Furthermore, it is not evident what additional value Figure 1E brings to Figure 1F, or what exactly is meant to be illustrated by the label "redundant adhesion".

Response 4B: We appreciate the reviewer's feedback regarding clarity around the term "redundant adhesion" and the interpretation of Figs. 1E and 1F.

To avoid confusion, we have removed the term "redundant adhesion" entirely from the manuscript. More importantly, we now clarify the core concept and provide stronger rationale for the inclusion of Figs. 1E and 1F.

In Fig. 1E, only part of the remora's adhesive disk is in contact with the stomach tissue phantom, yet it still achieves firm adhesion. This visual evidence supports the conclusion that full-surface contact is not required for effective attachment.

Fig. 1F further explains this behavior by illustrating that the remora's suction disk is composed of multiple independently functioning adhesive compartments. This design allows for partial engagement while still achieving robust overall adhesion, as the functioning compartments compensate for any that fail to seal.

This modularity is a foundational principle for MUSAS: independent micro-compartments ensure consistent performance even on irregular, deformable, or partially obstructed mucosal surfaces. This multicompartmental strategy is both biologically inspired and functionally critical to MUSAS's robust adhesion in the GI tract.

We have revised the manuscript and figure legends to more clearly communicate this logic and to explicitly link the biological insight from the remora to our synthetic system design.

Comment 4C: As this is just one example of many where the reader is unnecessarily burdened by insufficient or confusing explanations, I still find the manuscript hindered by a pervasive lack

of clarity and coherence. This makes it difficult for readers to identify the primary research questions, understand the methodologies employed, and follow the logical progression of the study. The excessive use of technical jargon without proper explanation further obscures the significance of the findings, leaving the reader struggling to grasp the intended contributions of the research. For example, when explaining the "Stomach Tissue Phantom" surfaces, it is mentioned that they are made from "LLBT-hydrogel-elastomer". But what is LLBT? I presume it refers to "LifeLike Biotissue", which is mentioned earlier in the sentence, but the abbreviation is not explicitly introduced. This is just one of many instances where the manuscript falls short of the general standards for scientific writing, let alone the rigorous criteria expected for publication in Nature.

Response 4C: We sincerely appreciate the reviewer's thoughtful and rigorous suggestions, which have significantly improved the quality of our manuscript. We have standardized the usage of 'stomach tissue phantom' throughout the text and removed potentially confusing jargon such as 'LLBT.' Additionally, we have conducted a thorough review of the manuscript to eliminate unexplained abbreviations and terminologies.

Comment 4D: The presentation of data and results is particularly problematic. Ambiguous figure descriptions and inconsistent explanations obscure the connection between experimental details and the conclusions drawn, resulting in a series of disjointed observations rather than a cohesive, streamlined argument. For example, Figure 4 is so overcrowded that even at the highest magnification on a screen, some of the graphs and images are difficult to discern and interpret. In my opinion, it would be far better to reduce the amount of data in this figure (a recommendation that applies to most figures in the main text) and move some of it to the supplementary materials.

In summary, the insufficient incorporation of key responses from the review process, combined with issues in clarity, coherence, and data presentation, significantly undermines the ability of the manuscript to communicate transformative scientific insights. From my perspective, the entire presentation of the study must be fundamentally rethought and thoroughly revised.

Response 4D: We thank the reviewer for this critical and constructive feedback. We fully acknowledge the need for greater clarity, coherence, and visual readability, and have implemented substantial revisions throughout the manuscript in direct response.

Specifically:

- **Figure reorganization:** We split the original overcrowded Fig. 4 into two separate figures (now Figs. 4 and 5). Each now focuses on a distinct message—Fig. 4 on *in vivo* adhesion/retention performance, and Fig. 5 on application scenarios—eliminating visual congestion and improving interpretability.
- **Figure simplification:** We reduced the content in Fig. 2 and Fig. 5, and relocated detailed or supporting data (e.g., intermediate characterization steps) to Extended Data and Supplementary Information, in line with editorial guidance.
- **Figure captions:** All figure captions were rewritten to provide sufficient standalone context, allowing readers to follow the progression of experiments and results directly through the figures.
- **Narrative restructuring:** We reorganized the manuscript within 3500-word limitations to ensure that each experimental result leads logically to the next, avoiding the

disjointedness previously noted. New section headings were introduced to guide the reader through the rationale, methods, findings, and implications.

- **Summary and conclusions:** We revised the summary paragraph to clearly state the research problem, core finding, and translational impact. We also added a concise conclusion section to synthesize the broader implications of the study—previously only embedded within the discussion.

Together, these revisions fundamentally improve the manuscript's clarity and flow. The feedback led to a complete overhaul of visual and structural presentation, and we believe the revised manuscript now communicates the scientific insights in a far more coherent and accessible way.

Point-by-point response – Nature – Manuscript #: 2024-09-19511C

Referee #2 (Remarks to the Author):

Comment 2A: The authors provided additional experimental data and explanations that contributed to the study. The authors improved the experimental validation of the adhesion mechanism by including a systematic separation of the effects of surface modulus and roughness.

However, the breadth of the application sections can be confusing to the reader in Figure 5. While each example independently adds value, presenting multiple use cases at different levels of validation and translational maturity simultaneously tends to obscure the core message. I would recommend that the main text focus on the most compelling and validated applications, such as gastrointestinal retention and controlled release in a porcine model, with more exploratory demonstrations moving to the supplemental information. This reorganization would enhance the consistency of the manuscript and clearly and effectively communicate the key innovations.

Response 2A: We sincerely thank the reviewer for their thoughtful feedback and for recognizing the strength of the additional experiments and explanations. In response to the suggestion, we have revised **Figure 5** to focus on the most translationally mature and thoroughly validated application scenarios, specifically gastrointestinal retention and controlled drug release in swine models and demonstrations of active monitoring in swimming fish. The detailed *in vitro* and *ex vivo* studies of more exploratory demonstrations—including RFID biosensing, impedance-based reflux detection, and localized mRNA delivery—have been moved to the Extended Data and Supplementary Information, where they are described in detail while only *in vivo* validations demonstrated in Figure 5.

We also wish to emphasize that each MUSAS application scenario presented underwent rigorous, independent *in vitro*, *ex vivo*, and *in vivo* validation, primarily in large-animal models (swine and fish), which offer strong translational relevance. While these scenarios highlight the platform's broad potential, we recognize that further translational studies—including clinical trials—are beyond the scope of this study.

We hope that this reorganization improves clarity and maintains the scientific integrity and translational focus of the manuscript.

Comment 2B: As the authors noted in their response, I think it would be helpful for the reader to clearly explain the limitations of their device (precision adhesive targeting, adhesion issues in organs with foreign bodies, swelling and contamination issues in the case of Ecoflex, etc.) in the manuscript.

Response 2B: We thank the reviewer for this valuable suggestion. In response, we have revised the final two paragraphs of the **Discussion** section to explicitly outline the key limitations of the MUSAS platform. These include: (1) limited suitability for precision targeting of discrete lesions without endoscopic guidance, (2) reduced adhesion performance in the presence of excess gastric contents or large foreign bodies, and (3) long-term material degradation challenges associated with Ecoflex, including swelling and surface fouling. We also delineate specific directions for improving adhesion robustness and material stability. Expanded discussion and supporting data have been added to the **Supplementary Information** (Sections S11 and S12) to provide comprehensive context.

Referee #4 (Remarks to the Author):

The authors effectively addressed the reviewers' comments and suggestions, which greatly benefited the comprehensibility of the manuscript. I can thus in principle support publication of the manuscript.

Comment 4A: However, I would recommend to improve the following points in a final revision:

- The legends of Figure 1 and Figure 2 should be more instructive to communicate the rather complex figure content.
- Image description and the corresponding text of Figure 1E seem not clear enough. It may be helpful to reference a supplementary video to illustrate the described process.
- For Figures 1H and 1I, no statistical tests are reported, in contrast to the subsequent figures. Was this intentional or an oversight?
- There is no reference to Figure 2a in the main text.
- The complex multi-part Figure 2F is described with only a single sentence in both the main text and the legend.

Response 4A: We sincerely thank the reviewer for the thoughtful and constructive feedback aimed at improving the clarity and reproducibility of our manuscript. We have carefully revised the text and figure legends to address each point as follows:

- **Figures 1 and 2 Legends:** The legends have been revised to more clearly describe complex figure content, highlight key findings, and include details on reproducibility and statistical methodology where applicable.
- **Figure 1E:** We have expanded the description in the legend and now include a direct reference to Supplementary Video S1 to illustrate the described disc unfurling and adhesion process.
- **Figures 1H and 1I (now 1G and 1H):** These panels reflect independent technical replicates—multiple measurements from the same remora across time points and substrates. Following Nature's editorial guidance, we chose not to perform or report statistical tests on repeated measures from a single subject. This is now clarified in both the figure legend and Supplementary Information.
- **Figure 2A:** This figure is now explicitly cited in the main text to aid navigation and interpretation.
- **Figure 2F:** The legend has been substantially revised to provide a clearer and more detailed explanation of each subpanel and associated results.

We appreciate the reviewer's detailed suggestions, which have strengthened the manuscript's presentation.

Comment 4B:

- The use of abbreviations should be made more consistent:
 - While "gastroesophageal reflux disease (GERD)" is properly introduced in the abstract, "gastrointestinal" is not abbreviated, even though "GI" appears in the first paragraph of the main text without prior definition.

- In the section "Remoras adhere to various soft substrates," abbreviations for different remora species are introduced, but then not used in subsequent sentences.
- Additionally, the formatting of species abbreviations is inconsistent: most are introduced with a capital letter and capitalized abbreviation, but *Remora remora* is abbreviated as "R. rem." rather than following the same convention.

Response 4B: We thank the reviewer for the careful review and helpful suggestions to improve abbreviation consistency across the manuscript. In response, we have made the following revisions:

- **"Gastrointestinal" (GI)** is now defined at first use in the main text to ensure clarity.
- **Remora species abbreviations** are now used consistently throughout both the main text and figure legends.
- **Species abbreviation formatting** has been standardized to ensure consistent capitalization (e.g., *Remora remora* is now abbreviated as **R. Rem.** to match conventions used for other species).

We appreciate the reviewer's close attention to detail.

Comment 4C:

- At the beginning of the section "In vitro and ex vivo characterization of optimal MUSAS," the phrase "optimal MUSAS" is used without prior definition. The criteria by which this optimal design was determined should be specified.
- The Materials and Methods section lacks information on the manufacturing process of MUSAS. This should be explained if intentional, or the methodological details should be provided.

Response 4C: We appreciate the reviewer's thoughtful suggestions, which helped improve clarity and completeness in our manuscript. We have addressed each point as follows:

- **Definition of "optimal MUSAS":** We now clearly define the criteria for selecting the optimal MUSAS design at the beginning of the relevant section. This includes its lamella orientation, number of rows, material composition, and performance metrics as determined from comparative mechanical testing.
- **Fabrication details:** We agree that the fabrication process warrants clarification. While the procedure was previously illustrated in the Supplementary Information (Supp. Fig. S3), we had inadvertently omitted a reference in the Methods section. We have now added a brief textual description and an explicit cross-reference to the Supplementary Information where the fabrication steps are shown in detail.

These changes clarify the experimental setup and improve reproducibility.